# Fast Frank–Wolfe Algorithms with Adaptive Bregman Step-Size for Weakly Convex Functions

**Shota Takahashi**
The University of Tokyo
shota@mist.i.u-tokyo.ac.jp

**Sebastian Pokutta**
Zuse Institute Berlin &
Technische Universität Berlin
pokutta@zib.de

**Akiko Takeda**
The University of Tokyo & RIKEN AIP
takeda@mist.i.u-tokyo.ac.jp

## Abstract

We propose Frank–Wolfe (FW) algorithms with an adaptive Bregman step-size strategy for smooth adaptable (also called: relatively smooth) (weakly-) convex functions. This means that the gradient of the objective function is not necessarily Lipschitz continuous, and we only require the smooth adaptable property. Compared with existing FW algorithms, our assumptions are less restrictive. We establish convergence guarantees in various settings, including convergence rates ranging from sublinear to linear, depending on the assumptions for convex and nonconvex objective functions. Assuming that the objective function is weakly convex and satisfies the local quadratic growth condition, we provide both local sublinear and local linear convergence with respect to the primal gap. We also propose a variant of the away-step FW algorithm using Bregman distances over polytopes. We establish faster global convergence (up to a linear rate) for convex optimization under the Hölder error bound condition and local linear convergence for nonconvex optimization under the local quadratic growth condition. Numerical experiments demonstrate that our proposed FW algorithms outperform existing methods.

## 1 Introduction

In this paper, we consider constrained optimization problems of the form

$$\min_{x \in P} \quad f(x), \tag{1.1}$$

where $f : \mathbb{R}^n \to (-\infty, +\infty]$ is a continuously differentiable function and $P \subset \mathbb{R}^n$ is a compact convex set. We are interested in both convex and nonconvex $f$ and assume that we have first-order oracle access to $f$, *i.e.*, given $x \in \mathbb{R}^n$, we can compute $\nabla f(x)$. The Frank–Wolfe (FW) algorithm is a projection-free first-order method. Instead of requiring access to a projection oracle, the FW algorithm requires only access to a so-called linear minimization oracle (LMO), which for a given linear function $a \in \mathbb{R}^n$ computes $y \in \arg\min_{v \in P} \langle a, v \rangle$. LMOs are often much cheaper than projection oracles, as shown in Combettes & Pokutta (2021) (see also (Braun et al., 2025, Table 1.1)). Consequently, in practice, FW algorithms are often faster than projected gradient methods even when the projection operation is nontrivial. Additionally, FW algorithms tend to be numerically quite robust and stable due to their affine-invariance and can also be used, *e.g.*, to provide theoretical guarantees for the approximate Carathéodory problem by Combettes & Pokutta (2023).

**Overview of Existing FW Studies:** The FW algorithm was originally proposed by Frank & Wolfe (1956) and was independently rediscovered and extended by Levitin & Polyak (1966) as the conditional gradient method; we will use these terms interchangeably. Canon & Cullum (1968) established an initial lower bound for the rate of the FW algorithm. GuéLat & Marcotte (1986) improved

this bound and also provided the first analysis of the away-step FW algorithm by Wolfe (1970). Jaggi (2013) provided a more detailed convergence analysis of the FW algorithm, establishing a new lower bound that demonstrated a trade-off between sparsity and error. Concurrently, Lan (2013) examined the complexity of linear programming-based first-order methods, establishing a similar lower bound. For a comprehensive review of FW algorithms, we refer the interested reader to the survey by Braun et al. (2025) and the brief introduction by Pokutta (2024).

The classical FW algorithm (Algorithm 1) has been improved in two main directions to enhance both performance and convergence. One focuses on refining the step-size rule. The short-step strategy computes $\gamma_t$ using the Lipschitz constant $L$ of $\nabla f$. Pedregosa et al. (2020) introduced an adaptive step-size strategy that dynamically estimates $L$. They proved that this strategy is at least as effective as the short-step strategy. It was enhanced by Pokutta (2024) to improve numerical stability.

Another direction is to modify the classical FW algorithm to eliminate the zigzag behavior when approaching the optimal face containing the optimal solution $x^*$. This behavior inspired Wolfe (1970) to propose the away-step FW algorithm in 1970 that shortcuts the zigzagging by removing atoms that slow down the iterative sequence. Lacoste-Julien & Jaggi (2015) showed the linear convergence of the away-step FW algorithm. Their analysis[1] introduced a geometric constant, the pyramidal width, that measures the conditioning of the polytope $P$, representing the feasible region.

**Our Research Idea:** To derive the convergence rate of the above-mentioned FW algorithms, previous studies have typically required $L$-smoothness, *i.e.*, that $\nabla f$ is Lipschitz continuous, and convexity for the objective function $f$, although exceptions exist.[2] These assumptions often narrow the application of FW algorithms. Even if they do not satisfy $L$-smoothness, there are many functions that satisfy the $L$-smooth adaptable property ($L$-smad) with kernel generating distances $\phi$ (see Definition 2.1) by choosing $\phi$ well to match the function $f$. Since the $L$-smad property is consistent with $L$-smoothness when $\phi = \frac{1}{2}\|\cdot\|^2$, the class of $L$-smad functions includes the $L$-smooth function class. Other first-order algorithms, such as the Bregman proximal gradient algorithm, are often analyzed under the $L$-smad property (see Appendix A and Bolte et al. (2018); Hanzely et al. (2021); Rebegoldi et al. (2018); Takahashi et al. (2022); Takahashi & Takeda (2025); Yang & Toh (2025)). $L$-smad functions appear in many applications, such as nonnegative linear inverse problems (Bauschke & Borwein, 1997; Takahashi & Takeda, 2025), $\ell_p$ loss problems (Kyng et al., 2015; Maddison et al., 2021), phase retrieval (Bolte et al., 2018; Takahashi et al., 2022), nonnegative matrix factorization (NMF) (Mukkamala & Ochs, 2019; Takahashi et al., 2026), and blind deconvolution (Takahashi et al., 2023). Any $\mathcal{C}^2$ function is locally $L$-smooth over compact sets, but the resulting Lipschitz constant $L$ can be overly conservative, hindering practical performance. Functions outside $\mathcal{C}^2$ may not satisfy $L$-smoothness at all. Moreover, functions in the above-mentioned applications are weakly convex on compact sets. For nonconvex functions, although Lacoste-Julien (2016) established sublinear convergence, little else has been done, and simply tracing existing research does not achieve convergence rates better than sublinear convergence. Then the following question naturally arises:

*Is it possible to relax the L-smoothness and convexity assumptions often made in FW algorithm studies while still achieving the theoretical guarantee of linear convergence rates?*

**Contribution:** The answer to the above question is yes. We develop FW algorithms for $L$-smad $f$ and give theoretical guarantees of linear convergence under weaker assumptions (Hölder error bound (HEB) condition with parameter $q \geq 1$ for convex $f$ and the special case with $q = 2$ called quadratic growth for nonconvex $f$) than the strong convexity assumption that is often made when linear convergence is shown for FW algorithms. The convexity assumption on $f$ is also weakened to weakly convex, *i.e.*, $f + \frac{\rho}{2}\|\cdot\|^2$ is convex with some $\rho > 0$ (see, *e.g.*, (Davis et al., 2018, Examples 3.1 and 3.2) and Nurminskii (1975)). Any $\mathcal{C}^2$ function on a compact set is weakly convex (Vial, 1983; Wu, 2007) (see Proposition C.1); since $\rho$ is only for the theoretical guarantee and not for the

---

[1]Similar analyses based on the pyramidal width have been developed for other advanced variants, such as the pairwise FW algorithm (Lacoste-Julien & Jaggi, 2015), Wolfe's algorithm (Lacoste-Julien & Jaggi, 2015), the blended conditional gradient method (Braun et al., 2019), and blended pairwise conditional gradient methods and similar variants (Combettes & Pokutta, 2020; Tsuji et al., 2022). Even Nesterov-style acceleration is possible under weak assumptions (Diakonikolas et al., 2020) building upon this analysis.

[2]Quite recently, Vyguzov & Stonyakin (2025) proposed a FW algorithm with the Bregman distance, similar to ours, yet it is restricted to convex problems and provides only partial convergence guarantees.

**Table 1:** Our convergence rates are generalizations of existing rates. Moreover, our convergence rate is better than $\mathcal{O}(\epsilon^{-1})$ in some cases. For nonconvex functions, convergence is measured using the Frank–Wolfe gap $\langle \nabla f(x_t), x_t - v_t \rangle \leq \epsilon$, instead of the primal gap $f(x_t) - f^* \leq \epsilon$. Convergence rates for weakly convex optimization hold locally.

| FW | Assumptions[3] | | | | Convergence rate | |
| | $f$ conv. | $f$ growth | IP | poly. | $L$-smooth | $L$-smad ($\nu > 0$) |
|---|---|---|---|---|---|---|
| Alg.1 (any step-size) | conv. | ✗ | ✗ | ✗ | $\mathcal{O}(\epsilon^{-1})$ | $\mathcal{O}(\epsilon^{-1/\nu})$ (Thm.D.2) |
| Alg.1 (short & adapt.) | conv. | $q$-HEB | ✓ | ✗ | $\mathcal{O}(\log \epsilon^{-1}), \mathcal{O}(\epsilon^{\frac{2-q}{q}})^4$ | $\mathcal{O}(\log \epsilon^{-1}), \mathcal{O}(\epsilon^{\frac{1+\nu-q}{\nu q}})$ (Thm. 4.2)[5] |
| away-step FW | conv. | $q$-HEB | ✗ | ✓ | $\mathcal{O}(\log \epsilon^{-1}), \mathcal{O}(\epsilon^{\frac{2-q}{q}})^4$ | $\mathcal{O}(\log \epsilon^{-1}), \mathcal{O}(\epsilon^{\frac{1+\nu-q}{\nu q}})$ (Thm. 4.4)[5] |
| Alg.1 (any step-size) | weak | 2-HEB | ✗ | ✗ | $\mathcal{O}(\epsilon^{-1})$ | $\mathcal{O}(\epsilon^{-1/\nu})$ (Thm. E.3) |
| Alg.1 (short & adapt.) | weak | 2-HEB | ✓ | ✗ | $\mathcal{O}(\log \epsilon^{-1})$ | $\mathcal{O}(\log \epsilon^{-1}), \mathcal{O}(\epsilon^{\frac{\nu-1}{2\nu}})$ (Thm. 5.3)[6] |
| away-step FW | weak | 2-HEB | ✗ | ✓ | $\mathcal{O}(\log \epsilon^{-1})$ | $\mathcal{O}(\log \epsilon^{-1}), \mathcal{O}(\epsilon^{\frac{\nu-1}{2\nu}})$ (Thm. 5.4)[6] |
| Alg.1 (any step-size) | ✗ | ✗ | ✗ | ✗ | $\mathcal{O}(\epsilon^{-2})$ | $\mathcal{O}(\epsilon^{-1-1/\nu})$ (Thm. E.1) |

algorithm, there is no need for its estimation, and considering weakly convex $f$ greatly broadens applicability.

Table 1 summarizes our contributions for the various cases that we consider. Relaxing the assumption from $L$-smoothness to the $L$-smad property extends the FW algorithm to a broader range of problems, while making the construction of the FW algorithm and its theoretical guarantee much more difficult. This is because we use the Bregman distance $D_\phi$, which is an extension of the Euclidean distance, for the $L$-smad property. While Equation 2.1 shown later is simple with $\nu = 1$ when using the Euclidean distance, $\nu > 0$ cannot be eliminated for the general Bregman distance; $1 + \nu$ represents its scaling exponent. $\nu$ needs to be estimated during the algorithm since its exact value is unknown, and thus, the extension to an $L$-smad $f$ is not straightforward.

- We establish sublinear convergence for the case of convex $L$-smad functions. Assuming that $f$ is convex and satisfies the HEB condition, and that the minimizer $x^* \in \text{int } P$, we provide faster convergence (up to a linear rate). In this setting, our algorithm always converges linearly if the exponent of the HEB condition $q$ equals the scaling exponent of the Bregman distance $1 + \nu$, i.e., $q = 1 + \nu$.[7] For $q > 1 + \nu$, if $t \leq t_0$ with some $t_0 \in \mathbb{N}$, linear convergence holds; otherwise, $\mathcal{O}(\epsilon^{(1+\nu-q)/\nu q})$, which is faster than existing sublinear rates. A similar argument also applies to the nonconvex case ($q = 2$) and the away-step FW algorithm.

- For nonconvex optimization, assuming that $f$ is weakly convex and satisfies the local quadratic growth condition, we establish local linear convergence. To the best of our knowledge, this is the first FW algorithm with linear convergence guarantees for a certain class of nonconvex optimization problems, albeit for a rather restricted subclass. These results are also new for the standard setting: $L$-smooth $f$. Without weak convexity, we prove global sublinear convergence to a stationary point of Equation 1.1 for nonconvex $L$-smad $f$.

- We also propose a variant of the away-step FW algorithm with the Bregman distance and establish its linear convergence under the HEB condition for convex optimization and under the local quadratic growth condition for weakly convex optimization.

---

[3] $f$ conv. means the convexity of $f$, IP means $x^* \in \text{int } P$, and poly. means that $P$ is a polytope.

[4] $\mathcal{O}(\log \epsilon^{-1})$ if $q = 2$ or if $q \neq 2$ and $t \leq t_0$ with $t_0 \in \mathbb{N}$; otherwise $\mathcal{O}(\epsilon^{(2-q)/q})$ (Kerdreux et al., 2022).

[5] $\mathcal{O}(\log \epsilon^{-1})$ if the exponent of the HEB condition $q$ equals the scaling exponent of the Bregman distance $1 + \nu$ ($(\nu, q) = (1, 2)$ for the away-step FW algorithm). For $q > 1 + \nu$, it holds if $t \leq t_0$ with $t_0 \in \mathbb{N}$; otherwise $\mathcal{O}(\epsilon^{(1+\nu-q)/\nu q})$.

[6] $\mathcal{O}(\log \epsilon^{-1})$ holds if $\nu = 1$. For $\nu \in (0, 1)$, it holds if $t \leq t_0$ with $t_0 \in \mathbb{N}$; otherwise $\mathcal{O}(\epsilon^{(\nu-1)/2\nu})$.

[7] As discussed in Sec. 2, linear convergence with the Euclidean distance is shown in Garber & Hazan (2015) under the quadratic growth over strongly convex sets. This corresponds to the $q = 1 + \nu$ case with $\nu = 1$.

## 2 PRELIMINARIES

Let $C$ be a nonempty open convex subset of $\mathbb{R}^n$. Let $\mathrm{dist}(x, C) := \inf_{y \in C} \|x - y\|$ and let $\mathcal{C}^k$ be the class of $k$-times continuously differentiable functions for $k \geq 0$.

**Definition 2.1** (Kernel Generating Distance (Bolte et al., 2018))**.** A function $\phi : \mathbb{R}^n \to (-\infty, +\infty]$ is called a *kernel generating distance* associated with $C$ if it satisfies the following conditions: (i) $\phi$ is proper, lower semicontinuous, and convex, with $\mathrm{dom}\, \phi \subset \mathrm{cl}\, C$ and $\mathrm{dom}\, \partial \phi = C$; (ii) $\phi$ is $\mathcal{C}^1$ on $\mathrm{int}\, \mathrm{dom}\, \phi \equiv C$. We denote the class of kernel generating distances associated with $C$ by $\mathcal{G}(C)$.

**Definition 2.2** (Bregman Distance (Bregman, 1967))**.** Given $\phi \in \mathcal{G}(C)$, a *Bregman distance* $D_\phi :$ $\mathrm{dom}\, \phi \times \mathrm{int}\, \mathrm{dom}\, \phi \to \mathbb{R}_+$ associated with $\phi$ is defined by $D_\phi(x, y) := \phi(x) - \phi(y) - \langle \nabla \phi(y), x - y \rangle$.

The Bregman distance $D_\phi(x, y)$ measures the proximity between $x \in \mathrm{dom}\, \phi$ and $y \in \mathrm{int}\, \mathrm{dom}\, \phi$. Since $\phi$ is convex, $D_\phi(x, y) \geq 0$ holds. Moreover, when $\phi$ is strictly convex, $D_\phi(x, y) = 0$ holds if and only if $x = y$. However, the Bregman distance is not always symmetric and does not have to satisfy the triangle inequality. See Example B.1. We use $D_f$ in Definition 2.2 for nonconvex $f$ instead of convex $\phi$ for brevity. In the remainder of the paper, we assume that, for a given strictly convex $\phi$ and $C$, there exists $\nu > 0$ such that the following holds for all $x, y \in \mathrm{int}\, \mathrm{dom}\, \phi$ and $\gamma \in [0, 1]$ (see Lemma B.3):

$$D_\phi((1 - \gamma)x + \gamma y, x) \leq \gamma^{1+\nu} D_\phi(y, x). \tag{2.1}$$

We recall the smooth adaptable property, which is a generalization of $L$-smoothness and was introduced by Bauschke et al. (2017); Bolte et al. (2018); Lu et al. (2018). Its characterizations can be found in Appendix B.2, Example B.5, Remark B.6, and (Bauschke et al., 2017, Proposition 1).

**Definition 2.3** ($L$-smooth Adaptable Property)**.** Let $\phi \in \mathcal{G}(C)$ and let $f : \mathbb{R}^n \to (-\infty, +\infty]$ be proper and lower semicontinuous with $\mathrm{dom}\, \phi \subset \mathrm{dom}\, f$, which is $\mathcal{C}^1$ on $C \equiv \mathrm{int}\, \mathrm{dom}\, \phi$. The pair of functions $(f, \phi)$ is said to be *L-smooth adaptable* (for short: *L-smad*) on $C$ if there exists $L > 0$ such that $L\phi - f$ and $L\phi + f$ are convex on $C$.

The $L$-smad property is equivalent to the extended descent lemma (Bolte et al., 2018, Lemma 2.1), which implies that the $L$-smad property for $(f, \phi)$ provides upper and lower approximations for $f$ majorized by $\phi$ with $L > 0$. For example, $-\log x$ is not $L$-smooth on $x > 0$ and $\frac{1}{4}x^4$ is not $L$-smooth on $\mathbb{R}$. It is the Bregman distance, rather than the Euclidean distance, that allows us to bound the first-order approximation of these functions. See Example B.5 for further examples of $L$-smad pairs. When $(f, \phi)$ is $L$-smad with a strictly convex function $\phi \in \mathcal{G}(C)$, we have Lemma B.7, *i.e.*,

$$f(x) - f(x_+) \geq \gamma \langle \nabla f(x), x - v \rangle - L\gamma^{1+\nu} D_\phi(v, x), \tag{2.2}$$

where $x_+ = (1 - \gamma)x + \gamma v$ with $x, v \in \mathrm{int}\, \mathrm{dom}\, \phi$ and $\gamma \in [0, 1]$. Often $v \in P \subset \mathrm{int}\, \mathrm{dom}\, \phi$ in Lemma B.7 is chosen as a *Frank–Wolfe vertex*, *i.e.*, $v \in \arg\max_{u \in P} \langle \nabla f(x), x - u \rangle$, but other choices, *e.g.*, those arising from away-directions, are also possible, as we will see in Section 3.2.

In Garber & Hazan (2015) (see also Garber (2020)) the linear convergence of the FW algorithm for convex optimization under the quadratic growth condition, which is a weaker assumption than assuming strong convexity, was established over strongly convex sets and later generalized to uniformly convex sets in Kerdreux et al. (2021) as well as to conditions weaker than quadratic growth in Kerdreux et al. (2019; 2022). Local variants of these notions, as necessary, *e.g.*, for the nonconvex case, have been studied in Kerdreux et al. (2026) in the context of FW algorithms. For $f : \mathbb{R}^n \to [-\infty, +\infty]$, let $[f \leq \zeta] := \{x \in \mathbb{R}^n \mid f(x) \leq \zeta\}$ be a $\zeta$-sublevel set of $f$ for some $\zeta \in \mathbb{R}$.

**Definition 2.4** (Hölder Error Bound (Braun et al., 2025; Roulet & d'Aspremont, 2017) and Quadratic Growth Conditions (Garber & Hazan, 2015; Garber, 2020; Liao et al., 2024))**.** Let $f : \mathbb{R}^n \to (-\infty, +\infty]$ be a proper lower semicontinuous function and $P \subset \mathbb{R}^n$ be a compact convex set. Let $\mathcal{X}^* \neq \emptyset$ be the set of optimal solutions, *i.e.*, $\mathcal{X}^* := \arg\min_{x \in P} f(x)$, and let $f^* = \min_{x \in P} f(x)$ and $\zeta > 0$. The function $f$ satisfies the *q-Hölder error bound* (HEB) condition on $P$ if there exist constants $q \geq 1$ and $\mu > 0$ such that $\mathrm{dist}(x, \mathcal{X}^*)^q \leq \frac{q}{\mu}(f(x) - f^*)$ holds for any $x \in P$. In particular, when $q = 2$, it is called the $\mu$-*quadratic growth* condition. Moreover, the function $f$ is said to satisfy *local $\mu$-quadratic growth* (with $\zeta$) if there exists a constant $\mu > 0$ such that $\mathrm{dist}(x, \mathcal{X}^*)^2 \leq \frac{2}{\mu}(f(x) - f^*)$ holds for any $x \in [f \leq f^* + \zeta] \cap P$.

For example, $f(x) = \log(1 + x^2)$ is nonconvex but satisfies local quadratic growth. This function is used for image restoration (Boţ et al., 2016; Stella et al., 2017). See, for more examples, Examples E.4 and E.5. The HEB condition shows sharpness bounds on the primal gap (see Bolte et al. (2017); Roulet & d'Aspremont (2017)), which has been extensively analyzed for FW algorithms (Kerdreux et al., 2019; 2022). Convergence of (sub)gradient algorithms for nonconvex optimization under the quadratic growth or HEB condition was established in Davis et al. (2018; 2024); Davis & Jiang (2024). We assume the HEB condition for convex optimization and the local quadratic growth for nonconvex optimization.

## 3 PROPOSED BREGMAN FW ALGORITHMS

Throughout this paper, we make the following assumptions.

**Assumption 3.1.** (i) $\phi \in \mathcal{G}(C)$ with $\operatorname{cl} C = \operatorname{cl} \operatorname{dom} \phi$ is strictly convex on $C \equiv \operatorname{int} \operatorname{dom} \phi$ and has $\nu > 0$ satisfying Equation 2.1; (ii) $f : \mathbb{R}^n \to (-\infty, +\infty]$ is proper and lower semicontinuous with $\operatorname{dom} \phi \subset \operatorname{dom} f$, which is $\mathcal{C}^1$ on $C$; (iii) The pair $(f, \phi)$ is $L$-smad on $P$; (iv) $P \subset \mathbb{R}^n$ is a nonempty compact convex set with $P \subset C$.

Assumption 3.1(i)-(iii) are standard for Bregman-type algorithms (Bolte et al., 2018; Takahashi & Takeda, 2025). The strict convexity of $\phi$ ensures the existence of $\nu$ by Lemma B.3 and is satisfied by many kernel generating distances (see, *e.g.*, Bauschke et al. (2017); Bauschke & Borwein (1997), and (Dhillon & Tropp, 2008, Table 2.1)). Assumption 3.1(iv) is standard for

**Algorithm 1:** Frank–Wolfe algorithm
**Input:** Initial point $x_0 \in P$, objective function $f$, step-size $\gamma_t \in [0, 1]$
1 **for** $t = 0, \dots$ **do**
2 $\quad v_t \leftarrow \arg\min_{v \in P} \langle \nabla f(x_t), v \rangle$
3 $\quad x_{t+1} \leftarrow (1 - \gamma_t) x_t + \gamma_t v_t$

FW algorithms, and $P \subset C$ naturally ensures that $D_\phi$ is well-defined on $P$. The original FW algorithm is given in Algorithm 1. In what follows, let $x^* \in \arg\min_{x \in P} f(x)$ and $f^* = f(x^*)$.

### 3.1 BREGMAN STEP-SIZE STRATEGY FOR $\gamma_t$

**Bregman Short Step-Size:** Assume that $(f, \phi)$ is $L$-smad. Maximizing the right-hand side of Equation 2.2 in terms of $\gamma$, we find

$$\gamma = \left( \frac{\langle \nabla f(x), x - v \rangle}{L(1 + \nu) D_\phi(v, x)} \right)^{\frac{1}{\nu}} \quad \text{and} \quad f(x) - f(x_+) \geq \frac{\nu}{1 + \nu} \frac{\langle \nabla f(x), x - v \rangle^{1 + 1/\nu}}{(L(1 + \nu) D_\phi(v, x))^{1/\nu}}.$$

Theoretically, when $x, v \in P$ and $\gamma \in [0, 1]$, the Frank–Wolfe step $x_+ = (1 - \gamma)x + \gamma v \in P$ is well-defined. We update $\gamma = \min \left\{ \left( \frac{\langle \nabla f(x), x - v \rangle}{L(1 + \nu) D_\phi(v, x)} \right)^{1/\nu}, \gamma_{\max} \right\}$ with some $\gamma_{\max} \in \mathbb{R}$ (usually set $\gamma_{\max} = 1$). This step-size strategy provides $f(x) - f(x_+) \geq \frac{\nu}{1 + \nu} \gamma \langle \nabla f(x), x - v \rangle$ for $0 \leq \gamma \leq \left( \frac{\langle \nabla f(x), x - v \rangle}{L(1 + \nu) D_\phi(v, x)} \right)^{1/\nu}$. Its proof can be found in Lemma B.8.

If $\phi = \frac{1}{2} \| \cdot \|^2$, we have $\nu = 1$ and $\gamma_t = \frac{\langle \nabla f(x_t), x_t - v_t \rangle}{L \| v_t - x_t \|^2}$, which is often called the (Euclidean) short step-size. Although the short step-size does not require line searches, it requires knowledge of the value of $L$.

**Adaptive Bregman Step-Size:** The exact value or the tight upper bound of $L$ is often unknown; however, the algorithm's performance heavily depends on it; an underestimation of $L$ might lead to non-convergence, and an overestimation of $L$ might lead to slow convergence. Moreover, a worst-case $L$ might be too conservative for regimes where the function is better behaved. For these reasons, Pedregosa et al. (2020) proposed an adaptive step-size strategy for FW algorithms in the case where $\phi = \frac{1}{2} \| \cdot \|^2$. We present our algorithm in Algorithm 2. This step-size strategy can be used as a drop-in replacement. For example, inserting $L_t, \nu_t, \gamma_t \leftarrow \texttt{step\_size}(f, \phi, x_t, v_t, L_{t-1}, 1)$ between lines 2 and 3 in Algorithm 1, we obtain that Algorithm 2 searches $L$ and $\nu$ satisfying Equation 2.2. For $\phi = \frac{1}{2} \| \cdot \|^2$, Algorithm 2 corresponds to a (Euclidean) adaptive step-size strategy (Pedregosa et al., 2020).

**Remark 3.2** (Well-definedness and termination of Algorithm 2)**.** By the $L$-smad property of $(f, \phi)$ and Lemma B.7, we know that Equation 2.2 holds for all $M \geq L$ and $\nu \geq \kappa > 0$. Therefore, the

condition in line 7 in Algorithm 2 is well-defined and guaranteed to terminate. It suffices to update $\kappa$ in line 10 only if Equation 2.1 does not hold. Thus, $\kappa$ cannot become too small.

We have a bound on the total number of evaluations of Algorithm 2. Its proof can be found in Appendix B.3.

**Theorem 3.3.** Let $L_{-1}$ be the initial $L$-smad estimate and $n_t$ be the total number of evaluations of Equation 2.2 up to iteration $t$. Then we have $n_t \leq \max\{(1 - \log\eta/\log\tau)(t + 1) + \max\{\log(\tau L/L_{-1}), 0\}/\log\tau, (1 + \log\nu/\log\beta)(t+1)\}$.

Pedregosa et al. (2020) showed that, asymptotically, no more than 16% of the iterations require more than one evaluation of the line search procedures when $\eta = 0.9$ and $\tau = 2$. This follows from the bound $(1 - \log\eta/\log\tau) \leq 1.16$.

---

**Algorithm 2:** Adaptive Bregman step-size strategy

**Output:** Estimates $\tilde{L}^*$ and $\tilde{\nu}^*$, step-size $\gamma$

1 **Procedure** step_size($f, \phi, x, v, \tilde{L}, \gamma_{\max}$)
2     Choose $\beta \in (0, 1), \eta \in (0, 1]$, and $\tau > 1$
3     $M \leftarrow \eta\tilde{L}, \kappa \leftarrow 1$
4     **loop**
5        $\gamma \leftarrow \min\left\{ \left(\frac{\langle \nabla f(x), x-v \rangle}{M(1+\kappa)D_\phi(v,x)}\right)^{1/\kappa}, \gamma_{\max} \right\}$
6        $x_+ \leftarrow (1-\gamma)x + \gamma v$
7        **if** $D_f(x_+, x) \leq M\gamma^{1+\kappa}D_\phi(v,x)$ **then**
8           $\tilde{L}^* \leftarrow M, \tilde{\nu}^* \leftarrow \kappa$
9           **return** $\tilde{L}^*, \tilde{\nu}^*, \gamma$
10        $M \leftarrow \tau M, \kappa \leftarrow \beta\kappa$

---

## 3.2 BREGMAN AWAY-STEP FRANK–WOLFE ALGORITHM

In the case where $P$ is a polytope, the classical FW algorithm might zigzag when approaching the optimal face and, in consequence, converge slowly. The *away-step FW algorithm* overcomes this drawback. *Away steps* allow the algorithm to move away from vertices in a convex combination of $x_t$, which can effectively short-circuit the zigzagging. The convergence properties of this algorithm were unknown for a long time, and it was only quite recently that Lacoste-Julien & Jaggi (2015) established the linear convergence of the away-step FW algorithm. Inspired by GuéLat & Marcotte (1986); Lacoste-Julien & Jaggi (2015); Wolfe (1970), we propose a variant of the away-step FW algorithm utilizing Bregman distances as given in Algorithm 3. The main difference between Algorithm 3 and the existing away-step FW algorithm lies in the update in line 8.

For the following discussion, we introduce some notions. In the same way as Braun et al. (2025), we define an active set $\mathcal{S} \subset \text{Vert}\, P$, where $\text{Vert}\, P$ denotes the set of vertices of $P$, and the away vertex as $v_t^A \in \arg\max_{v \in \mathcal{S}}\langle \nabla f(x_t), v \rangle$, where $x_t$ is a (strict) convex combination of elements in $\mathcal{S}$, *i.e.*, $x_t = \sum_{v \in \mathcal{S}} \lambda_v v$ with $\lambda_v > 0$ for all $v \in \mathcal{S}$ and $\sum_{v \in \mathcal{S}} \lambda_v = 1$. If $\nu$ is unknown, we can add line search procedures to search $\nu$ until Equation 2.1 holds. If $\nu$ and $L$ are unknown, we can use $L_t, \nu_t, \gamma_t \leftarrow$ step_size($f, \phi, x_t, v_t, L_{t-1}, \gamma_{t,\max}$) with $\gamma_t \leftarrow \min\left\{ \left(\frac{\langle \nabla f(x_t), d_t \rangle}{M(1+\kappa)D_\phi(v_t, x_t)}\right)^{1/\kappa}, \gamma_{t,\max} \right\}$ instead of line 5 in Algorithm 2. When $\phi = \frac{1}{2}\|\cdot\|^2$ and $\nu \equiv 1$, Algorithm 3 corresponds to the Euclidean away-step FW algorithm.

## 4 LINEAR CONVERGENCE FOR CONVEX OPTIMIZATION

In this section, we assume that $f$ is convex.

**Assumption 4.1.** The objective function $f$ is convex.

Let $D := \sqrt{\sup_{x,y \in P} D_\phi(x,y)}$ be the diameter of $P$. We will now establish faster convergence rates than $O(1/t)$ up to linear convergence depending on $\nu$ in Equation B.2 and $q$ in Definition 2.4. First, we establish the faster convergence of Algorithm 1 with the Bregman short step-size, *i.e.*, $\gamma_t = \min\left\{ \left(\frac{\langle \nabla f(x_t), x_t - v_t \rangle}{L(1+\nu)D_\phi(v_t, x_t)}\right)^{1/\nu}, 1 \right\}$, or Algorithm 2 in the case where the optimal solution lies in the relative interior. Its proof can be found in Appendix D.2.

**Theorem 4.2** (Linear convergence of FW algorithm with short step-size or adaptive step-size). Suppose that Assumptions 3.1 and 4.1 hold. Let $\nu > 0$ and let $f$ satisfy the HEB condition with $q \geq 1+\nu$ and $\mu > 0$ and $D$ be the diameter of $P$. Assume that there exists a minimizer $x^* \in \text{int}\, P$, *i.e.*, there exists an $r > 0$ with $B(x^*, r) \subset P$. Consider the iterates of Algorithm 1 with short step-size

---

**Algorithm 3:** Away-step Frank–Wolfe algorithm with the Bregman distance

---

**Input:** $x_0 \in \arg\min_{v \in P}\langle \nabla f(x), v\rangle$ for $x \in P$, $\beta < 1$

1   $\mathcal{S}_0 \leftarrow \{x_0\}$, $\lambda_{x_0,0} \leftarrow 1$

2   **for** $t = 0, \dots$ **do**

3     $v_t^{\mathrm{FW}} \leftarrow \arg\min_{v\in P}\langle\nabla f(x_t),v\rangle$, $v_t^{\mathrm{A}} \leftarrow \arg\max_{v\in\mathcal{S}_t}\langle\nabla f(x_t),v\rangle$

4     **if** $\langle\nabla f(x_t), x_t - v_t^{\mathrm{FW}}\rangle \geq \langle\nabla f(x_t), v_t^{\mathrm{A}} - x_t\rangle$ **then**

5       $v_t \leftarrow v_t^{\mathrm{FW}}$, $d_t \leftarrow x_t - v_t^{\mathrm{FW}}$, $\gamma_{t,\max} \leftarrow 1$

6     **else**

7       $v_t \leftarrow v_t^{\mathrm{A}}$, $d_t \leftarrow v_t^{\mathrm{A}} - x_t$, $\gamma_{t,\max} \leftarrow \lambda_{v_t^{\mathrm{A}},t}/(1-\lambda_{v_t^{\mathrm{A}},t})$

8     $\gamma_t \leftarrow \min\left\{\left(\frac{\langle\nabla f(x_t),d_t\rangle}{L(1+\nu)D_\phi(v_t,x_t)}\right)^{1/\nu}, \gamma_{t,\max}\right\}$, $x_{t+1} \leftarrow x_t - \gamma_t d_t$

9     **if** $\langle\nabla f(x_t), x_t - v_t^{\mathrm{FW}}\rangle \geq \langle\nabla f(x_t), v_t^{\mathrm{A}} - x_t\rangle$ **then**

10      $\lambda_{v,t+1} \leftarrow (1-\gamma_t)\lambda_{v,t}$ for all $v_t \in \mathcal{S}_t \setminus \{v_t^{\mathrm{FW}}\}$

11      $\lambda_{v_t^{\mathrm{FW}},t+1} \leftarrow \begin{cases}\gamma_t & \text{if } v_t^{\mathrm{FW}} \notin \mathcal{S}_t \\ (1-\gamma_t)\lambda_{v_t^{\mathrm{FW}},t}+\gamma_t & \text{if } v_t^{\mathrm{FW}} \in \mathcal{S}_t\end{cases}$, $\mathcal{S}_{t+1} \leftarrow \begin{cases}\mathcal{S}_t \cup \{v_t^{\mathrm{FW}}\} & \text{if } \gamma_t < 1 \\ \{v_t^{\mathrm{FW}}\} & \text{if } \gamma_t = 1\end{cases}$

12     **else**

13      $\lambda_{v,t+1} \leftarrow (1+\gamma_t)\lambda_{v,t}$ for all $v \in \mathcal{S}_t \setminus \{v_t^{\mathrm{A}}\}$, $\lambda_{v_t^{\mathrm{A}},t+1} \leftarrow (1+\gamma_t)\lambda_{v_t^{\mathrm{A}},t} - \gamma_t$

14      $\mathcal{S}_{t+1} \leftarrow \begin{cases}\mathcal{S}_t \setminus \{v_t^{\mathrm{A}}\} & \text{if } \lambda_{v_t^{\mathrm{A}},t+1} = 0 \\ \mathcal{S}_t & \text{if } \lambda_{v_t^{\mathrm{A}},t+1} > 0\end{cases}$

---

$\gamma_t = \min\left\{\left(\frac{\langle\nabla f(x_t),x_t-v_t\rangle}{L(1+\nu)D_\phi(v_t,x_t)}\right)^{1/\nu}, 1\right\}$ or the adaptive Bregman step-size strategy (Algorithm 2). Then, it holds that, for all $t \geq 1$,

$$f(x_t) - f^* \leq \begin{cases} \max\left\{\frac{1}{1+\nu}, 1 - \frac{\nu}{1+\nu}\frac{r^{1+1/\nu}}{c^{1+1/\nu}D^{2/\nu}}\right\}^{t-1} LD^2 & \text{if } q = 1+\nu, \\[2mm] \frac{LD^2}{(1+\nu)^{t-1}} & \text{if } 1 \leq t \leq t_0, q > 1+\nu, \\[2mm] \frac{(L(1+\nu)(c/r)^{1+\nu}D^2)^{q/(q-1-\nu)}}{\left(1+\frac{1-\nu}{2(1+\nu)}(t-t_0)\right)^{\nu q/(q-1-\nu)}} = \mathcal{O}(1/t^{\nu q/(q-1-\nu)}) & \text{if } t \geq t_0, q > 1+\nu, \end{cases}$$

where $c = (q/\mu)^{1/q}$ and $t_0 := \max\{\lfloor\log_{1/(1+\nu)}((L(1+\nu(c/r)^{1+\nu}D^2)^{q/(q-1-\nu)}/LD^2)\rfloor + 2, 1\}$.

Next, we establish the linear convergence of Algorithm 3. Recall that the pyramidal width is the minimal $\delta > 0$ satisfying Lemma C.9 (see (Braun et al., 2025, Lemma 2.26) or (Lacoste-Julien & Jaggi, 2015, Theorem 3) for an in-depth discussion). Under Assumption 4.3, it holds that $\delta \leq D$, and this will be important in establishing convergence rates of Algorithm 3. We make the following assumption:

**Assumption 4.3.** The kernel generating distance $\phi$ is $\sigma$-strongly convex.

We have the linear convergence of Algorithm 3. Its proof can be found in Appendix D.3.

**Theorem 4.4** (Linear convergence of the away-step FW algorithm). Suppose that Assumptions 3.1, 4.1, and 4.3 hold. Let $\nu > 0$ and let $P$ be a polytope and $f$ satisfy the HEB condition with $q > 1+\nu$ or $(\nu,q) = (1,2)$. The convergence rate of Algorithm 3 is linear: for all $t \geq 1$

$$f(x_t) - f^* \leq \begin{cases} \left(1 - \frac{\mu}{32L}\frac{\delta^2}{D^2}\right)^{\lceil(t-1)/2\rceil} LD^2 & \text{if } (\nu,q) = (1,2), \\[2mm] \frac{1}{(1+\nu)^{\lceil(t-1)/2\rceil}}LD^2 & \text{if } 1 \leq t \leq t_0, q > 1+\nu, \\[2mm] \frac{(L(1+\nu)D^2c/(\delta/2)^{1+\nu})^{q/(q-1-\nu)}}{\left(1+\frac{1-\nu}{2(1+\nu)}\lceil(t-t_0)/2\rceil\right)^{\nu q/(q-1-\nu)}} = \mathcal{O}(1/t^{\nu q/(q-1-\nu)}) & \text{if } t \geq t_0, q > 1+\nu, \end{cases}$$

where $c := (q/\mu)^{1/q} D$ and $\delta$ are the diameter and the pyramidal width of the polytope $P$, respectively, and $t_0 := \max\{\lfloor\log_{1/(1+\nu)}(L(1+\nu)D^2/((\mu/q)^{1/q}\delta/2)^{1+\nu})^{q/(q-1-\nu)}/LD^2\rfloor + 2, 1\}$.

When $\phi = \frac{1}{2}\|\cdot\|^2$, i.e., $\nu = 1$, Theorems 4.2 and 4.4 correspond to existing results (Braun et al., 2025; Kerdreux et al., 2022).

## 5 LOCAL LINEAR CONVERGENCE FOR NONCONVEX OPTIMIZATION

In this section, we consider a nonconvex objective function. We establish global sublinear, local sublinear, and local linear convergence (see Theorem E.1 and Theorem E.3 for global and local sublinear convergence, respectively). We show that the FW algorithm converges to a minimizer $x^*$ when an initial point is close enough to $x^*$. We need weak convexity and local quadratic growth for local convergence.

**Assumption 5.1.** $f$ is $\rho$-weakly convex and has local $\mu$-quadratic growth with $\zeta > 0$.

Assumption 5.1 is not too restrictive because the (local) strong convexity or the (local) Polyak–Łojasiewicz (PL) inequality implies the (local) quadratic growth condition for weakly convex functions (Liao et al., 2024, Theorem 3.1). For more examples satisfying Assumption 5.1 and $\rho \leq \mu$, see Examples E.4 and E.5.

**Remark 5.2.** Linear convergence rates for nonconvex optimization problems are, to the best of our knowledge, the first such result for the Frank–Wolfe algorithm. Their proofs are technically challenging. Proposition C.8 is new and key for establishing a linear rate. In the nonconvex case, one cannot derive primal-gap inequalities such as Lemma D.1. We overcome this by restricting our function class to weakly convex functions and proving Lemma C.2. There remains an obstacle that could be resolved by the assumption of the quadratic growth condition. Assumption 5.1 enables us to derive Lemma C.3 and Proposition C.8 (as well as Remark C.4), which are crucial for establishing linear convergence.

We have local linear convergence, and its proof can be found in Appendix E.3.

**Theorem 5.3** (Local linear convergence of FW algorithm with short step-size or adaptive step-size). Suppose that Assumptions 3.1 and 5.1 hold. Let $\nu \in (0, 1]$ and let $D$ be the diameter. Assume that there exists a minimizer $x^* \in \operatorname{int} P$, *i.e.*, there exists an $r > 0$ with $B(x^*, r) \subset P$. Consider the iterates of Algorithm 1 with $\gamma_t = \min\left\{ \left( \frac{\langle \nabla f(x_t), x_t - v_t \rangle}{L(1+\nu)D_\phi(v_t, x_t)} \right)^{1/\nu}, 1 \right\}$ or the adaptive Bregman step-size strategy (Algorithm 2). Then, if $\rho/\mu < 1$, it holds that, for all $t \geq 1$,

$$f(x_t) - f^* \leq \begin{cases} \max\left\{ \frac{1}{2}\left(1 + \frac{\rho}{\mu}\right), 1 - \frac{r^2}{2Mc^2D^2} \right\}^{t-1} LD^2 & \text{if } \nu = 1, \\ \left( \frac{1}{1+\nu}\left(1 + \frac{\nu\rho}{\mu}\right) \right)^{t-1} LD^2 & \text{if } 1 \leq t \leq t_0, \nu \in (0,1), \\ \frac{(L(1+\nu)(1-\rho/\mu)^\nu c^{1+\nu} D^2/r^{1+\nu})^{2/(1-\nu)}}{\left(1 + \frac{1-\nu}{2(1+\nu)}\left(1 - \frac{\rho}{\mu}\right)(t - t_0)\right)^{2\nu/(1-\nu)}} = \mathcal{O}(1/t^{2\nu/(1-\nu)}) & \text{if } t \geq t_0, \nu \in (0,1), \end{cases}$$

where $c = \sqrt{2\mu}/(\mu - \rho)$, $M := (L(1+\nu))^{1/\nu}$, and $t_0 := \max\{\lfloor \log_{(1+\nu\rho/\mu)/(1+\nu)}((L(1+\nu)(1-\rho/\mu)^\nu c^{1+\nu} D^2/r^{1+\nu})^{2/(1-\nu)}/LD^2) \rfloor + 2, 1\}$.

Finally, we establish the local linear convergence of Algorithm 3. Its proof can be found in Appendix E.4. We assume $\rho \leq L$, which is not restrictive (see Appendix E.2 and Example E.4).

**Theorem 5.4** (Local linear convergence by the away-step FW algorithm). Suppose that Assumptions 3.1, 4.3, and 5.1 hold. Let $\nu \in (0, 1]$ and let $P \subset \mathbb{R}^n$ be a polytope. The convergence rate of Algorithm 3 with $f$ is linear: if $\rho < \mu \leq L$, for all $t \geq 1$

$$f(x_t) - f^* \leq \begin{cases} 2\left(1 - \frac{\omega}{4L}\frac{\delta^2}{D^2}\right)^{\lceil (t-1)/2 \rceil} LD^2 & \text{if } \nu = 1, \\ 2\left( \frac{1}{1+\nu}\left(1 + \frac{\nu\rho}{\mu}\right) \right)^{\lceil (t-1)/2 \rceil} LD^2 & \text{if } 1 \leq t \leq t_0, \nu \in (0,1), \\ \frac{(L(1+\nu)(1-\rho/\mu)^\nu D^2/(\sqrt{\omega}\delta)^{1+\nu})^{2/(1-\nu)}}{\left(1 + \frac{1-\nu}{2(1+\nu)}\left(1 - \frac{\rho}{\mu}\right)\lceil(t-t_0)/2\rceil\right)^{2\nu/(1-\nu)}} = \mathcal{O}(1/t^{2\nu/(1-\nu)}) & \text{if } t \geq t_0, \nu \in (0,1), \end{cases}$$

where $D$ and $\delta$ are the diameter and the pyramidal width of $P$, respectively, and $\omega := (\mu - \rho)^2/8\mu$, and $t_0 := \max\{\lfloor \log_{(1+\nu\rho/\mu)/(1+\nu)}(L^2(1+\nu)^2 D^4(1-\rho/\mu)^{2\nu}/(\omega\delta^2)^{1+\nu})^{1/(1-\nu)}/2LD^2 \rfloor + 2, 1\}$.

Without loss of generality, $\sigma = 1$. If $\rho > L$, we can use $(\rho + L)D^2$, instead of $2LD^2$ as the initial bound of $f(x_1) - f^*$. When $\phi = \frac{1}{2}\|\cdot\|^2$, *i.e.*, $\nu = 1$, Theorems 5.3 and 5.4 show linear convergence rates for Euclidean cases. Even if we use the Euclidean distance, our convergence results are novel in that linear convergence is achieved for the first time for a certain type of nonconvex problem. Moreover, when $P$ is $(\alpha, p)$-uniformly convex set (see Definition E.7) and $\phi = \frac{1}{2}\|\cdot\|^2$, we also establish local linear convergence (Theorems E.9 and E.10) in Appendix E.5.

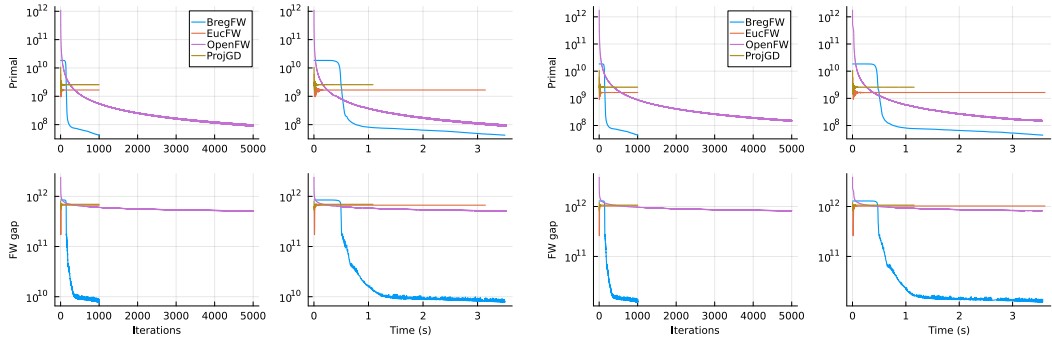

**Figure 1:** Log plot of primal and FW gaps on $\ell_p$ loss for gas sensor data with $b_{\max} = 130$.

**Figure 2:** Log plot of primal and FW gaps on $\ell_p$ loss for gas sensor data with $b_{\max} = 200$.

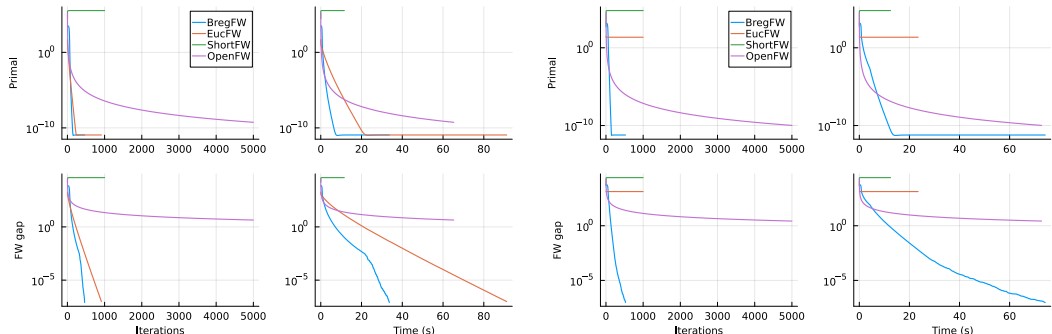

**Figure 3:** Log plot of primal and FW gaps on phase retrieval for $(m, n) = (1000, 10000)$.

**Figure 4:** Log plot of primal and FW gaps on phase retrieval for $(m, n) = (2000, 10000)$.

## 6 NUMERICAL EXPERIMENTS

In this section, we conduct numerical experiments to examine the performance of our algorithms. All numerical experiments were performed in Julia 1.11 using the `FrankWolfe.jl` package (Besançon et al., 2022)[8] on a MacBook Pro with an Apple M2 Max and 64GB LPDDR5 memory.

We compare `BregFW` (Algorithm 1) and `BregAFW` (Algorithm 3, whose results are shown in Figures 7) using the adaptive Bregman step-size strategy (Algorithm 2), with existing algorithms, `EucFW`, `EucAFW`, `ShortFW`, `ShortAFW`, `OpenFW`, `OpenAFW`, `MD`, and `ProjGD` (their notation is described in Appendix F). In this section, we consider $\ell_p$ loss problems and phase retrieval, and we report additional experiments on nonnegative linear inverse problems, low-rank minimization, and NMF in Appendix F. Note that we included `OpenAFW` only for comparison purposes; there is no established convergence theory for the away-step FW algorithm with the open loop. Indeed, there are no proper drop steps (operations corresponding to line 14 in Algorithm 3), and the favorable properties of the away-step FW algorithm are lost; see Braun et al. (2025). We use $\beta = 0.9$, $\eta = 0.9$, $\tau = 2$, $\gamma_{\max} = 1$, and the termination criterion $\max_{v \in P} \langle \nabla f(x_t), x_t - v \rangle \leq 10^{-7}$ throughout all experiments.

### 6.1 $\ell_p$ LOSS PROBLEM

We consider the $\ell_p$ loss problem (Kyng et al., 2015; Maddison et al., 2021) to find $x \in P$ such that $Ax \simeq b$, defined by a convex optimization problem $\min_{x \in P} \|Ax - b\|_p^p$, where $A \in \mathbb{R}^{m \times n}$, $b \in \mathbb{R}^m$, and $p > 1$. Let $f(x) = \|Ax - b\|_p^p$. The gradient $\nabla f$ is not Lipschitz continuous on $\mathbb{R}^n$ when $p \neq 2$. Furthermore, when $p < 2$, $f$ is not $\mathcal{C}^2$ but $\mathcal{C}^1$. Therefore, for $1 < p < 2$, $\nabla f$ is also not Lipschitz continuous over compact sets. Since $f$ is convex, the pair $(f, \phi)$ is 1-smad with $\phi := f$.

---

[8] https://github.com/ZIB-IOL/FrankWolfe.jl

We consider real-world gas sensor data.[9] The data size is $(m, n) = (13910, 128)$. We use the $\ell_2$ constraint $P = \{x \in \mathbb{R}^n \mid \|x\|^2 \le b_{\max}\}$. We compare `BregFW` with `EucFW`, `OpenFW`, and `ProjGD` because `ShortFW` does not converge. The maximum number of iterations is 1000 for all methods except `OpenFW`, for which it is 5000. The initial point $x_0$ is an all-ones vector. We set $p = 1.1$, *i.e.*, $f$ is $\mathcal{C}^1$ not $\mathcal{C}^2$. Figures 1 and 2 show the primal and FW gaps per iteration and gaps per second for $b_{\max} = 130$ and $b_{\max} = 200$, respectively. Because $\nabla f$ is not Lipschitz continuous, `ShortFW` and `EucFW` do not converge. The primal gap and the FW gap achieved by `BregFW` are the smallest among these methods because only `BregFW` has a theoretical guarantee for this problem.

### 6.2 PHASE RETRIEVAL

We are interested in phase retrieval, which involves finding a signal $x \in P \subset \mathbb{R}^n$ such that $|\langle a_i, x \rangle|^2 \simeq b_i$, $i = 1, \ldots, m$, where $a_i \in \mathbb{R}^n$ describes the model and $b \in \mathbb{R}^m$ is a vector of measurements. Phase retrieval arises in many fields of science and engineering, such as image processing (Candés et al., 2015), X-ray crystallography (Patterson, 1934; 1944), and optics (Shechtman et al., 2015). To achieve the goal of phase retrieval, we focus on the nonconvex optimization problem $\min_{x \in P} f(x) := \frac{1}{4} \sum_i (|\langle a_i, x \rangle|^2 - b_i)^2$. Let $\phi$ be defined by $\sigma$-strongly convex $\phi(x) = \frac{1}{4}\|x\|^4 + \frac{1}{2}\|x\|^2$ for $\sigma \le 1$. For any $L$ satisfying $L \ge \sum_i (3\|a_i\|^4 + \|a_i\|^2 |b_i|)$, the pair $(f, \phi)$ is $L$-smad on $\mathbb{R}^n$ (Bolte et al., 2018, Lemma 5.1). In addition, $\nabla^2 f(x) + \rho I$ is positive semidefinite for $\rho \ge \sum_i \|a_i\|^2 |b_i|$, *i.e.*, $f$ is $\rho$-weakly convex. Thus, $L \ge \rho/\sigma$ holds with $\sigma = 1$.

We use a $K$-sparse polytope as a constraint, *i.e.*, $P = \{x \in \mathbb{R}^n \mid \|x\|_1 \le K, \|x\|_\infty \le 1\}$. We compare `BregFW` with `EucFW`, `ShortFW`, and `OpenFW`. The maximum number of iterations is 1000 for all methods except `OpenFW`, for which it is 5000. We generated $a_i$, $i = 1, \ldots, m$ from an i.i.d. normal distribution and normalized them to have norm 1 (random seed 1234). We also generated $x^*$ from an i.i.d. uniform distribution in $[0, 1]$ and normalized $x^*$ to have sum 1. The initial point $x_0$ was generated by computing an extreme point of $P$ that minimizes the linear approximation of $f$. Figures 3 and 4 show the primal and FW gaps per iteration and gaps per second for $(m, n) = (1000, 10000)$ and $K = 2000$ and for $(m, n) = (2000, 10000)$ and $K = 2000$, respectively. The primal gap and the FW gap by the adaptive Bregman step-size strategy are the smallest among these step-size strategies. In both cases, `BregFW` stopped before the 1000th iteration.

## 7 CONCLUSION AND FUTURE WORK

**Summary:** We propose FW algorithms with an adaptive Bregman step-size strategy for smooth adaptable functions, which do not require Lipschitz continuous gradients. We have established convergence rates ranging from sublinear to linear, depending on various assumptions for convex and nonconvex optimization. We also propose a variant of the away-step FW algorithm using Bregman distances over polytopes and have established global convergence for convex optimization under the HEB condition and local linear convergence for nonconvex optimization under the local quadratic growth condition. Numerical experiments show that our algorithms outperform existing algorithms.

**Limitations and future work:** Our step-size strategy for $L$-smad functions requires the value of $\nu$, which must be estimated because it depends on the choice of the Bregman distance. Theoretically, linear convergence of the FW algorithm always holds for convex optimization if the exponent of the HEB condition $q$ equals the scaling exponent of the Bregman distance $1 + \nu$, *i.e.*, $q = 1 + \nu$. For $q > 1 + \nu$, if $t \le t_0$, linear convergence holds; otherwise, sublinear convergence. The condition $q = 1 + \nu$, derived from the growth of $f$ and $\phi$, is natural because $\phi$ has a similar structure to $f$. On the other hand, for nonconvex optimization, we assume the local quadratic growth condition, *i.e.*, the 2-HEB condition. Since this is derived from the weak convexity of $f$, relaxing the quadratic growth condition may not be possible. One direction is to consider an extension using the difference of convex functions (DC) optimization. Recently, Maskan et al. (2025) incorporated DC optimization into FW algorithms and established sublinear convergence. Our algorithms could potentially benefit from DC optimization techniques.

---

[9]The dataset is freely available at https://archive.ics.uci.edu/dataset/270/gas+sensor+array+drift+dataset+at+different+concentrations

## ACKNOWLEDGMENTS

This project has been partially supported by the Japan Society for the Promotion of Science (JSPS) through Invitational Fellowships for Research in Japan program (L24503), JSPS KAKENHI Grant Number JP23K19953 and JP25K21156; JSPS KAKENHI Grant Number JP23H03351 and JST ERATO Grant Number JPMJER1903. Research reported in this paper was also partially supported by the Deutsche Forschungsgemeinschaft (DFG) through the DFG Cluster of Excellence MATH+ (grant number EXC-2046/1, project ID 390685689).

## LLM USAGE

We used large language models, ChatGPT-5/5.1/5.2, solely to polish the grammar and wording of the author-written text (*e.g.*, to improve clarity and tone). The model did not generate novel ideas, analyses, code, or results. All outputs were reviewed and edited by the authors, who take full responsibility for the content.

## ETHICS STATEMENT

This study uses only publicly available datasets that contain no personally identifiable information. We did not collect any new human-subject data; therefore, IRB approval was not required. We also reviewed the datasets for potentially harmful content.

## REPRODUCIBILITY STATEMENT

We include all evaluation code, configuration files, and scripts in the supplementary material. We provide the exact preprocessing steps for all datasets used. We fix and disclose random seeds for all experiments. Results are reported as mean $\pm$ standard deviation over multiple runs.

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

## A    RELATED WORK

**Overview of Existing Bregman-Based Algorithm Studies:**    The Bregman gradient method was first introduced as the mirror descent by Nemirovskij & Yudin (1983). Accelerated mirror descent methods were proposed by Krichene et al. (2015); Allen-Zhu & Orecchia (2017). Some first-order methods based on the Bregman distance do not require the global Lipschitz continuity of $\nabla f$. Bolte et al. (2018) proposed the Bregman proximal gradient algorithm (BPG) for nonconvex optimization problems and established its global convergence under the smooth adaptable property (see Definition 2.3). Hanzely et al. (2021) introduced the accelerated BPGs. Takahashi et al. (2022) developed the BPG for the DC optimization and its accelerated version. The subproblem of BPG is often not solvable in closed form. Takahashi & Takeda (2025); Fujiki et al. (2025) used approximations to Bregman distances to solve subproblems in closed form. Takahashi et al. (2026) developed the majorization-minimization BPG and its accelerated version, and applied them to NMF. Rebegoldi et al. (2018) proposed an inexact version of the Bregman proximal gradient algorithm.

## B    RESULTS FROM SMOOTH ADAPTABLE PROPERTY

### B.1    PROPERTIES OF BREGMAN DISTANCES

The Bregman distance $D_\phi(x, y)$ measures the proximity between $x \in \operatorname{dom} \phi$ and $y \in \operatorname{int} \operatorname{dom} \phi$. Indeed, since $\phi$ is convex, it holds that $D_\phi(x, y) \geq 0$ for all $x \in \operatorname{dom} \phi$ and $y \in \operatorname{int} \operatorname{dom} \phi$. Moreover, when $\phi$ is strictly convex, $D_\phi(x, y) = 0$ holds if and only if $x = y$. However, the Bregman distance is not always symmetric and does not have to satisfy the triangle inequality. The Bregman distance is also called the Bregman divergence.

**Example B.1.** Well-known choices for $\phi$ and $D_\phi$ are listed below; for more examples, see, *e.g.*, (Bauschke et al., 2017, Example 1), (Bauschke & Borwein, 1997, Section 6), and (Dhillon & Tropp, 2008, Table 2.1).

(i) Let $\phi(x) = \frac{1}{2}\|x\|^2$ and $\operatorname{dom} \phi = \mathbb{R}^n$. Then, the Bregman distance corresponds to the squared Euclidean distance, *i.e.*, $D_\phi(x, y) = \frac{1}{2}\|x - y\|^2$.

(ii) The Boltzmann–Shannon entropy $\phi(x) = \sum_{i=1}^n x_i \log x_i$ with $0 \log 0 = 0$ and $\operatorname{dom} \phi = \mathbb{R}_+^n$. Then, $D_\phi(x, y) = \sum_{i=1}^n (x_i \log \frac{x_i}{y_i} - x_i + y_i)$ is called the Kullback–Leibler (KL) divergence (Kullback & Leibler, 1951).

(iii) The Burg entropy $\phi(x) = -\sum_{i=1}^n \log x_i$ and $\operatorname{dom} \phi = \mathbb{R}_{++}^n$. Then, $D_\phi(x, y) = \sum_{i=1}^n (\frac{x_i}{y_i} - \log \frac{x_i}{y_i} - 1)$ is called the Itakura–Saito divergence (Itakura & Saito, 1968).

(iv) Let $\phi(x) = \frac{1}{4}\|x\|^4 + \frac{1}{2}\|x\|^2$ and $\operatorname{dom} \phi = \mathbb{R}^n$. The Bregman distance $D_\phi$ is used in phase retrieval (Bolte et al., 2018), low-rank minimization (Dragomir et al., 2021), NMF (Mukkamala & Ochs, 2019), and blind deconvolution (Takahashi et al., 2023).

We recall the triangle scaling property for Bregman distances from (Hanzely et al., 2021, Section 2), where several properties of the triangle scaling property are also shown.

**Definition B.2** (Triangle Scaling Property ([Hanzely et al., 2021](#), Definition 2)). Given a kernel generating distance $\phi \in \mathcal{G}(C)$, the Bregman distance $D_\phi$ has the *triangle scaling property* if there exists a constant $\nu > 0$ such that, for all $x, y, z \in \text{int dom } \phi$ and all $\gamma \in [0, 1]$, it holds that

$$D_\phi((1 - \gamma)x + \gamma y, (1 - \gamma)x + \gamma z) \leq \gamma^\nu D_\phi(y, z). \tag{B.1}$$

Now, substituting $z \leftarrow x$ on the left-hand side of Equation [B.1](#), we obtain

$$\begin{aligned} D_\phi((1 - \gamma)x + \gamma y, x) &= \phi((1 - \gamma)x + \gamma y) - \phi(x) - \gamma\langle\nabla\phi(x), y - x\rangle \\ &\leq (1 - \gamma)\phi(x) + \gamma\phi(y) - \phi(x) - \gamma\langle\nabla\phi(x), y - x\rangle = \gamma D_\phi(y, x), \end{aligned}$$

where the inequality holds because of the convexity of $\phi$. Therefore, there exists $\nu > 0$ such that $D_\phi((1 - \gamma)x + \gamma y, x) \leq \gamma^\nu D_\phi(y, x)$ holds for all $x, y \in \text{int dom } \phi$ and all $\gamma \in [0, 1]$.

We will now show that a stronger version can be obtained if $\phi$ is strictly convex, so that we can rephrase $\nu$ with $1 + \nu$ where $\nu > 0$, *i.e.*, the right-hand side is superlinear, which will be crucial for the convergence analysis later.

**Lemma B.3.** Given a kernel generating distance $\phi \in \mathcal{G}(C)$, if $\phi$ is strictly convex, then there exists $\nu > 0$ such that, for all $x, y \in \text{int dom } \phi$ and all $\gamma \in [0, 1]$, it holds that

$$D_\phi((1 - \gamma)x + \gamma y, x) \leq \gamma^{1+\nu} D_\phi(y, x). \tag{B.2}$$

*Proof.* Equation [B.2](#) holds if $y = x$ or $\gamma \in \{0, 1\}$. In what follows, we assume $y \neq x$ and $0 < \gamma < 1$. Let $g(\nu) := \gamma^{1+\nu} D_\phi(y, x) - D_\phi((1 - \gamma)x + \gamma y, x)$ for all $x, y \in \text{int dom } \phi$, $y \neq x$ and all $\gamma \in (0, 1)$. Using $D_\phi(y, x) > 0$ due to the strict convexity of $\phi$, we have, for any $\nu \geq 0$,

$$g'(\nu) = \gamma^{1+\nu} D_\phi(y, x) \log \gamma < 0,$$

which implies $g$ monotonically decreases. In addition, it holds that

$$\begin{aligned} g(0) &= \gamma D_\phi(y, x) - D_\phi((1 - \gamma)x + \gamma y, x) \\ &= \gamma(\phi(y) - \phi(x) - \langle\nabla\phi(x), y - x\rangle) - \phi((1 - \gamma)x + \gamma y) + \phi(x) + \gamma\langle\nabla\phi(x), y - x\rangle \\ &= (1 - \gamma)\phi(x) + \gamma\phi(y) - \phi((1 - \gamma)x + \gamma y) > 0, \end{aligned}$$

where the last inequality holds because $\phi$ is strictly convex, *i.e.*, $\phi((1 - \gamma)x + \gamma y) < (1 - \gamma)\phi(x) + \gamma\phi(y)$. Therefore, there exists $\nu > 0$ such that $g(\nu) \geq 0$ by the intermediate value theorem. $\square$

Because $\phi$ is quadratic when $D_\phi$ is symmetric ([Bauschke & Borwein, 2001](#), Lemma 3.16), it always holds that $D_\phi((1 - \gamma)x + \gamma y, x) = \gamma^{1+\nu} D_\phi(y, x)$ with $\nu = 1$.

## B.2 PROPERTIES OF SMOOTH ADAPTABLE PROPERTY

The convexity of $L\phi - f$ and $L\phi + f$ plays a central role in developing and analyzing algorithms, and the smooth adaptable property implies the extended descent lemma.

**Lemma B.4** (Extended Descent Lemma ([Bolte et al., 2018](#), Lemma 2.1)). The pair of functions $(f, \phi)$ is $L$-smad on $C$ if and only if for all $x, y \in \text{int dom } \phi$,

$$|f(x) - f(y) - \langle\nabla f(y), x - y\rangle| \leq L D_\phi(x, y).$$

From this, it can be seen that the $L$-smad property for $(f, \phi)$ provides upper and lower approximations for $f$ majorized by $\phi$ with $L > 0$. In addition, if $\phi(x) = \frac{1}{2}\|x\|^2$ on $\text{int dom } \phi = \mathbb{R}^n$, Lemma [B.4](#) corresponds to the classical descent lemma. While the $L$-smad property might seem unfamiliar at first, it is a natural generalization of $L$-smoothness, and examples of functions $f$ and $\phi$ satisfying the $L$-smad property are given, *e.g.*, in ([Bauschke et al., 2017](#), Lemmas 7 and 8), ([Bolte et al., 2018](#), Lemma 5.1), ([Dragomir et al., 2021](#), Propositions 2.1 and 2.3), ([Mukkamala & Ochs, 2019](#), Proposition 2.1), ([Takahashi & Takeda, 2025](#), Proposition 24), ([Takahashi et al., 2023](#), Theorem 1), and ([Takahashi et al., 2026](#), Theorem 15).

**Example B.5.** Other examples can be found in Sections [6.1](#), [6.2](#), Appendices [F.1](#), [F.4](#), and [F.5](#).

(i) Blind deconvolution: Let $f : \mathbb{R}^{d_1} \times \mathbb{R}^{d_2} \to \mathbb{R}$ be $f(h, x) = \frac{1}{4}\|Bh \odot Ax - y\|^2$, where $\odot$ denotes the Hadamard product (also known as the element-wise product). The function $f$ can be decomposed into a DC function $f(h, x) = f_1(h, x) - f_2(h, x)$ with $f_1(h, x) = \frac{1}{4}\|Bh\|_4^4 + \frac{1}{4}\|Ax\|_4^4 + \frac{1}{2}\left(\|Bh \odot Ax\|^2 + \|y \odot Bh\|^2 + \|Ax\|^2 + \|y\|^2\right)$ and $f_2(h, x) = \frac{1}{4}\|Bh\|_4^4 + \frac{1}{4}\|Ax\|_4^4 + \frac{1}{2}\|y \odot Bh + Ax\|^2$, where $B \in \mathbb{R}^{m \times d_1}$, $A \in \mathbb{R}^{m \times d_2}$, and $y \in \mathbb{R}^m$. Let $\phi(h, x) = \frac{1}{4}(\|h\|^2 + \|x\|^2)^2 + \frac{1}{2}(\|h\|^2 + \|x\|^2)$ For any $L$ satisfying

$$L \geq \sum_{j=1}^{m} \left(3\|b_j\|^4 + 3\|a_j\|^4 + \|b_j\|\|a_j\| + |y_j|\|b_j\|^2 + \|a_j\|^2\right),$$

where $b_j$ and $a_j$ are the $j$th row vectors of $B$ and $A$, respectively, $(f_1, \phi)$ is $L$-smad (Takahashi et al., 2023, Theorem 1). Note that Bregman proximal DC algorithms require the smooth adaptable property of $(f_1, \phi)$, instead of $(f, \phi)$.

(ii) KL-NMF: Let $f : \mathbb{R}_+^{m \times r} \times \mathbb{R}_+^{r \times n} \to \mathbb{R}$ be $f(W, H) = \sum_{i=1}^{m} \sum_{j=1}^{n} (X_{ij} \log \frac{X_{ij}}{(WH)_{ij}} - X_{ij} + (WH)_{ij})$, where $X \in \mathbb{R}_+^{m \times n}$. $f$ is bounded by $\hat{f}(W, H) = \sum_{i,j} (X_{ij} \log X_{ij} - X_{ij} \sum_{l=1}^{r} \alpha_{ilj} \log \frac{W_{il}H_{lj}}{\alpha_{ilj}} - X_{ij} + (WH)_{ij})$ with $\alpha_{ilj} \in [0, 1]$, $i = 1, \ldots, m$, $l = 1, \ldots, r$, $j = 1, \ldots, n$. Let $\phi(W, H) = \sum_{i,l}(-\log W_{il} + \frac{1}{2}W_{il}^2) + \sum_{l,j}(-\log H_{lj} + \frac{1}{2}H_{lj}^2)$. Then, for any $L_k > 0$ satisfying

$$L \geq \max \left\{ \max_{i,l} \left\{ \sum_{j=1}^{n} \alpha_{ilj} X_{ij} \right\}, \max_{l,j} \left\{ \sum_{i=1}^{m} \alpha_{ilj} X_{ij} \right\}, m, n \right\},$$

the pair $(\hat{f}, \phi)$ is $L$-smad on $\mathbb{R}_{++}^{m \times r} \times \mathbb{R}_{++}^{r \times n}$ (Takahashi et al., 2026, Theorem 4.1). Note that the majorization-minimization BPGs (Takahashi et al., 2026) require the smooth adaptable property of $(\hat{f}, \phi)$, instead of $(f, \phi)$.

**Remark B.6.** In general, it is difficult to choose the best $\phi$ such that $(f, \phi)$ is $L$-smad. On the other hand, it is known how to choose a suitable $\phi$ empirically. An easy way to find $\phi$ is to use elementary functions similar to $f$. See also the above examples, Section 6, and Appendix F.

The extended descent lemma immediately implies primal progress of FW algorithms under the $L$-smad property as a straightforward generalization of the $L$-smooth case.

**Lemma B.7** (Primal progress from the smooth adaptable property). Let the pair of functions $(f, \phi)$ be $L$-smad with a strictly convex function $\phi \in \mathcal{G}(C)$ and let $x_+ = (1 - \gamma)x + \gamma v$ with $x, v \in \text{int dom } \phi$ and $\gamma \in [0, 1]$. Then it holds:

$$f(x) - f(x_+) \geq \gamma \langle \nabla f(x), x - v \rangle - L\gamma^{1+\nu} D_\phi(v, x).$$

*Proof.* Using Lemma B.4 and substituting $x_+$ for $x$ and $x$ for $y$, we have
$$f(x) - f(x_+) \geq \gamma \langle \nabla f(x), x - v \rangle - L D_\phi(x_+, x).$$
It holds that $D_\phi(x_+, x) \leq \gamma^{1+\nu} D_\phi(v, x)$ for some $\nu > 0$ by Lemma B.3. Therefore, this provides the desired inequality. □

We also have a progress lemma from the Bregman short step-size.

**Lemma B.8** (Progress lemma from the Bregman short step-size). Suppose that Assumption 3.1 holds. Define $x_{t+1} = (1 - \gamma_t)x_t + \gamma_t v_t$. Then if $x_{t+1} \in \text{dom } f$,

$$f(x_t) - f(x_{t+1}) \geq \frac{\nu}{1+\nu}\gamma_t \langle \nabla f(x_t), x_t - v_t \rangle \quad \text{for} \quad 0 \leq \gamma_t \leq \left(\frac{\langle \nabla f(x_t), x_t - v_t \rangle}{L(1+\nu)D_\phi(v_t, x_t)}\right)^{\frac{1}{\nu}}.$$

*Proof.* Substituting $x_{t+1} = (1 - \gamma_t)x_t + \gamma_t v_t$ with $x_t, v_t \in P$ for $x_+$ in Lemma B.7, we have

$$f(x_t) - f(x_{t+1}) \geq \gamma_t \langle \nabla f(x_t), x_t - v_t \rangle - L\gamma_t^{1+\nu} D_\phi(v_t, x_t)$$
$$\geq \gamma_t \langle \nabla f(x_t), x_t - v_t \rangle - \frac{\gamma_t}{1+\nu} \langle \nabla f(x_t), x_t - v_t \rangle$$
$$= \frac{\nu}{1+\nu}\gamma_t \langle \nabla f(x_t), x_t - v_t \rangle,$$

where the second inequality holds because of an upper bound of $\gamma_t$. □

### B.3 PROOF OF THEOREM 3.3

*Proof of Theorem 3.3.* By following the same argument as (Pedregosa et al., 2020, Theorem 1), we have a bound on estimating $L$ as $(1 - \log\eta/\log\tau)(t + 1) + \max\{\log(\tau L/L_{-1}), 0\}/\log\tau$. Let $n_i$ be the number of evaluations needed to estimate $\nu$. We have $\nu = \beta^{n_i-1}$, which provides $n_i = 1 + \log\nu/\log\beta$. Therefore, we obtain $n_t \leq \max\{(1 - \log\eta/\log\tau)(t + 1) + \max\{\log(\tau L/L_{-1}), 0\}/\log\tau, (1 + \log\nu/\log\beta)(t + 1)\}$ $\qquad\square$

## C PROPERTIES FROM SHARPNESS

### C.1 RESULTS FROM WEAK CONVEXITY

A function $f$ is $\rho$-weakly convex and differentiable if and only if, for all $x, y \in \text{dom } f$,

$$f(y) - f(x) \geq \langle \nabla f(x), y - x \rangle - \frac{\rho}{2}\|y - x\|^2.$$

Any $\mathcal{C}^2$ function defined on a compact set is weakly convex (Vial, 1983, Proposition 4.11) and Wu (2007). Since the proof of this fact is omitted therein, we provide the argument for the sake of completeness below.

**Proposition C.1.** Let $P \subset \mathbb{R}^n$ be a compact set and $f$ be a proper and $\mathcal{C}^2$ function on $P$. Then, $f$ is weakly convex on $P$.

*Proof.* Let $g = \lambda_{\min}(\nabla^2 f(\cdot))$, where $\lambda_{\min}(\cdot)$ is the smallest eigenvalue. The function $g$ is continuous because $\lambda_{\min}(\cdot)$ and $\nabla^2 f(\cdot)$ are continuous. Therefore, there exists the minimum value of $g$ on $P$ due to the continuity of $g$ and the compactness of $P$. For $\rho = |\min_{x \in P} g(x)|$, we have $\nabla^2 f(x) + \rho I \succeq O$ for any $x \in P$, which implies $f + \frac{\rho}{2}\|\cdot\|^2$ is convex on $P$, *i.e.*, $f$ is $\rho$-weakly convex on $P$. $\qquad\square$

Minimizing a nonconvex $\mathcal{C}^2$-function over a compact set $P$ is equivalent to minimizing a weakly convex function over $P$. However, it is important to note that when $\rho$ becomes large, the assumptions of algorithms may not be satisfied (see also Theorems E.3 and 5.4).

We have the key lemma for convergence analysis from weak convexity.

**Lemma C.2** (Primal gap, dual gap, and Frank–Wolfe gap for weakly convex $f$). Suppose that Assumptions 3.1(ii), (iv), and that $f$ is $\rho$-weakly convex. Then for all $x \in P$, it holds:

$$f(x) - f^* \leq \langle \nabla f(x), x - x^* \rangle + \frac{\rho}{2}\|x - x^*\|^2 \leq \max_{v \in P}\langle \nabla f(x), x - v \rangle + \frac{\rho}{2}\|x - x^*\|^2. \quad \text{(C.1)}$$

*Proof.* The first inequality follows from the weak convexity of $f$ and the second follows from the maximality of $\langle \nabla f(x), x - v \rangle$. $\qquad\square$

### C.2 PRIMAL GAP BOUND

We show in the following lemma that the HEB condition immediately provides a bound on the primal optimality gap, which will be useful for establishing convergence rates of the proposed algorithms as well as the Łojasiewicz inequality (Bolte et al., 2007; Łojasiewicz, 1963; 1965), provided $f$ is convex.

**Lemma C.3** (Primal gap bound from Hölder error bound). Let $f$ be a convex function and satisfy the HEB condition with $q \geq 1$ and $\mu > 0$. Let $x^*$ be the unique minimizer of $f$ over $P$. Then, the following argument holds in general, for all $x \in P$:

$$f(x) - f^* \leq \left(\frac{q}{\mu}\right)^{\frac{1}{q-1}} \left(\frac{\langle \nabla f(x), x - x^* \rangle}{\|x - x^*\|}\right)^{\frac{q}{q-1}}, \quad \text{(C.2)}$$

or equivalently,

$$\left(\frac{\mu}{q}\right)^{1/q} (f(x) - f^*)^{1-1/q} \leq \frac{\langle \nabla f(x), x - x^* \rangle}{\|x - x^*\|}.$$

*Proof.* By first applying convexity and then the HEB condition for any $x^* \in \mathcal{X}^*$ with $f^* = f(x^*)$ it holds:

$$
\begin{aligned}
f(x) - f^* &\leq \langle \nabla f(x), x - x^* \rangle \\
&= \frac{\langle \nabla f(x), x - x^* \rangle}{\|x - x^*\|} \|x - x^*\| \\
&\leq \frac{\langle \nabla f(x), x - x^* \rangle}{\|x - x^*\|} \left( \frac{q}{\mu} (f(x) - f^*) \right)^{1/q},
\end{aligned}
$$

which implies

$$
\left( \frac{\mu}{q} \right)^{1/q} (f(x) - f^*)^{1 - 1/q} \leq \frac{\langle \nabla f(x), x - x^* \rangle}{\|x - x^*\|},
$$

or equivalently

$$
f(x) - f^* \leq \left( \frac{q}{\mu} \right)^{\frac{1}{q-1}} \left( \frac{\langle \nabla f(x), x - x^* \rangle}{\|x - x^*\|} \right)^{\frac{q}{q-1}}.
$$

$\square$

**Remark C.4.** With the remark from above, we can immediately relate the HEB condition to the Łojasiewicz inequality (Bolte et al., 2007; Łojasiewicz, 1963; 1965); the Łojasiewicz inequality is used to establish convergence analysis (Bolte et al., 2007). To this end, using Equation C.2, we estimate $\frac{\langle \nabla f(x), x - x^* \rangle}{\|x - x^*\|} \leq \frac{\|\nabla f(x)\| \|x - x^*\|}{\|x - x^*\|} = \|\nabla f(x)\|$ to then obtain the weaker condition:

$$
f(x) - f^* \leq \left( \frac{q}{\mu} \right)^{\frac{1}{q-1}} \|\nabla f(x)\|^{\frac{q}{q-1}} = c^{\frac{1}{\theta}} \|\nabla f(x)\|^{\frac{1}{\theta}}, \tag{C.3}
$$

where $c = \left( \frac{q}{\mu} \right)^{\frac{1}{q}}$ and $\theta = \frac{q-1}{q}$. Inequality Equation C.3 is called the $c$-Łojasiewicz inequality with $\theta \in [0, 1)$. If $\mathcal{X}^* \subseteq \mathrm{ri}\, P$, then the two conditions are equivalent. However, if the optimal solution(s) are on the boundary of $P$ as is not infrequently the case, then the two conditions *are not* equivalent as $\|\nabla f(x)\|$ might not vanish for $x \in \mathcal{X}^*$, whereas $\langle \nabla f(x), x - x^* \rangle$ does, *i.e.*, the HEB condition is tighter than the one induced by the Łojasiewicz inequality.

The next lemma shows that we can also obtain primal gap bounds in the weakly convex case together with (local) quadratic growth.

**Lemma C.5** (Primal gap bound from the quadratic growth)**.** Let $f$ be a $\rho$-weakly convex function that satisfies the local $\mu$-quadratic growth condition such that $\rho < \mu$. Let $x^*$ be the unique minimizer of $f$ over $P$ and let $\zeta > 0$. Then, the following holds: for all $x \in [f \leq f^* + \zeta] \cap P$,

$$
f(x) - f^* \leq \frac{2\mu}{(\mu - \rho)^2} \left( \frac{\langle \nabla f(x), x - x^* \rangle}{\|x - x^*\|} \right)^2, \tag{C.4}
$$

or equivalently,

$$
\left( \frac{\mu}{2} \right)^{1/2} \left( 1 - \frac{\rho}{\mu} \right) (f(x) - f^*)^{1/2} \leq \frac{\langle \nabla f(x), x - x^* \rangle}{\|x - x^*\|}.
$$

*Proof.* By first applying weak convexity and then the local quadratic growth condition for $x^* \in \mathcal{X}^*$ with $f(x^*) = f^*$ and for all $x \in [f \leq f^* + \zeta] \cap P$, it holds:

$$
\begin{aligned}
f(x) - f^* &= f(x) - f(x^*) \\
&\leq \langle \nabla f(x), x - x^* \rangle + \frac{\rho}{2} \|x - x^*\|^2 \\
&\leq \langle \nabla f(x), x - x^* \rangle + \frac{\rho}{\mu} (f(x) - f^*),
\end{aligned}
$$

which implies

$$\left(1 - \frac{\rho}{\mu}\right)(f(x) - f^*) \leq \langle \nabla f(x), x - x^* \rangle = \frac{\langle \nabla f(x), x - x^* \rangle}{\|x - x^*\|} \|x - x^*\|. \tag{C.5}$$

Using the local quadratic growth condition again, we have

$$\left(1 - \frac{\rho}{\mu}\right)(f(x) - f^*) \leq \frac{\langle \nabla f(x), x - x^* \rangle}{\|x - x^*\|} \left(\frac{2}{\mu}(f(x) - f^*)\right)^{1/2},$$

which provides

$$\left(\frac{\mu}{2}\right)^{1/2} \left(1 - \frac{\rho}{\mu}\right)(f(x) - f^*)^{1/2} \leq \frac{\langle \nabla f(x), x - x^* \rangle}{\|x - x^*\|},$$

or equivalently

$$f(x) - f^* \leq \frac{2\mu}{(\mu - \rho)^2} \left(\frac{\langle \nabla f(x), x - x^* \rangle}{\|x - x^*\|}\right)^2.$$

$\square$

**Remark C.6.** In the same vein as the discussion in Remark C.4, we can immediately relate the local $\mu$-quadratic growth condition with $\zeta > 0$ to the local Polyak–Łojasiewicz (PL) inequality. Using Equation C.4 in Lemma C.5, for all $x \in [f \leq f^* + \zeta] \cap P$, we have

$$f(x) - f^* \leq \frac{2\mu}{(\mu - \rho)^2} \|\nabla f(x)\|^2, \tag{C.6}$$

which is equivalent to the PL inequality (Polyak, 1963), also called the gradient dominance property (Combettes & Pokutta, 2020), with $c = \frac{\sqrt{2\mu}}{\mu - \rho}$. The PL inequality is also equivalent to the Łojasiewicz inequality with $\theta = \frac{1}{2}$. Strongly convex functions satisfy the PL inequality (Braun et al., 2025, Lemma 2.13). Note that we will not have the primal gap bound under the HEB condition and weak convexity because an inequality like Equation C.5 does not follow from the HEB condition.

### C.3 SCALING INEQUALITIES, GEOMETRIC HÖLDER ERROR BOUNDS, AND CONTRACTIONS

In the following, we will now bring things together to derive tools that will be helpful in establishing convergence rates.

SCALING INEQUALITIES

Scaling inequalities are a key tool in establishing convergence rates of FW algorithms. We will introduce two such inequalities that we will use in the following. The first scaling inequality is useful for analyzing the case where the optimal solution lies in the relative interior of the feasible region. While its formulation in (Braun et al., 2025, Proposition 2.16) required $L$-smoothness of $f$, it is actually not used in the proof, and the results hold more broadly. We restate it here for the sake of completeness.

**Proposition C.7** (Scaling inequality from convexity when $x^* \in \text{int } P$ (Braun et al., 2025, Proposition 2.16)). Let $P \subset \mathbb{R}^n$ be a nonempty compact convex set. Let $f : \mathbb{R}^n \to (-\infty, +\infty]$ be $\mathcal{C}^1$ and convex on $P$. If there exists $r > 0$ so that $B(x^*, r) \subset P$ for a minimizer $x^*$ of $f$, then for all $x \in P$, we have

$$\langle \nabla f(x), x - v \rangle \geq r \|\nabla f(x)\| \geq \frac{r \langle \nabla f(x), x - x^* \rangle}{\|x - x^*\|},$$

where $v \in \arg\max_{u \in P} \langle \nabla f(x), x - u \rangle$.

When $f$ is not convex, assuming that $f$ is weakly convex and local quadratic growth, we have the scaling inequality for a nonconvex objective function.

**Proposition C.8** (Scaling inequality from weak convexity when $x^* \in \operatorname{int} P$). Let $P \subset \mathbb{R}^n$ be a nonempty compact convex set. Let $f : \mathbb{R}^n \to (-\infty, +\infty]$ be $\mathcal{C}^1$, $\rho$-weakly convex, local $\mu$-quadratic growth with $\zeta > 0$ on $P$. If there exists $r > 0$ so that $B(x^*, r) \subset P$ for a minimizer $x^*$ of $f$ and $\rho \leq \mu$, then for all $x \in [f \leq f^* + \zeta] \cap P$, we have

$$\langle \nabla f(x), x - v \rangle \geq r \|\nabla f(x)\| \geq \frac{r \langle \nabla f(x), x - x^* \rangle}{\|x - x^*\|},$$

where $v \in \arg\max_{u \in P} \langle \nabla f(x), x - u \rangle$.

*Proof.* We consider $x^* - rz$, where $z$ is a point with $\|z\| = 1$ and $\langle \nabla f(x), z \rangle = \|\nabla f(x)\|$. It holds that

$$\langle \nabla f(x), v \rangle \leq \langle \nabla f(x), x^* - rz \rangle = \langle \nabla f(x), x^* \rangle - r \|\nabla f(x)\|. \tag{C.7}$$

From weak convexity, for all $x \in [f \leq f^* + \zeta] \cap P$, we have

$$\begin{aligned} f^* - f(x) &\geq \langle \nabla f(x), x^* - x \rangle - \frac{\rho}{2} \|x - x^*\|^2 \\ &\geq \langle \nabla f(x), x^* - x \rangle - \frac{\rho}{\mu}(f(x) - f^*), \end{aligned}$$

where the last inequality holds due to the local quadratic growth condition. Because of $\rho \leq \mu$, this inequality implies

$$\langle \nabla f(x), x - x^* \rangle \geq \left(1 - \frac{\rho}{\mu}\right)(f(x) - f^*) \geq 0.$$

By rearranging Equation C.7 and using $\langle \nabla f(x), x - x^* \rangle \geq 0$, we obtain

$$\langle \nabla f(x), x - v \rangle \geq \langle \nabla f(x), x - x^* \rangle + r \|\nabla f(x)\| \geq r \|\nabla f(x)\|.$$

In addition, it holds that

$$\frac{r \langle \nabla f(x), x - x^* \rangle}{\|x - x^*\|} \leq \frac{r \|\nabla f(x)\| \|x - x^*\|}{\|x - x^*\|} = r \|\nabla f(x)\|,$$

where the first inequality holds because of the Cauchy–Schwarz inequality. $\qquad \square$

Lacoste-Julien & Jaggi (2015) defined a geometric distance-like constant of a polytope, known as the *pyramidal width*, to analyze the convergence of the away-step Frank–Wolfe algorithm (and other variants that use the pyramidal width) over polytopes. It can be interpreted as the minimal $\delta > 0$ satisfying the following scaling inequality Equation C.8, which plays a central role in establishing convergence rates for the away-step FW algorithm; see Braun et al. (2025) for an in-depth discussion.

**Lemma C.9** (Scaling inequality via pyramidal width (Braun et al., 2025, Theorem 2.26), (Lacoste-Julien & Jaggi, 2015, Theorem 3)). Let $P \subset \mathbb{R}^n$ be a polytope and let $\delta$ denote the pyramidal width of $P$. Let $x \in P$, and let $\mathcal{S}$ denote any set of vertices of $P$ with $x \in \operatorname{conv} \mathcal{S}$, where $\operatorname{conv} \mathcal{S}$ denotes the convex hull of $\mathcal{S}$. Let $\psi$ be any vector, so that we define $v^{\mathrm{FW}} = \arg\min_{v \in P} \langle \psi, v \rangle$ and $v^{\mathrm{A}} = \arg\max_{v \in \mathcal{S}} \langle \psi, v \rangle$. Then for any $y \in P$

$$\langle \psi, v^{\mathrm{A}} - v^{\mathrm{FW}} \rangle \geq \delta \frac{\langle \psi, x - y \rangle}{\|x - y\|}. \tag{C.8}$$

Note that Lemma C.9 does not require the convexity of $f$, and we will use it for nonconvex optimization. When $\phi$ is $\sigma$-strongly convex and $\langle \psi, x - y \rangle \geq 0$, Lemma C.9 implies

$$\langle \psi, v^{\mathrm{A}} - v^{\mathrm{FW}} \rangle^2 \geq \delta^2 \frac{\langle \psi, x - y \rangle^2}{\|x - y\|^2} \geq \delta^2 \sigma \frac{\langle \psi, x - y \rangle^2}{2 D_\phi(x, y)}.$$

Note that the last inequality of the above also holds for $\delta^2 \sigma \frac{\langle \psi, x - y \rangle^2}{2 D_\phi(y, x)}$, *i.e.*, with $x$ and $y$ swapped in the divergence.

GEOMETRIC HÖLDER ERROR BOUND CONDITION

We will now introduce the more compact notion of geometric Hölder error bound condition, which simply combines the pyramidal width and the HEB condition of the function $f$; see also (Braun et al., 2025, Lemma 2.27) for details when $q = 2$.

**Lemma C.10** (Geometric Hölder error bound). Let $P$ be a polytope with pyramidal width $\delta > 0$ and let $f$ be a convex function and satisfy the HEB condition with $v^{\mathrm{FW}} = \arg\min_{v \in P} \langle \nabla f(x), v \rangle$ and $v^{\mathrm{A}} = \arg\max_{v \in \mathcal{S}} \langle \nabla f(x), v \rangle$ with $\mathcal{S} \subseteq \mathrm{Vert}\, P$, so that $x \in \mathrm{conv}\, \mathcal{S}$, we have

$$f(x) - f^* \leq \left(\frac{q}{\mu}\right)^{\frac{1}{q-1}} \left(\frac{\langle \nabla f(x), v^{\mathrm{A}} - v^{\mathrm{FW}} \rangle}{\delta}\right)^{\frac{q}{q-1}}.$$

*Proof.* Combining Equation C.8 with Lemma C.3 for $\psi = \nabla f(x)$, we have

$$f(x) - f^* \leq \left(\frac{q}{\mu}\right)^{\frac{1}{q-1}} \left(\frac{\langle \nabla f(x), x - x^* \rangle}{\|x - x^*\|}\right)^{\frac{q}{q-1}} \leq \left(\frac{q}{\mu}\right)^{\frac{1}{q-1}} \left(\frac{\langle \nabla f(x), v^{\mathrm{A}} - v^{\mathrm{FW}} \rangle}{\delta}\right)^{\frac{q}{q-1}}.$$

$\square$

FROM CONTRACTIONS TO CONVERGENCE RATES

Besides scaling inequalities and the geometric Hölder error bound, we also utilize the following lemma for convex and nonconvex optimization, which allows us to turn a contraction into a convergence rate.

**Lemma C.11** (From contractions to convergence rates (Braun et al., 2025, Lemma 2.21)). Let $\{h_t\}_t$ be a decreasing sequence of positive numbers and $c_0$, $c_1$, $c_2$, $\theta_0$ be positive numbers with $c_1 < 1$ such that $h_1 \leq c_0$ and $h_t - h_{t+1} \geq h_t \min\{c_1, c_2 h_t^{\theta_0}\}$ for $t \geq 1$, then

$$h_t \leq \begin{cases} c_0(1 - c_1)^{t-1} & \text{if } 1 \leq t \leq t_0, \\ \frac{(c_1/c_2)^{1/\theta_0}}{(1 + c_1\theta_0(t - t_0))^{1/\theta_0}} = \mathcal{O}(1/t^{1/\theta_0}) & \text{if } t \geq t_0, \end{cases}$$

where

$$t_0 := \max\left\{ \left\lfloor \log_{1-c_1}\left(\frac{(c_1/c_2)^{1/\theta_0}}{c_0}\right) \right\rfloor + 2, 1 \right\}.$$

In particular, we have $h_t \leq \epsilon$ if $t \geq t_0 + \frac{1}{\theta_0 c_2 \epsilon^{\theta_0}} - \frac{1}{\theta_0 c_1}$ and $\epsilon \leq (c_1/c_2)^{1/\theta_0}$.

# D PROOF OF CONVERGENCE ANALYSIS FOR CONVEX OPTIMIZATION

We would also like to stress that while we formulate some of the results for the case that $x^* \in \mathrm{int}\, P$, the results can be extended to the case that $x^* \in \mathrm{ri}\, P$, *i.e.*, the relative interior of $P$, which we did not do for the sake of clarity. Basically, the analysis is performed in the affine space spanned by the optimal face of $P$ in that case; the interested reader is referred to Braun et al. (2025) for details on how to extend the results.

## D.1 SUBLINEAR CONVERGENCE

We recall a key property for analyzing convergence rates in convex optimization.

**Lemma D.1** (Primal gap, dual gap, and Frank–Wolfe gap for convex $f$ (Pokutta, 2024, Lemma 4.1)). Suppose that Assumptions 3.1(ii), (iv), and that $f$ is convex. Then for all $x \in P$, it holds:

$$f(x) - f^* \leq \langle \nabla f(x), x - x^* \rangle \leq \max_{v \in P} \langle \nabla f(x), x - v \rangle. \tag{D.1}$$

We establish a sublinear convergence rate of the FW algorithm under the smooth adaptable property. It is a similar result to (Vyguzov & Stonyakin, 2025, Theorem 1). We conducted its proof following Braun et al. (2025); Jaggi (2013); Pokutta (2024). The FW algorithm uses the open loop step-size, *i.e.*, $\gamma_t = \frac{2}{2+t}$ in the following theorem.

**Theorem D.2** (Primal convergence of the Frank–Wolfe algorithm). Suppose that Assumptions 3.1 and 4.1 hold. Let $D := \sqrt{\sup_{x,y \in P} D_\phi(x,y)}$ be the diameter of $P$ characterized by the Bregman distance $D_\phi$ and let $\nu \in (0,1]$. Consider the iterates of Algorithm 1 with the open loop step-size, i.e., $\gamma_t = \frac{2}{2+t}$. Then, it holds that, for all $t \geq 1$,

$$f(x_t) - f^* \leq \frac{2^{1+\nu} L D^2}{(t+2)^\nu}, \tag{D.2}$$

and hence for any accuracy $\epsilon > 0$ we have $f(x_t) - f^* \leq \epsilon$ for all $t \geq (2^{1+\nu} L D^2/\epsilon)^{1/\nu}$.

*Proof.* For some $\nu \in (0,1]$, we have

$$\begin{aligned}
f(x_t) - f(x_{t+1}) &\geq \gamma_t \langle \nabla f(x_t), x_t - v_t \rangle - L \gamma_t^{1+\nu} D_\phi(v_t, x_t) \\
&\geq \gamma_t (f(x_t) - f^*) - L \gamma_t^{1+\nu} D_\phi(v_t, x_t),
\end{aligned}$$

where the first inequality holds because of Lemma B.7 and the last inequality holds because of Lemma D.1. Subtracting $f^*$ on both sides, using $D_\phi(v_t, x_t) \leq D^2$, and rearranging leads to

$$f(x_{t+1}) - f^* \leq (1 - \gamma_t)(f(x_t) - f^*) + L \gamma_t^{1+\nu} D^2.$$

When $t = 0$, it follows $f(x_1) - f^* \leq L D^2 \leq 2 L D^2$. Now, we consider $t \geq 1$ and obtain

$$\begin{aligned}
f(x_{t+1}) - f^* &\leq (1 - \gamma_t)(f(x_t) - f^*) + L \gamma_t^{1+\nu} D^2 \\
&\leq \frac{t}{2+t}(f(x_t) - f^*) + \frac{2^{1+\nu}}{(2+t)^{1+\nu}} L D^2 \\
&\leq \frac{t}{2+t} \frac{2^{1+\nu} L D^2}{(2+t)^\nu} + \frac{2^{1+\nu}}{(2+t)^{1+\nu}} L D^2 \\
&= \frac{2^{1+\nu} L D^2}{(3+t)^\nu} \left( \frac{(3+t)^\nu (1+t)}{(2+t)^{1+\nu}} \right) \leq \frac{2^{1+\nu} L D^2}{(3+t)^\nu},
\end{aligned}$$

where the last inequality holds due to $(3+t)^\nu (1+t) \leq (2+t)^{1+\nu}$ with $0 < \nu \leq 1$. $\qquad\square$

If $\phi = \frac{1}{2}\|\cdot\|^2$ and $\nu = 1$ in Equation D.2, we have $f(x_t) - f^* \leq 4 L D^2/(t+2)$, which is the same as a sublinear convergence rate of the classical FW algorithm.

**Remark D.3.** While the convergence rate Equation D.2 is the same as Vyguzov & Stonyakin (2025), Vyguzov and Stonyakin assume that the triangle scaling property holds for $D_\phi$. In contrast, we require significantly weaker assumptions: it is enough to assume that $\phi$ is strictly convex due to Lemma B.3 in order to establish Theorem D.2.

### D.2    PROOF OF THEOREM 4.2

Because Algorithm 2 is well-defined (see Remark 3.2), the convergence result of the FW algorithm using the adaptive step-size strategy (Algorithm 2) is essentially the same as the one that uses Bregman short steps (Theorems 4.2 and 5.3); up to small errors arising from the approximation whose precise analysis we skip for the sake of brevity.

*Proof of Theorem 4.2.* Let $h_t := f(x_t) - f^*$. Using Lemma B.8, we have

$$h_t - h_{t+1} = f(x_t) - f(x_{t+1}) \geq \frac{\nu}{1+\nu} \langle \nabla f(x_t), x_t - v_t \rangle \gamma_t,$$

where $\gamma_t = \min \left\{ \left( \frac{\langle \nabla f(x_t), x_t - v_t \rangle}{L(1+\nu) D_\phi(v_t, x_t)} \right)^{\frac{1}{\nu}}, 1 \right\}$. We consider two cases: (i) $\gamma_t < 1$ and (ii) $\gamma_t = 1$.

(i) $\gamma_t < 1$: Using $\gamma_t = \left( \frac{\langle \nabla f(x_t), x_t - v_t \rangle}{L(1+\nu) D_\phi(v_t, x_t)} \right)^{\frac{1}{\nu}}$, we have

$$h_t - h_{t+1} \geq \frac{\nu}{1+\nu} \frac{\langle \nabla f(x_t), x_t - v_t \rangle^{1+1/\nu}}{(L(1+\nu) D_\phi(v_t, x_t))^{1/\nu}}$$

$$\geq \frac{\nu}{1+\nu} \frac{r^{1+1/\nu} \|\nabla f(x_t)\|^{1+1/\nu}}{(L(1+\nu))^{1/\nu} D^{2/\nu}}$$

$$\geq \frac{\nu}{1+\nu} \frac{r^{1+1/\nu}}{Mc^{1+1/\nu} D^{2/\nu}} h_t^{\frac{(1+\nu)(q-1)}{\nu q}},$$

where the second inequality holds from $\langle f(x_t), x_t - v_t \rangle \geq r\|\nabla f(x_t)\|$ in Proposition C.7, and the last inequality holds because $f$ satisfies the $c$-Łojasiewicz inequality with $c = (q/\mu)^{1/q}$ (see Remark C.4), and $M := (L(1+\nu))^{1/\nu}$.

(ii) $\gamma_t = 1$: Using Equation D.1, we have

$$h_t - h_{t+1} = f(x_t) - f(x_{t+1}) \geq \frac{\nu}{1+\nu} \langle \nabla f(x_t), x_t - v_t \rangle \geq \frac{\nu}{1+\nu}(f(x_t) - f^*) = \frac{\nu}{1+\nu} h_t,$$

where the second inequality holds due to the convexity of $f$.

From (i) and (ii), we have

$$h_t - h_{t+1} \geq \min\left\{ \frac{\nu}{1+\nu} h_t, \frac{\nu}{1+\nu} \frac{r^{1+1/\nu}}{Mc^{1+1/\nu} D^{2/\nu}} h_t^{\frac{(1+\nu)(q-1)}{\nu q}} \right\}.$$

When $q = 1 + \nu$, we have

$$h_t - h_{t+1} \geq \min\left\{ \frac{\nu}{1+\nu}, \frac{\nu}{1+\nu} \frac{r^{1+1/\nu}}{Mc^{1+1/\nu} D^{2/\nu}} \right\} \cdot h_t.$$

This inequality and the initial bound $f(x_1) - f^* \leq LD^2$ due to Lemma B.4 imply

$$h_t \leq \max\left\{ \frac{1}{1+\nu}, 1 - \frac{\nu}{1+\nu} \frac{r^{1+1/\nu}}{Mc^{1+1/\nu} D^{2/\nu}} \right\}^{t-1} LD^2.$$

On the other hand, when $q > 1 + \nu$, we have

$$h_t - h_{t+1} \geq \min\left\{ \frac{\nu}{1+\nu}, \frac{\nu}{1+\nu} \frac{r^{1+1/\nu}}{Mc^{1+1/\nu} D^{2/\nu}} h_t^{\frac{q-1-\nu}{\nu q}} \right\} \cdot h_t.$$

Using $f(x_1) - f^* \leq LD^2$ and Lemma C.11 with $c_0 = LD^2$, $c_1 = \frac{\nu}{1+\nu}$, $c_2 = \frac{\nu}{1+\nu} \frac{r^{1+1/\nu}}{Mc^{1+1/\nu} D^{2/\nu}}$, and $\theta_0 = \frac{q-1-\nu}{\nu q} > 0$ from $q > 1 + \nu$, we have the claim. $\qquad\square$

### D.3 PROOF OF THEOREM 4.4

*Proof of Theorem 4.4.* By using the induced guarantee on the primal gap via Lemma C.10, we have

$$h_t = f(x_t) - f^* \leq \left(\frac{q}{\mu}\right)^{\frac{1}{q-1}} \left(\frac{\langle \nabla f(x_t), v_t^{\mathrm{A}} - v_t^{\mathrm{FW}} \rangle}{\delta}\right)^{\frac{q}{q-1}} \leq \left(\frac{q}{\mu}\right)^{\frac{1}{q-1}} \left(\frac{2\langle \nabla f(x_t), d_t \rangle}{\delta}\right)^{\frac{q}{q-1}},$$
(D.3)

where the last inequality holds because $d_t$ is either $x_t - v_t^{\mathrm{FW}}$ or $v_t^{\mathrm{A}} - x_t$ with $\langle \nabla f(x_t), d_t \rangle \geq \langle \nabla f(x), v^{\mathrm{A}} - v^{\mathrm{FW}} \rangle / 2$ in Lines 4 and 7 of Algorithm 3. We obtain

$$h_t - h_{t+1} \geq \frac{\nu}{1+\nu} \langle \nabla f(x_t), d_t \rangle \min\left\{ \gamma_{t,\max}, \left(\frac{\langle \nabla f(x_t), d_t \rangle}{L(1+\nu) D_\phi(v_t, x_t)}\right)^{\frac{1}{\nu}} \right\}$$

$$= \min\left\{ \gamma_{t,\max} \frac{\nu}{1+\nu} \langle \nabla f(x_t), d_t \rangle, \frac{\nu}{1+\nu} \frac{\langle \nabla f(x_t), d_t \rangle^{1+1/\nu}}{(L(1+\nu) D_\phi(v_t, x_t))^{1/\nu}} \right\}$$

$$\geq \min\left\{ \gamma_{t,\max} \frac{\nu h_t}{1+\nu}, \frac{\nu}{1+\nu} \frac{((\mu/q)^{1/q} h_t^{1-1/q} \delta/2)^{\frac{1+\nu}{\nu}}}{(L(1+\nu) D^2)^{1/\nu}} \right\}$$

$$= \min\left\{ \frac{\nu h_t}{1+\nu} \gamma_{t,\max}, \frac{\nu}{1+\nu} \frac{((\mu/q)^{1/q} \delta/2)^{\frac{1+\nu}{\nu}}}{(L(1+\nu) D^2)^{1/\nu}} h_t^{\frac{(1+\nu)(q-1)}{\nu q}} \right\}$$

where the first inequality holds due to Lemma B.8 with $\gamma_{t,\max} = 1$ (Frank–Wolfe steps) and $\gamma_{t,\max} = \frac{\lambda_{v_t^A,t}}{1-\lambda_{v_t^A,t}}$ (away steps), and the second inequality holds because of Equation D.3 and $\langle \nabla f(x_t), d_t \rangle \geq \langle \nabla f(x_t), x_t - v_t \rangle \geq h_t$.

In $(\nu, q) = (1, 2)$, for Frank–Wolfe steps, $\gamma_{t,\max} = 1 \geq \mu\delta^2/LD^2 \geq \mu\delta^2/32LD^2$. For away steps, we only rely on monotone progress $h_{t+1} < h_t$ because it is difficult to estimate $\gamma_{t,\max}$ below. However, $\gamma_{t,\max}$ cannot be small too often, which is the key point. Let us consider $\gamma_t = \gamma_{t,\max}$ in an away step. In that case, $v_t^A$ is removed from the active set $\mathcal{S}_{t+1}$ in line 14. Moreover, the active set can only grow in Frank–Wolfe steps (line 11). It is impossible to remove more vertices from $\mathcal{S}_{t+1}$ than have been added in Frank–Wolfe steps. Therefore, at most half of iterations until $t$ iterations are in away steps, i.e., in all other steps, we have $\gamma_t = \frac{\langle \nabla f(x_t), d_t \rangle}{2LD_\phi(v_t,x_t)} < \gamma_{t,\max}$ and then $h_t - h_{t+1} \geq \frac{\mu\delta^2}{32LD^2} h_t$. Because we have $h_{t+1} \leq \left(1 - \frac{\mu\delta^2}{32LD^2}\right) h_t$ for at least half of the iterations and $h_{t+1} \leq h_t$ for the rest, we obtain

$$f(x_t) - f^* \leq \left(1 - \frac{\mu}{32L}\frac{\delta^2}{D^2}\right)^{\lceil (t-1)/2 \rceil} LD^2.$$

In $q > 1 + \nu$, we have $h_t - h_{t+1} \geq \min\left\{\frac{\nu}{1+\nu}, \frac{\nu}{1+\nu}\frac{((\mu/q)^{1/q}\delta/2)^{\frac{1+\nu}{\nu}}}{(L(1+\nu)D^2)^{1/\nu}}h_t^{\frac{q-1-\nu}{\nu q}}\right\} \cdot h_t$ for at least half of the iterations (Frank–Wolfe steps) and $h_{t+1} \leq h_t$ for the rest (away steps). The initial bound $f(x_1) - f^* \leq LD^2$ holds generally for the Frank–Wolfe algorithm (see the proof of Theorem D.2 and (Braun et al., 2025, Remark 2.4)). We use Lemma C.11 with $c_0 = LD^2$, $c_1 = \frac{\nu}{1+\nu}$, $c_2 = c_1\frac{((\mu/q)^{1/q}\delta/2)^{\frac{1+\nu}{\nu}}}{(L(1+\nu)D^2)^{1/\nu}}$, and $\theta_0 = \frac{q-1-\nu}{\nu q} > 0$ from $q > 1 + \nu$ and obtain the claim. $\square$

**Remark D.4** (Compatibility of parameters). The condition $q > 1 + \nu$ is necessary in case that $(\nu, q) \neq (1, 2)$. The reason is as follows. In the case $q = 1 + \nu$, for Frank–Wolfe steps, we have

$$\frac{((\mu/q)^{1/q}\delta/2)^{\frac{1+\nu}{\nu}}}{(L(1+\nu)D^2)^{1/\nu}} = \left(\frac{\mu}{L}\right)^{1/\nu}\left(\frac{\delta}{D}\right)^{\frac{1+\nu}{\nu}}\frac{1}{2^{\frac{1+\nu}{\nu}}(1+\nu)^{2/\nu}D^{(1-\nu)/\nu}}, \tag{D.4}$$

which might be greater than 1 because $\frac{1}{D^{(1-\nu)/\nu}} \geq 1$ if $D < 1$ and $\nu$ are small enough. In order to make Equation D.4 smaller than 1, $\nu$ should be 1, i.e., $(\nu, q) = (1, 2)$. When $\nu = 1$ and $q \neq 2$, it reduces to the $q > 1 + \nu$ case.

# E  PROOF OF CONVERGENCE ANALYSIS FOR NONCONVEX OPTIMIZATION

## E.1  GLOBAL CONVERGENCE

We show that Algorithm 1 with $\gamma_t = \gamma := 1/(T+1)^{\frac{1}{1+\nu}}$ globally converges to a stationary point, where $T \in \mathbb{N}$ is the number of iterations. Its proof is inspired by (Lacoste-Julien, 2016, Theorem 1) and (Pokutta, 2024, Theorem 4.7) and identical to those for the case when $\phi = \frac{1}{2}\|\cdot\|^2$.

**Theorem E.1** (Global sublinear convergence for nonconvex optimization). Suppose that Assumption 3.1 holds. Let $\nu > 0$ and $D := \sqrt{\sup_{x,y\in P} D_\phi(x,y)}$ be the diameter of $P$ characterized by $D_\phi$ and let $T \in \mathbb{N}$. Then, the iterates of the FW algorithm with $\gamma_t = \gamma := 1/(T+1)^{\frac{1}{1+\nu}}$ satisfy

$$G_T := \min_{0\leq t\leq T}\max_{v_t\in P}\langle \nabla f(x_t), x_t - v_t \rangle \leq \frac{2\max\{h_0, LD^2\}}{(T+1)^{\frac{\nu}{1+\nu}}},$$

where $h_0 = f(x_0) - f^*$ is the primal gap at $x_0$.

*Proof.* Substituting $x_{t+1}$ for $x_+$ and $x_t$ for $x$ in Lemma B.7, we have the following inequality:

$$f(x_t) - f(x_{t+1}) \geq \gamma\langle \nabla f(x_t), x_t - v_t \rangle - L\gamma^{1+\nu}D_\phi(v_t, x_t).$$

Summing up the above inequality along $t = 1, \ldots, T$ and rearranging provides

$$\gamma \sum_{t=0}^{T} \langle \nabla f(x_t), x_t - v_t \rangle \leq f(x_0) - f(x_{T+1}) + \gamma^{1+\nu} \sum_{t=0}^{T} LD_\phi(x_t, v_t)$$

$$\leq f(x_0) - f^* + \gamma^{1+\nu} \sum_{t=0}^{T} LD^2 = h_0 + \gamma^{1+\nu}(T+1)LD^2.$$

Dividing by $\gamma(T+1)$ on the both sides, we obtain

$$G_T \leq \frac{1}{T+1} \sum_{t=0}^{T} \langle \nabla f(x_t), x_t - v_t \rangle \leq \frac{h_0}{\gamma(T+1)} + \gamma^\nu LD^2,$$

which, for $\gamma = 1/(T+1)^{\frac{1}{1+\nu}}$, implies

$$G_T \leq \frac{1}{T+1} \sum_{t=0}^{T} \langle \nabla f(x_t), x_t - v_t \rangle \leq (h_0 + LD^2)(T+1)^{-\frac{\nu}{1+\nu}} \leq 2\max\{h_0, LD^2\}(T+1)^{-\frac{\nu}{1+\nu}}.$$

This is the desired claim. □

As mentioned above, we generalize prior similar results. In fact, in the case where $\phi = \frac{1}{2}\|\cdot\|^2$ and $\nu = 1$, we have $\gamma_t = \frac{1}{\sqrt{T+1}}$ and obtain as guarantee

$$\min_{0 \leq t \leq T} \max_{v_t \in P} \langle \nabla f(x_t), x_t - v_t \rangle \leq \frac{2\max\{h_0, LD^2\}}{\sqrt{T+1}},$$

which is the same rate as (Lacoste-Julien, 2016, Theorem 1).

### E.2 LOCAL SUBLINEAR CONVERGENCE

We prove a lemma used in the proof of convergence analysis.

**Lemma E.2.** For any $\nu \in (0, 1]$ and $t \geq 0$, it holds that

$$h(t) := \frac{(t+3)^\nu(t+2-\nu)}{(t+2)^{1+\nu}} \leq 1. \tag{E.1}$$

*Proof.* We have

$$\lim_{t \to \infty} h(t) = \lim_{t \to \infty} \frac{(t+3)^\nu(t+2-\nu)}{(t+2)^{1+\nu}} = \lim_{t \to \infty} \left(1 + \frac{1}{t+2}\right)^\nu \left(1 - \frac{\nu}{t+2}\right) = 1.$$

We take the logarithm of $h(t)$, that is, $\log h(t) = \nu \log(t+3) + \log(t+2-\nu) - (1+\nu)\log(t+2)$ and obtain

$$\frac{h'(t)}{h(t)} = \frac{\nu}{t+3} + \frac{1}{t+2-\nu} - \frac{1+\nu}{t+2} = \frac{-2\nu^2 + 10\nu}{(t+2)(t+3)(t+2-\nu)} > 0,$$

where the last inequality holds for $\nu \in (0, 1]$ and $t \geq 0$. Since $h(t) > 0$ for $t \geq 0$, the above inequality implies $h'(t) > 0$. Therefore, we have $\sup h(t) = 1$, which implies $h(t) \leq 1$. □

Now we show sublinear convergence with $\gamma_t = \frac{2}{2+t}$. The proof is a modified version of Theorem D.2.

**Theorem E.3** (Local sublinear convergence). Suppose that Assumptions 3.1, 4.3, and 5.1 hold. Let $\nu \in (0, 1]$ and let $D := \sqrt{\sup_{x,y \in P} D_\phi(x, y)}$ be the diameter of $P$ characterized by the Bregman distance $D_\phi$. Consider the iterates of Algorithm 1 with the open loop step-size, *i.e.*, $\gamma_t = \frac{2}{2+t}$. Then, if $\rho/\sigma \leq L$ and $\frac{3\rho}{\mu} \leq 2 - \nu$ hold, it holds that, for all $t \geq 1$,

$$f(x_t) - f^* \leq \frac{2^{1+\nu}\mu LD^2}{\rho(t+2)^\nu}, \tag{E.2}$$

and hence for any accuracy $\epsilon > 0$ we have $f(x_t) - f^* \leq \epsilon$ for all $t \geq (2^{1+\nu}\mu LD^2/\rho\epsilon)^{1/\nu}$.

*Proof.* Using Lemma B.7, we have

$$f(x_t) - f(x_{t+1}) \geq \gamma_t \langle \nabla f(x_t), x_t - v_t \rangle - L\gamma_t^{1+\nu} D_\phi(v_t, x_t)$$
$$\geq \gamma_t \left( f(x_t) - f^* - \frac{\rho}{2} \|x_t - x^*\|^2 \right) - L\gamma_t^{1+\nu} D_\phi(v_t, x_t),$$

where the last inequality holds because of Equation C.1 in Lemma C.2. Subtracting $f^*$ and rearranging provides

$$f(x_{t+1}) - f^* \leq (1 - \gamma_t)(f(x_t) - f^*) + \frac{\rho\gamma_t}{2}\|x_t - x^*\|^2 + L\gamma_t^{1+\nu} D_\phi(v_t, x_t) \qquad \text{(E.3)}$$

$$\leq \left( 1 - \left( 1 - \frac{\rho}{\mu} \right) \gamma_t \right) (f(x_t) - f^*) + L\gamma_t^{1+\nu} D^2, \qquad \text{(E.4)}$$

where the last inequality holds due to the quadratic growth condition. For $t = 0$, using Equation E.3, we have

$$f(x_1) - f^* \leq \frac{\rho}{2}\|x_0 - x^*\|^2 + LD_\phi(v_0, x_0) \leq \left( \frac{\rho}{\sigma} + L \right) D^2 \leq 2LD^2 \leq \frac{2\mu}{\rho} LD^2, \qquad \text{(E.5)}$$

where the second inequality holds because $\phi$ is $\sigma$-strongly convex and the last inequality holds because of $3 < 6/(2 - \nu) \leq 2\mu/\rho$. Now we consider $t \geq 1$. Using Equation E.4 and $\gamma_t = \frac{2}{2+t}$, we have

$$f(x_{t+1}) - f^* \leq \left( t + \frac{2\rho}{\mu} \right) \frac{2^{1+\nu}\mu LD^2}{\rho(t+2)^{1+\nu}} + \frac{2^{1+\nu} LD^2}{(t+2)^{1+\nu}}$$
$$= \left( t + \frac{2\rho}{\mu} + \frac{\rho}{\mu} \right) \frac{\mu}{\rho} \frac{2^{1+\nu} LD^2}{(t+2)^{1+\nu}}$$
$$= \frac{\mu}{\rho} \frac{2^{1+\nu} LD^2}{(t+3)^\nu} \left( \frac{(t+3)^\nu(t+3\rho/\mu)}{(t+2)^{1+\nu}} \right)$$
$$\leq \frac{2^{1+\nu}\mu LD^2}{\rho(t+3)^\nu},$$

where the last inequality holds because of $(t+3)^\nu(t+3\rho/\mu) \leq (t+3)^\nu(t+2-\nu) \leq (t+2)^{1+\nu}$ from Equation E.1. $\qquad \square$

We can apply the quadratic growth condition again as before and obtain

$$\text{dist}(x_t, \mathcal{X}^*)^2 \leq \frac{2}{\mu}(f(x_t) - f^*) \leq \frac{2^{2+\nu} LD^2}{\rho(t+2)^\nu},$$

and hence

$$\text{dist}(x_t, \mathcal{X}^*) \leq \frac{2^{1+\nu/2} D\sqrt{L}}{\sqrt{\rho}(t+2)^{\nu/2}}.$$

Note that Theorem E.3 does not require knowledge of the number of iterations $T$ ahead of time compared to Theorem E.1. We stress, nonetheless, that the latter can also be adjusted using a different step-size strategy to obtain an any-time guarantee; see Braun et al. (2025) for details for the standard Euclidean case, which can be generalized to our setup. Moreover, we can apply Theorem E.3 to the Euclidean FW algorithm, *i.e.*, $\phi = \frac{1}{2}\|\cdot\|^2$ and $\nu = 1$. Using Equation E.2, $D_{\text{Euc}} := \sup_{x,y \in P} \|x - y\|^2$, and $D_{\text{Euc}} = \sqrt{2}D$, we obtain

$$f(x_t) - f^* \leq \frac{2\mu LD_{\text{Euc}}^2}{\rho(t+2)},$$

Theorem E.3 requires $\rho/\sigma \leq L$ and $\frac{3\rho}{\mu} \leq 2 - \nu$. These assumptions are easy to satisfy.

**Example E.4** (Example 3.1 in the arXiv version of Liao et al. (2024))**.** Let us consider

$$f(x) = \begin{cases} -x^2 + 1, & \text{if } -1 < x < -0.5, \\ 3(x+1)^2, & \text{otherwise.} \end{cases}$$

The function $f$ is not convex but $\rho$-weakly convex with $\rho = 2$. A global optimal solution of $f$ is $x^* = -1$ and its value is $f(x^*) = 0$. Moreover, $f$ satisfies the quadratic growth condition with $0 < \mu \leq 6$. Because $f$ is a quadratic function, we have $L \geq 6$. It holds that $2 = \rho \leq L$ and $1 = \frac{3\rho}{\mu} \leq 1 < 2 - \nu$ (set $\mu = 6$). Therefore, the assumption of Theorem E.3 holds.

In order to verify $\frac{3\rho}{\mu} \leq 2 - \nu$, it is easier to examine the sufficient condition $\frac{3\rho}{\mu} \leq 1$ instead because the exact value of $\nu$ is difficult to estimate. On the other hand, without loss of generality, we can assume $\sigma = 1$. When $\sigma > 1$, $\rho/\sigma < \rho \leq L$ holds. When $\sigma < 1$, we can use $\phi_1 = \phi + \frac{1-\sigma}{2}\|\cdot\|^2$, which is 1-strongly convex. When $\phi$ is convex but not strongly convex, we can use $\phi_2 = \phi + \frac{1}{2}\|\cdot\|^2$. Therefore, it suffices to verify $\rho/\sigma \leq \rho \leq L$, which often holds (for example, see an example of phase retrieval in Section 6.2).

### E.3 PROOF OF THEOREM 5.3

Theorems 5.3, 5.4, and E.9 require the weak convexity and the local quadratic growth condition of $f$ with $\rho \leq \mu$. We show several examples as follows.

**Example E.5.** We show some examples of $\rho$-weakly convex functions satisfying the (local) quadratic growth condition with $\rho < \mu$.

(i) Let $f(x) = \log(1 + x^2)$. $f$ attains the minimum $f^* = 0$ at $x^* = 0$. The function $f$ is 1/4-weakly convex because $f''(x) = -\frac{2(1-x^2)}{(1+x^2)^2}$ and its minimum is $-1/4$. Moreover, $(x - x^*)^2 = x^2 \leq \frac{2}{\mu}\log(1 + x^2) = \frac{2}{\mu}(f(x) - f^*)$ holds for any $x \in [f \leq \zeta]$. Letting $\mu = 1/3$, we have $6\log(1 + x^2) - x^2 \geq 0$ for any $x \in [f \leq 16]$. Note that $\zeta = 16$ is not an exact value. In this case, we have $1/4 = \rho < \mu = 1/3$. Through simple calculations and visualization, we can obtain a larger value of $\mu$ than $1/3$. This function is used for image restoration (Boţ et al., 2016; Stella et al., 2017).

(ii) Let $f(x) = \frac{1}{2}x^2 + 2(1 - e^{-x^2})$. $f$ attains the minimum $f^* = 0$ at $x^* = 0$. The function $f$ is $(8e^{-3/2} - 1)$-weakly convex because $f''(x) = 1 + 4e^{-x^2}(1 - 2x^2)$ and its minimum is $1 - 8e^{-3/2} \simeq -0.785$. Moreover, $(x - x^*) = x^2 \leq \frac{2}{\mu}(f(x) - f^*)$ holds for any $x \in [f \leq \zeta]$. Letting $\mu = 1$, we have $\frac{2}{\mu}(f(x) - f^*) - x^2 = 4(1 - e^{-x^2}) \geq 0$ for any $x \in \mathbb{R}$. The function $f$ satisfies the global quadratic growth condition with $8e^{-3/2} - 1 = \rho < \mu = 1$.

(iii) Let $f$ be the function defined in Example E.4. We set $\mu = \mu_0$ for some $\mu_0 \in (2, 6]$ because $\mu$ can be from $(0, 6]$. It holds that $2 = \rho < \mu$.

Because Algorithm 2 is well-defined (see also Remark 3.2), the convergence result of the FW algorithm with Algorithm 2 is the same as Theorem 5.3.

*Proof of Theorem 5.3.* Let $h_t := f(x_t) - f^*$. Using Lemma B.8, we have

$$h_t - h_{t+1} = f(x_t) - f(x_{t+1}) \geq \frac{\nu}{1+\nu}\langle \nabla f(x_t), x_t - v_t\rangle\gamma_t,$$

where $\gamma_t = \min\left\{1, \left(\frac{\langle\nabla f(x_t), x_t - v_t\rangle}{L(1+\nu)D_\phi(v_t, x_t)}\right)^{\frac{1}{\nu}}\right\}$. We consider two cases: (i) $\gamma_t < 1$ and (ii) $\gamma_t = 1$.

(i) $\gamma_t < 1$: We have

$$
\begin{aligned}
h_t - h_{t+1} &\geq \frac{\nu\langle\nabla f(x_t), x_t - v_t\rangle}{1+\nu}\min\left\{1, \left(\frac{\langle\nabla f(x_t), x_t - v_t\rangle}{L(1+\nu)D_\phi(v_t, x_t)}\right)^{1/\nu}\right\} \\
&\geq \frac{\nu}{1+\nu}\frac{\langle\nabla f(x_t), x_t - v_t\rangle^{1+1/\nu}}{(L(1+\nu)D_\phi(v_t, x_t))^{1/\nu}} \\
&\geq \frac{\nu}{1+\nu}\frac{r^{1+1/\nu}\|\nabla f(x_t)\|^{1+1/\nu}}{(L(1+\nu))^{1/\nu}D^{2/\nu}}
\end{aligned}
$$

$$\geq \frac{\nu}{1+\nu} \frac{r^{1+1/\nu}}{Mc^{1+1/\nu}D^{2/\nu}} h_t^{\frac{1+\nu}{2\nu}},$$

where the third inequality holds due to Proposition C.8 and the definition of $D$ and the last inequality holds due to the local PL inequality from Remark C.6 with $c = \frac{\sqrt{2\mu}}{\mu - \rho}$ and $M := (L(1+\nu))^{1/\nu}$.

(ii) In the case where $\gamma_t = 1$, *i.e.*, $\left( \frac{\langle \nabla f(x_t), x_t - v_t \rangle}{L(1+\nu)D_\phi(v_t, x_t)} \right)^{\frac{1}{\nu}} \geq 1$, this implies

$$\langle \nabla f(x_t), x_t - v_t \rangle \geq L(1+\nu)D_\phi(v_t, x_t), \tag{E.6}$$

because of $1/\nu > 1$. Using Lemma B.7 and Equation E.6, we have

$$
\begin{aligned}
h_{t+1} - h_t &\leq LD_\phi(v_t, x_t) - \langle \nabla f(x_t), x_t - v_t \rangle \\
&\leq -\frac{\nu}{1+\nu} \langle \nabla f(x_t), x_t - v_t \rangle \\
&\leq -\frac{\nu}{1+\nu} \left( h_t - \frac{\rho}{2} \|x_t - x^*\|^2 \right) \\
&\leq -\frac{\nu}{1+\nu} \left( 1 - \frac{\rho}{\mu} \right) h_t,
\end{aligned}
$$

where the third inequality holds because of Equation C.1 in Lemma C.2, and the last inequality holds because of the local quadratic growth condition of $f$ with $\mu > 0$. Therefore, we have

$$h_{t+1} \leq \frac{1}{1+\nu} \left( 1 + \frac{\rho\nu}{\mu} \right) h_t.$$

From (i) and (ii), we have

$$h_t - h_{t+1} \geq \min \left\{ \frac{\nu}{1+\nu} \left( 1 - \frac{\rho}{\mu} \right) h_t, \frac{\nu}{1+\nu} \frac{r^{1+1/\nu}}{Mc^{1+1/\nu}D^{2/\nu}} h_t^{\frac{1+\nu}{2\nu}} \right\}.$$

When $\nu = 1$, we have $h_t - h_{t+1} \geq \min \left\{ \frac{1}{2} \left( 1 - \frac{\rho}{\mu} \right), \frac{r^2}{2Mc^2D^2} \right\} \cdot h_t$. This inequality and the initial bound $f(x_1) - f^* \leq LD^2$ due to Lemma B.4 imply

$$h_t \leq \max \left\{ \frac{1}{2} \left( 1 + \frac{\rho}{\mu} \right), 1 - \frac{r^2}{2Mc^2D^2} \right\}^{t-1} LD^2.$$

On the other hand, when $\nu \in (0, 1)$,

$$h_t - h_{t+1} \geq \min \left\{ \frac{\nu}{1+\nu} \left( 1 - \frac{\rho}{\mu} \right), \frac{\nu}{1+\nu} \frac{r^{1+1/\nu}}{Mc^{1+1/\nu}D^{2/\nu}} h_t^{\frac{1-\nu}{2\nu}} \right\} \cdot h_t.$$

Using $f(x_1) - f^* \leq LD^2$ and Lemma C.11 with $c_0 = LD^2$, $c_1 = \frac{\nu}{1+\nu} \left( 1 - \frac{\rho}{\mu} \right)$, $c_2 = \frac{\nu}{1+\nu} \frac{r^{1+1/\nu}}{Mc^{1+1/\nu}D^{2/\nu}}$, and $\theta_0 = \frac{1-\nu}{2\nu} > 0$, we have the claim. $\qquad \square$

When $\phi = \frac{1}{2}\| \cdot \|^2$ and $\nu = 1$, we have local linear convergence with $\gamma_t = \min \left\{ \frac{\langle \nabla f(x_t), x_t - v_t \rangle}{L\|v_t - x_t\|^2}, 1 \right\}$ from Theorem 5.3. When $\phi = \frac{1}{2}\| \cdot \|$, *i.e.*, $D_\phi(x, y) = \frac{1}{2}\|x - y\|^2$, $D_{\text{Euc}} = \sqrt{2}D$ provides

$$f(x_t) - f^* \leq \max \left\{ \frac{1}{2} \left( 1 + \frac{\rho}{\mu} \right), 1 - \frac{r^2}{2Lc^2D_{\text{Euc}}^2} \right\}^{t-1} \frac{LD_{\text{Euc}}^2}{2}.$$

## E.4 PROOF OF THEOREM 5.4

In the same way as Theorem 4.4, the pyramidal width is the minimal $\delta > 0$ satisfying Lemma C.9 (see also (Braun et al., 2025, Lemma 2.26) or (Lacoste-Julien & Jaggi, 2015, Theorem 3)). An upper bound exists on the primal gap for weakly convex functions.

**Lemma E.6** (Upper bound on primal gap for weakly convex functions). Suppose that Assumptions 3.1 and 5.1 hold. Let $P$ be a polytope with the pyramidal width $\delta > 0$. Let $\mathcal{S}$ denote any set of vertices of $P$ with $x \in \operatorname{conv} \mathcal{S}$. Let $\psi$ be any vector, so that we define $v^{\mathrm{FW}} = \arg\min_{v \in P} \langle \psi, v \rangle$ and $v^{\mathrm{A}} = \arg\max_{v \in \mathcal{S}} \langle \psi, v \rangle$. If $\rho/\mu < 1$, then it holds that, for all $x \in [f \le f^* + \zeta] \cap P$,

$$f(x) - f^* \le \frac{2\mu}{(\mu - \rho)^2 \delta^2} \langle \nabla f(x), v^{\mathrm{A}} - v^{\mathrm{FW}} \rangle^2.$$

*Proof.* Using the weak convexity and the local quadratic growth condition of $f$, it holds that, for all $x \in [f \le f^* + \zeta] \cap P$,

$$
\begin{aligned}
f(x) - f(x^*) &\le \langle \nabla f(x), x - x^* \rangle + \frac{\rho}{2} \|x - x^*\|^2 \\
&\le \langle \nabla f(x), x - x^* \rangle + \frac{\rho}{\mu} (f(x) - f(x^*)),
\end{aligned}
$$

which implies

$$0 \le \left(1 - \frac{\rho}{\mu}\right)(f(x) - f(x^*)) \le \langle \nabla f(x), x - x^* \rangle,$$

where the first inequality follows from $1 - \rho/\mu > 0$. Using Equation C.8 with $\psi = \nabla f(x)$ and $y = x^*$ and the above inequality, we obtain

$$
\begin{aligned}
\frac{\langle \nabla f(x), v^{\mathrm{A}} - v^{\mathrm{FW}} \rangle^2}{\delta^2} &\ge \frac{\langle \nabla f(x), x - x^* \rangle^2}{\|x - x^*\|^2} \\
&\ge \left(1 - \frac{\rho}{\mu}\right)^2 \frac{(f(x) - f(x^*))^2}{\|x - x^*\|^2} \\
&\ge \frac{\mu}{2} \left(1 - \frac{\rho}{\mu}\right)^2 \frac{(f(x) - f(x^*))^2}{f(x) - f(x^*)} \\
&= \frac{\mu}{2} \left(1 - \frac{\rho}{\mu}\right)^2 (f(x) - f(x^*)),
\end{aligned}
$$

where the third inequality holds because of the local quadratic growth condition. $\square$

*Proof of Theorem 5.4.* By letting $h_t = f(x_t) - f^*$ and using Lemma E.6, we have

$$h_t \le \frac{2\mu}{(\mu - \rho)^2 \delta^2} \langle \nabla f(x_t), v^{\mathrm{A}} - v^{\mathrm{FW}} \rangle^2 \le \frac{8\mu \langle \nabla f(x_t), d_t \rangle^2}{(\mu - \rho)^2 \delta^2} = \frac{\langle \nabla f(x_t), d_t \rangle^2}{\omega \delta^2}, \qquad \text{(E.7)}$$

where the second inequality holds because $d_t$ is either $x_t - v_t^{\mathrm{FW}}$ or $v_t^{\mathrm{A}} - x_t$ with $\langle \nabla f(x_t), d_t \rangle \ge \langle \nabla f(x), v^{\mathrm{A}} - v^{\mathrm{FW}} \rangle / 2$ in Lines 4 and 7 of Algorithm 3. From Lemma B.8 with $\gamma_{t,\max} = 1$ (Frank–Wolfe steps) and $\gamma_{t,\max} = \frac{\lambda_{v_t^{\mathrm{A}},t}}{1 - \lambda_{v_t^{\mathrm{A}},t}}$ (away steps), we obtain

$$
\begin{aligned}
h_t - h_{t+1} &\ge \frac{\nu}{1+\nu} \langle \nabla f(x_t), d_t \rangle \min\left\{ \gamma_{t,\max}, \left( \frac{\langle \nabla f(x_t), d_t \rangle}{L(1+\nu) D_\phi(v_t, x_t)} \right)^{\frac{1}{\nu}} \right\} \\
&= \min\left\{ \gamma_{t,\max} \frac{\nu}{1+\nu} \langle \nabla f(x_t), d_t \rangle, \frac{\nu}{1+\nu} \frac{\langle \nabla f(x_t), d_t \rangle^{1+1/\nu}}{(L(1+\nu) D_\phi(v_t, x_t))^{1/\nu}} \right\} \\
&\ge \min\left\{ \frac{\nu \gamma_{t,\max}}{1+\nu} \left(1 - \frac{\rho}{\mu}\right) h_t, \frac{\nu}{1+\nu} \frac{(h_t \omega \delta^2)^{\frac{1+\nu}{2\nu}}}{(L(1+\nu) D^2)^{1/\nu}} \right\} \\
&= \min\left\{ \frac{\nu \gamma_{t,\max}}{1+\nu} \left(1 - \frac{\rho}{\mu}\right) h_t, \frac{\nu}{1+\nu} \frac{(\omega \delta^2)^{\frac{1+\nu}{2\nu}}}{(L(1+\nu) D^2)^{1/\nu}} h_t^{\frac{1+\nu}{2\nu}} \right\},
\end{aligned}
$$

where the second inequality holds because of $\langle \nabla f(x_t), d_t \rangle \ge \langle \nabla f(x_t), x_t - v_t \rangle \ge h_t - \frac{\rho}{2} \|x_t - v_t\|^2 \ge \left(1 - \frac{\rho}{\mu}\right) h_t$ (from Lemma C.2 and the quadratic growth condition of $f$) and Equation E.7.

In $\nu = 1$, for Frank–Wolfe steps, we have $\gamma_{t,\max} \left(1 - \frac{\rho}{\mu}\right) = \left(1 - \frac{\rho}{\mu}\right) \geq \frac{1}{8} \left(1 - \frac{\rho}{\mu}\right)^2 \frac{\delta^2}{D^2} = \frac{\omega\delta^2}{\mu D^2} \geq \frac{\omega\delta^2}{2LD^2}$ by $0 < \rho < \mu \leq L$ and $\delta \leq D$. For away steps, it seems that we only obtain a monotone progress $h_{t+1} < h_t$ because it is difficult to estimate $\gamma_{t,\max}$ below. However, $\gamma_{t,\max}$ cannot be small too often. Let us consider $\gamma_t = \gamma_{t,\max}$ in an away step. In that case, $v_t^{\mathrm{A}}$ is removed from the active set $\mathcal{S}_{t+1}$ in line 14. Moreover, the active set can only grow in Frank–Wolfe steps (line 11). It is impossible to remove more vertices from $\mathcal{S}_{t+1}$ than have been added in Frank–Wolfe steps. Therefore, at most half of iterations until $t$ iterations are in away steps, i.e., in all other steps, we have $\gamma_t = \frac{\langle \nabla f(x_t), d_t \rangle}{2LD_\phi(v_t, x_t)} < \gamma_{t,\max}$ and then $h_{t+1} \leq \left(1 - \frac{\omega\delta^2}{4LD^2}\right) h_t$. Because we have $h_{t+1} \leq \left(1 - \frac{\omega\delta^2}{4LD^2}\right) h_t$ for at least half of the iterations and $h_{t+1} \leq h_t$ for the rest, using $h_1 \leq 2LD^2$ from Equation E.5 and $\rho < L$, we obtain

$$f(x_t) - f^* \leq 2\left(1 - \frac{\omega}{4L}\frac{\delta^2}{D^2}\right)^{\lceil (t-1)/2 \rceil} LD^2.$$

In $\nu \in (0,1)$, we have $h_t - h_{t+1} \geq \min\left\{ \frac{\nu}{1+\nu}\left(1 - \frac{\rho}{\mu}\right), \frac{\nu}{1+\nu}\frac{(\omega\delta^2)^{\frac{1+\nu}{2\nu}}}{(L(1+\nu)D^2)^{1/\nu}} h_t^{\frac{1-\nu}{2\nu}} \right\} \cdot h_t$ for at least half of the iterations and $h_{t+1} \leq h_t$ for the rest. The initial bound $f(x_1) - f^* \leq (\rho + L)D^2 \leq 2LD^2$ holds from Equation E.5 and $\rho < L$. We use Lemma C.11 with $c_0 = 2LD^2$, $c_1 = \frac{\nu}{1+\nu}\left(1 - \frac{\rho}{\mu}\right)$, $c_2 = \frac{\nu}{1+\nu}\frac{(\omega\delta^2)^{\frac{1+\nu}{2\nu}}}{(L(1+\nu)D^2)^{1/\nu}}$, and $\theta_0 = \frac{1-\nu}{2\nu}$ and obtain the claim. $\qquad\square$

When $\phi = \frac{1}{2}\|\cdot\|^2$ and $\nu = 1$, the local linear convergence of Algorithm 3 is equivalent to

$$f(x_t) - f(x^*) \leq \left(1 - \frac{\omega}{2L}\frac{\delta^2}{D_{\mathrm{Euc}}^2}\right)^{\lceil (t-1)/2 \rceil} LD_{\mathrm{Euc}}^2.$$

### E.5 LOCAL LINEAR CONVERGENCE OVER UNIFORMLY CONVEX SETS

In the convex optimization case, Canon & Cullum (1968) established an early lower bound on the convergence rate of the FW algorithm. However, in the special case of $P$ being strongly convex, Garber & Hazan (2015) showed that one can improve upon that lower bound. Kerdreux et al. (2021) establish it in the case of $P$ being uniformly convex. We will now carry over this result to establish local linear convergence for the case where $f$ is weakly convex and $L$-smad on $P$.

To this end, we recall the definition of uniformly convex sets.

**Definition E.7** $((\alpha, p)$-uniformly convex set (Braun et al., 2025, Definition 2.18), (Kerdreux et al., 2021, Definition 1.1)). Let $\alpha$ and $p$ be positive numbers. The set $P \subset \mathbb{R}^n$ is $(\alpha, p)$-*uniformly convex* with respect to the norm $\|\cdot\|$ if for any $x, y \in P$, any $\gamma \in [0,1]$, and any $z \in \mathbb{R}^n$ with $\|z\| \leq 1$ the following holds:

$$y + \gamma(x - y) + \gamma(1 - \gamma) \cdot \alpha\|x - y\|^p z \in P.$$

Moreover, $P$ is said to be *strongly convex* if $P$ is $(\alpha, 2)$-uniformly convex.

We will use the scaling condition for uniformly convex sets to establish linear convergence.

**Proposition E.8** (Scaling inequality (Braun et al., 2025, Proposition 2.19), (Kerdreux et al., 2021, Lemma 2.1)). Let $P$ be a full dimensional compact $(\alpha, p)$-uniformly convex set, $u$ any non-zero vector, and $v = \arg\min_{y \in P}\langle u, y \rangle$. Then for all $x \in P$

$$\frac{\langle u, x - v \rangle}{\|x - v\|^p} \geq \alpha\|u\|.$$

Now we establish local linear convergence for weakly convex optimization on uniformly convex sets.

**Theorem E.9** (Local linear convergence over uniformly convex sets). Suppose that Assumptions 3.1 and 5.1 with $\phi = \frac{1}{2}\|\cdot\|^2$ and $\mathrm{int}\,\mathrm{dom}\,\phi = \mathbb{R}^n$ hold and that $P$ is $(\alpha, p)$-uniformly convex set. Let $\nabla f$ be bounded away from 0, i.e., $\|\nabla f(x)\| \geq c > 0$ for all $x \in P$. Let $D_{\mathrm{Euc}} := \sup_{x,y\in P} \|x - y\|$ be the diameter of $P$. Consider the iterates of Algorithm 1 with $\gamma_t = \min\left\{ \frac{\langle \nabla f(x_t), x_t - v_t\rangle}{L\|x_t - v_t\|^2}, 1 \right\}$. Then, if $\rho < \mu$ and $\rho \leq L$, it holds that

$$
f(x_t) - f^* \leq
\begin{cases}
\max\left\{ \frac{1}{2}\left(1 + \frac{\rho}{\mu}\right), 1 - \left(1 - \frac{\rho}{\mu}\right)\frac{\alpha c}{2L} \right\}^{t-1} LD_{\mathrm{Euc}}^2 & \text{if } p = 2, \\[2mm]
\left(\frac{1}{2} + \frac{\rho}{2\mu}\right)^{t-1} LD_{\mathrm{Euc}}^2 & \text{if } 1 \leq t \leq t_0, p \geq 2, \\[2mm]
\frac{L\left((1-\rho/\mu)^{1-p/2}L/\alpha c\right)^{2/(p-2)}}{(1+(1-\rho/\mu)(1/2-1/p)(t-t_0))^{p/(p-2)}} = \mathcal{O}(1/t^{p/(p-2)}) & \text{if } t \geq t_0, p \geq 2,
\end{cases}
$$

for all $t \geq 1$ where

$$
t_0 := \max\left\{ \left\lceil \log_{\frac{1}{2}\left(1+\frac{\rho}{\mu}\right)} \frac{L\left((1-\rho/\mu)^{1-p/2}L/\alpha c\right)^{2/(p-2)}}{LD_{\mathrm{Euc}}^2} \right\rceil + 2, 1 \right\}.
$$

*Proof.* Let $g_t := \langle \nabla f(x_t), x_t - v_t\rangle$ and $h_t := f(x_t) - f^*$. Lemma B.8 with $\phi = \frac{1}{2}\|\cdot\|^2$ and $\nu = 1$ is followed by $f(x_t) - f(x_{t+1}) \geq \frac{g_t}{2}\gamma_t$. Using Proposition E.8, we have

$$
\begin{aligned}
h_t - h_{t+1} &\geq \frac{g_t}{2}\min\left\{ \frac{g_t}{L\|x_t - v_t\|^2}, 1 \right\} \\[2mm]
&\geq \frac{g_t}{2}\min\left\{ \frac{g_t^{1-2/p}\alpha^{2/p}\|\nabla f(x_t)\|^{2/p}}{L}, 1 \right\} \\[2mm]
&\geq \frac{1}{2}\left(h_t - \frac{\rho}{2}\|x - x^*\|^2\right)\min\left\{ \left(h_t - \frac{\rho}{2}\|x - x^*\|^2\right)^{1-2/p}\frac{(\alpha c)^{2/p}}{L}, 1 \right\} \\[2mm]
&\geq \frac{1}{2}\left(1 - \frac{\rho}{\mu}\right)\min\left\{ \left(1 - \frac{\rho}{\mu}\right)^{1-2/p}\frac{(\alpha c)^{2/p}}{L}h_t^{1-2/p}, 1 \right\}\cdot h_t,
\end{aligned}
$$

where the third inequality holds from Lemma C.2 and $\|\nabla f(x)\| \geq c$, and the last inequality holds because of the local quadratic growth property of $f$. The initial bound $h_1 \leq LD_{\mathrm{Euc}}^2 = 2LD^2$ holds from $\rho \leq L$ and Equation E.5. For $q = 2$, we have $h_t - h_{t+1} \geq \frac{1}{2}\left(1 - \frac{\rho}{\mu}\right)\min\left\{ \frac{\alpha c}{L}, 1 \right\}\cdot h_t$, which implies

$$
h_{t+1} \leq \max\left\{ \frac{1}{2}\left(1 + \frac{\rho}{\mu}\right), 1 - \left(1 - \frac{\rho}{\mu}\right)\frac{\alpha c}{2L} \right\}\cdot h_t.
$$

Thus, we have the claim. For $p > 2$, we use Lemma C.11 with $c_0 = LD_{\mathrm{Euc}}^2$, $c_1 = \frac{1}{2}\left(1 - \frac{\rho}{\mu}\right)$, $c_2 = c_1 \cdot \left(1 - \frac{\rho}{\mu}\right)^{1-2/p}\frac{(\alpha c)^{2/p}}{L}$, and $\theta_0 = 1 - 2/p$ and obtain the claim. $\square$

Note that Assumption 3.1 with $\phi = \frac{1}{2}\|\cdot\|^2$ and $\mathrm{int}\,\mathrm{dom}\,\phi = \mathbb{R}^n$ holds when $f$ is $L$-smooth over $P$. If $\rho \leq L$ does not hold, we can use the initial bound $h_1 \leq \frac{\rho+L}{2}D_{\mathrm{Euc}}^2$ from Equation E.5. In that case, a local linear rate in Theorem E.9 is unchanged. Moreover, we have local linear convergence without assuming $\|\nabla f(x)\| > 0$.

**Theorem E.10** (Local linear convergence over uniformly convex sets without $\|\nabla f(x)\| > 0$). Suppose that Assumptions 3.1 and 5.1 with $\phi = \frac{1}{2}\|\cdot\|^2$ and $\mathrm{int}\,\mathrm{dom}\,\phi = \mathbb{R}^n$ hold and that $P$ is $(\alpha, p)$-uniformly convex set. Let $D_{\mathrm{Euc}} := \sup_{x,y\in P}\|x - y\|$ be the diameter of $P$. Consider the iterates of Algorithm 1 with $\gamma_t = \min\left\{ \frac{\langle \nabla f(x_t), x_t - v_t\rangle}{L\|x_t - v_t\|^2}, 1 \right\}$. Then, if $\rho < \mu$ and $\rho \leq L$, it holds that

$$
f(x_t) - f^* \leq
\begin{cases}
\left(\frac{1}{2} + \frac{\rho}{2\mu}\right)^{t-1} LD_{\mathrm{Euc}}^2 & \text{if } 1 \leq t \leq t_0, p \geq 2, \\[2mm]
\frac{\left((1-\rho/\mu)^{2-p}Lc^2/\alpha^2\right)^{1/(p-1)}}{(1+(1/2)\cdot(1-\rho/\mu)(1-1/p)(t-t_0))^{p/(p-1)}} = \mathcal{O}(1/t^{p/(p-1)}) & \text{if } t \geq t_0, p \geq 2,
\end{cases}
$$

for all $t \geq 1$ where $c = \frac{\sqrt{2\mu}}{\mu - \rho}$ and

$$t_0 := \max \left\{ \left\lfloor \log_{\frac{1}{2}\left(1 + \frac{\rho}{\mu}\right)} \frac{L\left((1 - \rho/\mu)^{2-p} Lc^2/\alpha^2\right)^{1/(p-1)}}{LD_{\text{Euc}}^2} \right\rfloor + 2, 1 \right\}.$$

*Proof.* Using an argument similar to Theorem E.9, we obtain

$$\begin{aligned}
h_t - h_{t+1} &\geq \frac{g_t}{2} \min \left\{ \frac{g_t}{L\|x_t - v_t\|^2}, 1 \right\} \\
&\geq \frac{g_t}{2} \min \left\{ \frac{g_t^{1-2/p} \alpha^{2/p} \|\nabla f(x_t)\|^{2/p}}{L}, 1 \right\} \\
&\geq \frac{1}{2} \left( h_t - \frac{\rho}{2}\|x - x^*\|^2 \right) \min \left\{ \left( h_t - \frac{\rho}{2}\|x - x^*\|^2 \right)^{1-2/p} \frac{\alpha^{2/p}(h_t/c^2)^{1/p}}{L}, 1 \right\} \\
&\geq \frac{1}{2} \left( 1 - \frac{\rho}{\mu} \right) \min \left\{ \left( 1 - \frac{\rho}{\mu} \right)^{1-2/p} \frac{(\alpha c^{-1})^{2/p}}{L} h_t^{1-1/p}, 1 \right\} \cdot h_t,
\end{aligned}$$

where the third inequality holds from the PL inequality Equation C.6 with $c = \frac{\sqrt{2\mu}}{\mu - \rho}$. Therefore, we use Lemma C.11 with $c_0 = LD_{\text{Euc}}^2$, $c_1 = \frac{1}{2}\left(1 - \frac{\rho}{\mu}\right)$, $c_2 = c_1 \cdot \left(1 - \frac{\rho}{\mu}\right)^{1-2/p} \frac{(\alpha c^{-1})^{2/p}}{L}$, and $\theta_0 = 1 - 1/p$ and obtain the claim. $\qquad\square$

## F  EXPERIMENTS

We use the following notation:

- `BregFW`: the FW algorithm with the adaptive Bregman step-size strategy (Algorithm 2, our proposed)
- `BregAFW`: the away-step FW algorithm with the adaptive Bregman step-size strategy (Algorithm 3 using Algorithm 2, *i.e.*, $L_t, \nu_t, \gamma_t \leftarrow \texttt{step\_size}(f, \phi, x_t, v_t, L_{t-1}, \gamma_{t,\max})$, our proposed update)
- `EucFW`: the FW algorithm with the adaptive (Euclidean) step-size strategy (Pedregosa et al., 2020)
- `EucAFW`: the away-step FW algorithm with the adaptive (Euclidean) step-size strategy (Pedregosa et al., 2020)
- `ShortFW`: the FW algorithm with the (Euclidean) short step
- `ShortAFW`: the away-step FW algorithm with the (Euclidean) short step
- `OpenFW`: the FW algorithm with the open loop with $\gamma_t = \frac{2}{2+t}$
- `OpenAFW`: the away-step FW algorithm with the open loop with $\gamma_t = \frac{2}{2+t}$
- `MD`: the mirror descent (Nemirovskij & Yudin, 1983)
- `ProjGD`: the projected gradient descent algorithm (see, *e.g.*, Beck (2017))

### F.1  NONNEGATIVE LINEAR INVERSE PROBLEMS

Given a nonnegative matrix $A \in \mathbb{R}_+^{m \times n}$ and a nonnegative vector $b \in \mathbb{R}_+^m$, the goal of nonnegative (Poisson) linear inverse problems is to recover a signal $x \in \mathbb{R}_+^n$ such that $Ax \simeq b$. This class of problems has been studied in image deblurring (Bertero et al., 2009) and positron emission tomography (Vardi et al., 1985) as well as optimization (Bauschke et al., 2017; Takahashi & Takeda, 2025). Since the dimension of $x$ is often larger than the number of observations $m$, the system is indeterminate. From this point of view, we consider the constraint $\Delta_n := \{x \in \mathbb{R}_+^n \mid \sum_{j=1}^n x_j \leq 1\}$. Recovering $x$ can be formulated as a minimization problem:

$$\min_{x \in \Delta_n} \quad f(x) := d(Ax, b), \tag{F.1}$$

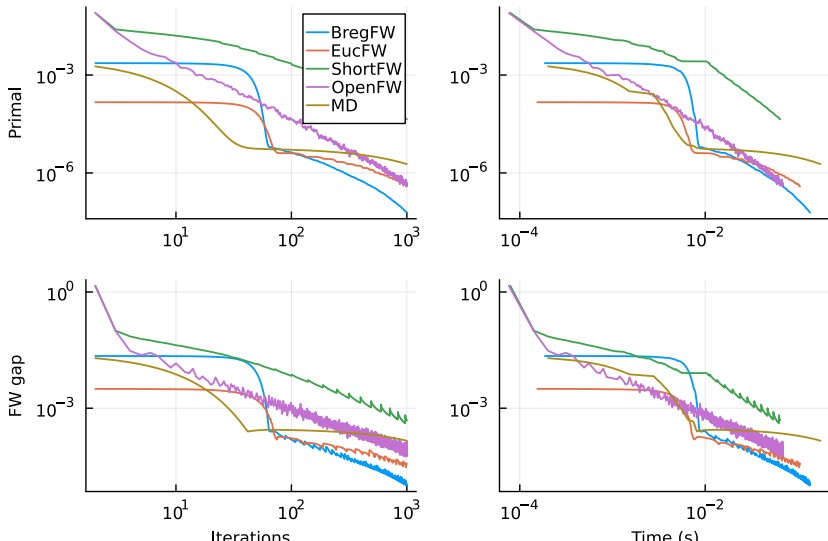

**Figure 5:** Log-log plot of primal and FW gaps on nonnegative linear inverse problem for $(m, n) = (100, 1000)$.

**Table 2:** Average values of the primal gap, the FW gap, and computational time (s) over 20 different instances of nonnegative linear inverse problems for $(m, n) = (100, 1000)$. The results are reported as mean $\pm$ standard deviation.

| algorithm | primal gap | FW gap | time (s) |
|---|---|---|---|
| BregFW | **6.9637e-08 $\pm$ 2.0e-08** | **1.1455e-05 $\pm$ 1.4e-06** | 1.4577e-01 $\pm$ 4.1e-02 |
| EucFW | 3.0287e-07 $\pm$ 9.4e-08 | 2.9223e-05 $\pm$ 4.1e-06 | 1.0778e-01 $\pm$ 5.8e-03 |
| ShortFW | 4.7470e-05 $\pm$ 3.5e-06 | 4.6135e-04 $\pm$ 7.2e-05 | 6.2518e-02 $\pm$ 3.6e-03 |
| OpenFW | 4.9576e-07 $\pm$ 6.3e-08 | 8.5382e-05 $\pm$ 1.3e-05 | 6.5036e-02 $\pm$ 6.3e-03 |
| MD | 2.2494e-06 $\pm$ 3.1e-07 | 1.7214e-04 $\pm$ 1.9e-05 | 1.8885e-01 $\pm$ 6.4e-03 |

where $d(x, y) := \sum_{i=1}^{m} \left( x_i \log \frac{x_i}{y_i} + y_i - x_i \right)$ is the KL divergence. Problem Equation F.1 is convex, while $\nabla f$ is not Lipschitz continuous on $\mathbb{R}_+^n$. The pair $(f, \phi)$ is $L$-smad on $\mathbb{R}_+^n$ with $\phi(x) = \sum_{j=1}^{n} x_j \log x_j$ and $L \geq \max_{1 \leq j \leq n} \sum_{i=1}^{m} a_{ij}$ from (Bauschke et al., 2017, Lemma 8).

We compared BregFW with EucFW, ShortFW, OpenFW, and the mirror descent algorithm (MD) (Nemirovskij & Yudin, 1983). The subproblem of MD can be solved in closed-form for $\{x \in \mathbb{R}_+^n \mid \sum_{j=1}^{n} x_j = 1\}$ by (Beck, 2017, Example 3.71), and this can be readily extended to $\Delta_n$. We used 1000 as maximum iteration limit and we generated $\tilde{A}$ from an i.i.d. normal distribution and set $a_{ij} = |\tilde{a}_{ij}| / \sum_{i=1}^{m} |\tilde{a}_{ij}|$. We also generated $\tilde{x}$ from an i.i.d. uniform distribution in $[0, 1]$ and set the ground truth $x^* = 0.8\tilde{x}/\sum_{j=1}^{n} \tilde{x}_j$ so that $x^* \in \text{int } \Delta_n$ (random seed 1234). All components of the initial point $x_0$ were $1/n$. For $(m, n) = (100, 1000)$, Figure 5 shows the primal gap $f(x_t) - f^*$ and the FW gap $\max_{v \in P} \langle \nabla f(x_t), x_t - v \rangle$ per iteration (left) and the primal gap per second (right). Table 2 shows the average values of the primal gap, the FW gap, and computational time over 20 different instances for $(m, n) = (100, 1000)$. Here, BregFW outperformed other algorithms, both in iterations and time.

### F.2 $\ell_p$ Loss Problem

We use an $\ell_2$ norm ball as a constraint, *i.e.*, $P = \{x \in \mathbb{R}^n \mid \|x\| \leq 1\}$. We compared BregFW with EucFW, ShortFW, and OpenFW. We generated $A$ from an i.i.d. normal distribution and normalized it so that $\|a_i\| = 1$. We also generated $\tilde{x}$ from an i.i.d. normal distribution and set $x^* = 0.8\tilde{x}/\|\tilde{x}\|$ to ensure $x^* \in \text{int } P$ (random seed 1234). The initial point $x_0$ was generated by computing an extreme

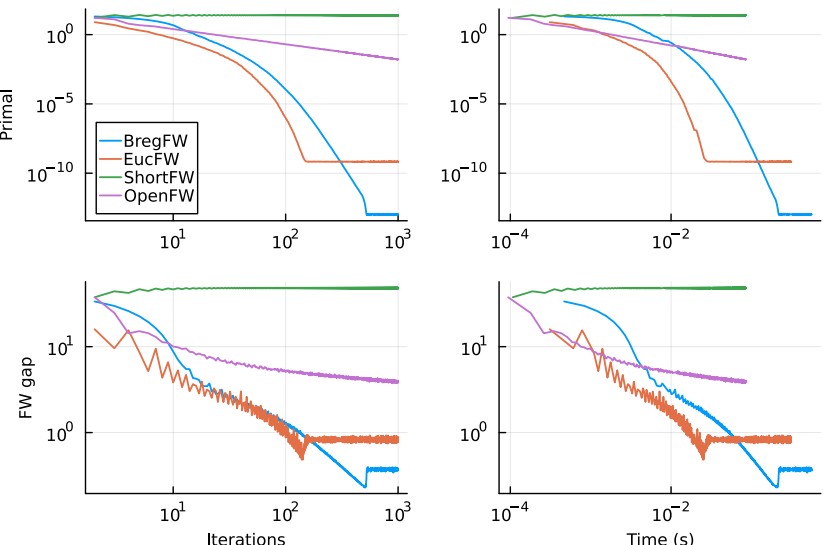

**Figure 6:** Log-log plot of primal and FW gaps on the $\ell_p$ loss problem for $(m, n) = (1000, 100)$.

**Table 3:** Average values of the primal gap, FW gap, and computational time (s) over 20 different instances of $\ell_p$ loss problems for $(m, n) = (1000, 100)$. The results are reported as mean $\pm$ standard deviation.

| algorithm | primal gap | FW gap | time (s) |
|---|---|---|---|
| BregFW | **1.0589e-13 $\pm$ 8.6e-15** | **3.7686e-01 $\pm$ 8.6e-03** | 5.6037e-01 $\pm$ 1.8e-02 |
| EucFW | 6.2980e-10 $\pm$ 9.9e-11 | 8.5940e-01 $\pm$ 1.0e-01 | 3.1421e-01 $\pm$ 7.4e-03 |
| ShortFW | 2.4697e+01 $\pm$ 1.2e+00 | 4.7621e+01 $\pm$ 8.6e-01 | 9.2352e-02 $\pm$ 3.7e-03 |
| OpenFW | 1.6961e-02 $\pm$ 9.9e-04 | 4.0077e+00 $\pm$ 1.0e-01 | 8.7292e-02 $\pm$ 3.5e-03 |

point of $P$ that minimizes the linear approximation of $f$. For $(n, m) = (100, 100)$ and $p = 1.1$, Figure 6 shows the primal gap $f(x_t) - f^*$ and the FW gap $\max_{v \in P}\langle \nabla f(x_t), x_t - v \rangle$ per iteration and those gaps per second up to 1000 iterations. Table 3 also shows the average performance over 20 different instances. BregFW outperformed the other algorithms. Since $\nabla f$ is not Lipschitz continuous, ShortFW did not reduce the primal and FW gaps.

### F.3 PHASE RETRIEVAL

We show the average performance for the setting of Section 6.2. Tables 4 and 5 show the average performance over 20 different instances of $(m, n) = (1000, 10000)$ and $(m, n) = (2000, 10000)$, respectively.

We also compared BregAFW with EucAFW, ShortAFW, and OpenAFW. In only this setting, we generated $x^*$ from an i.i.d. uniform distribution in $[0, 1]$ and did not normalize it (random seed 1234); that is, $x^*$ might be in the face of $P$. Figure 7 shows another setting's results for $(m, n) = (200, 200)$ and $K = 110$. Table 6 shows the average performance over 20 different instances for $(m, n) = (200, 200)$ and $K = 110$. The primal gap by BregAFW is the smallest among these algorithms, while ShortAFW has the smallest value of the FW gap.

### F.4 LOW-RANK MINIMIZATION

Given a symmetric matrix $M \in \mathbb{R}^{n \times n}$, our goal is to find $X \in \mathbb{R}^{n \times r}$ such that $M \simeq XX^{\mathsf{T}}$. This is accomplished by minimizing the function

$$\min_{X \in P} \quad f(X) := \frac{1}{2}\|XX^{\mathsf{T}} - M\|_F^2,$$

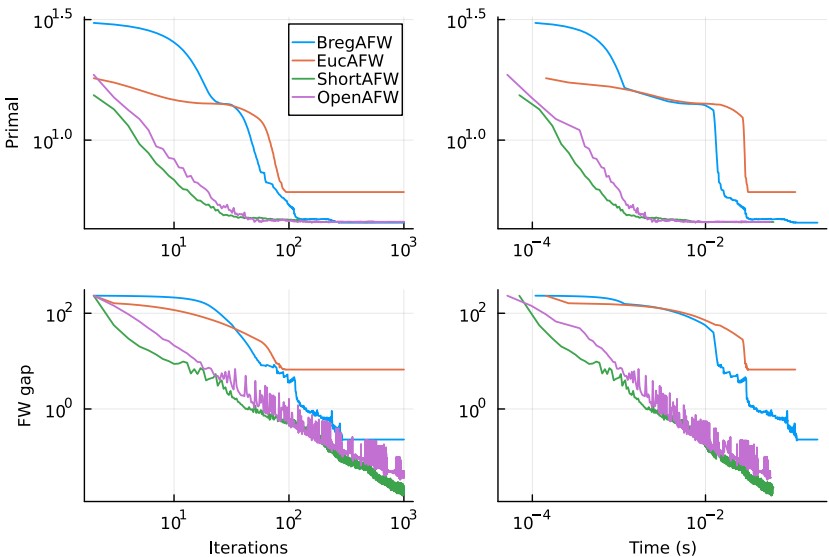

**Figure 7:** Log-log plot of primal and FW gaps on phase retrieval for $(m, n) = (200, 200)$ and $K = 110$ via AFW.

**Table 4:** Average values of the primal gap, FW gap, and computational time (s) over 20 different instances of phase retrieval for $(m, n) = (1000, 10000)$ and $K = 2000$. The results are reported as mean $\pm$ standard deviation.

| algorithm | primal gap | FW gap | time |
|---|---|---|---|
| BregFW | **1.3107e-11 $\pm$ 1.1e-12** | **2.1518e-02 $\pm$ 5.9e-02** | 4.6990e+01 $\pm$ 1.5e+01 |
| EucFW | 1.7170e+00 $\pm$ 3.9e+00 | 2.4612e+02 $\pm$ 4.5e+02 | 4.4986e+01 $\pm$ 3.2e+01 |
| ShortFW | 3.2739e+05 $\pm$ 1.9e+04 | 4.2232e+04 $\pm$ 5.0e+02 | 1.0942e+01 $\pm$ 3.2e-01 |
| OpenFW | 5.2496e-10 $\pm$ 3.9e-11 | 4.3802e+00 $\pm$ 5.5e-02 | 6.0353e+01 $\pm$ 1.6e+00 |

**Table 5:** Average values of the primal gap, FW gap, and computational time (s) over 20 different instances of phase retrieval for $(m, n) = (2000, 10000)$ and $K = 2000$. The results are reported as mean $\pm$ standard deviation.

| algorithm | primal gap | FW gap | time |
|---|---|---|---|
| BregFW | **6.6743e-12 $\pm$ 5.5e-13** | **9.5656e-08 $\pm$ 7.9e-09** | 7.2592e+01 $\pm$ 8.8e+00 |
| EucFW | 3.2499e+00 $\pm$ 7.1e+00 | 3.1304e+02 $\pm$ 4.8e+02 | 1.4348e+02 $\pm$ 4.3e+01 |
| ShortFW | 6.0937e+04 $\pm$ 2.3e+03 | 2.5461e+04 $\pm$ 2.3e+02 | 1.2648e+01 $\pm$ 2.4e-01 |
| OpenFW | 9.3513e-11 $\pm$ 4.9e-12 | 2.6628e+00 $\pm$ 1.7e-02 | 7.1916e+01 $\pm$ 1.3e+00 |

**Table 6:** Average values of the primal gap, FW gap, and computational time (s) over 20 different instances of phase retrieval for $(m, n) = (200, 200)$ and $K = 110$ via away-step FW algorithms. The results are reported as mean $\pm$ standard deviation.

| algorithm | primal gap | FW gap | time (s) |
|---|---|---|---|
| BregAFW | **4.3430e+00 $\pm$ 1.7e+00** | 9.7578e-01 $\pm$ 9.1e-01 | 2.3336e-01 $\pm$ 6.4e-02 |
| EucAFW | 4.3968e+00 $\pm$ 1.7e+00 | 1.2212e+00 $\pm$ 2.3e+00 | 3.9421e-01 $\pm$ 1.6e-01 |
| ShortAFW | 4.4092e+00 $\pm$ 1.7e+00 | **2.5978e-02 $\pm$ 7.3e-03** | 7.3540e-02 $\pm$ 3.5e-02 |
| OpenAFW | 4.4094e+00 $\pm$ 1.7e+00 | 7.1992e-02 $\pm$ 4.9e-02 | 5.8998e-02 $\pm$ 2.4e-03 |

where $P \subset \mathbb{R}^{n \times r}$. We assume that $r \leq n$. This problem is known as low-rank minimization (Dragomir et al., 2021). In this paper, we define $P = \{X \in \mathbb{R}^{n \times r} \mid \|X\|_* \leq \xi\}$, where $\|\cdot\|_*$ denotes the nuclear norm and $\xi \in \mathbb{R}_+$ for the low-rank assumption.

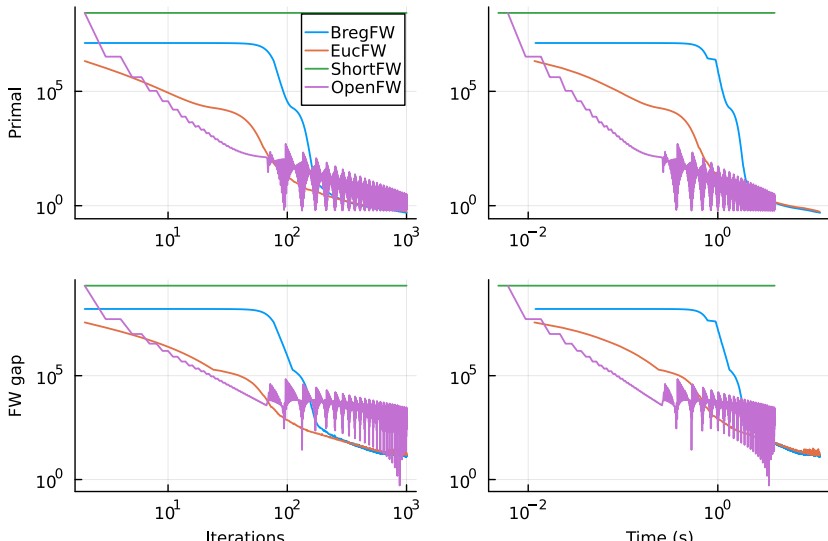

**Figure 8:** Log-log plot of primal and FW gaps on low-rank minimization for $(n, r) = (1000, 20)$.

**Table 7:** Average values of the primal gap, FW gap, and computational time (s) over 20 different instances of low-rank minimization for $(n, r) = (1000, 20)$. The results are reported as mean $\pm$ standard deviation.

| algorithm | primal gap | FW gap | time (s) |
|---|---|---|---|
| BregFW | **5.6519e-01 $\pm$ 4.2e-02** | **1.4727e+01 $\pm$ 1.0e+00** | 1.2602e+01 $\pm$ 4.5e-01 |
| EucFW | 5.7765e-01 $\pm$ 4.6e-02 | 1.7237e+01 $\pm$ 2.0e+00 | 1.2280e+01 $\pm$ 3.8e-01 |
| ShortFW | 2.7101e+08 $\pm$ 1.9e+06 | 2.1694e+09 $\pm$ 1.5e+07 | 4.1379e+00 $\pm$ 7.9e-02 |
| OpenFW | 1.9782e+00 $\pm$ 3.6e-01 | 2.1212e+03 $\pm$ 3.1e+02 | 4.1770e+00 $\pm$ 1.7e-01 |

We define

$$\phi(X) = \frac{1}{4}\|X\|_F^4 + \frac{1}{2}\|X\|_F^2.$$

There exists a constant $L$ such that the pair $(f, \phi)$ is $L$-smad on $\mathbb{R}^n$ (Dragomir et al., 2021). Additionally, $f(X)$ is weakly convex on any compact set due to Proposition C.1, which follows from the twice continuous differentiability of $f$.

We also compared BregFW with EucFW, ShortFW, and OpenFW. The parameter settings are the same as those in the previous subsection. We generated $X^*$ from an i.i.d. uniform distribution in $[0, 1]$, normalized each column of $X^*$, and set $M = X^*(X^*)^\mathsf{T}$ (random seed 42). The initial point $X_0$ was generated from an i.i.d. uniform distribution in $[0, 1]$. We set $\xi = 10\lambda_{\max}(M)$ for $P$. Figure 8 shows the primal and FW gaps per iteration and gaps per second for $(n, r) = (1000, 20)$ up to the 1000th iteration. Table 7 presents the average performance over 20 different instances for $(n, r) = (1000, 20)$. BregFW performed slightly better than EucFW. OpenFW also performed as fast as BregFW and EucFW, but its performance was unstable. ShortFW did not converge due to the lack of Lipschitz continuity of $\nabla f$.

### F.5 Nonnegative Matrix Factorization

Given a nonnegative matrix $V \in \mathbb{R}_+^{m \times n}$, nonnegative matrix factorization (NMF) aims to find nonnegative matrices $W \in \mathbb{R}_+^{m \times r}$ and $H \in \mathbb{R}_+^{r \times n}$ such that $V \simeq WH$. NMF can be formulated as a minimization problem of the loss function that measures the difference between $V$ and $WH$, i.e., $\min_{(W,H) \in P} f(W, H) := \frac{1}{2}\|WH - V\|_F^2$, where $P$ is a compact convex subset of $\mathbb{R}_+^{m \times r} \times \mathbb{R}_+^{r \times n}$. The objective function $f$ is weakly convex over $P$ due to Proposition C.1. The

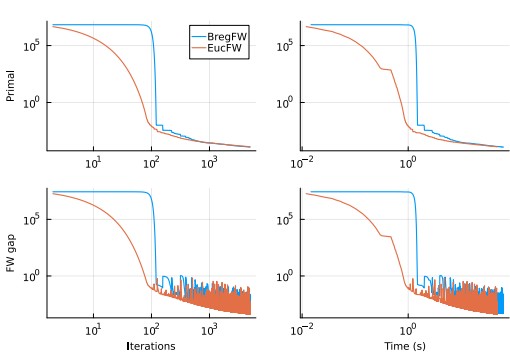

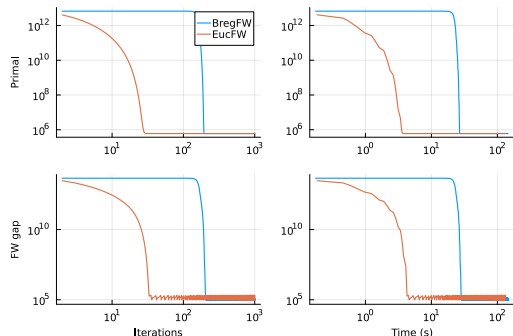

**Figure 9:** Log-log plot of the primal and the FW gaps on NMF for $(m, n, r) = (100, 5000, 20)$.

**Figure 10:** Log-log plot of the primal and the FW gaps on NMF for MovieLens 100K Dataset.

**Table 8:** Average values of the primal gap, FW gap, and computational time (s) over 20 different instances of NMF for $(m, n, r) = (100, 5000, 20)$. The results are reported as mean $\pm$ standard deviation.

| algorithm | primal gap | FW gap | time (s) |
|---|---|---|---|
| BregFW | **1.2017e-04 $\pm$ 5.7e-06** | 1.1515e-02 $\pm$ 1.4e-02 | 5.0983e+01 $\pm$ 1.1e+00 |
| EucFW | 1.2819e-04 $\pm$ 4.6e-06 | **7.6259e-04 $\pm$ 8.0e-04** | 4.3592e+01 $\pm$ 1.0e+00 |

gradient $\nabla f$ is not Lipschitz continuous, while $(f, \phi)$ is smooth adaptable (Mukkamala & Ochs, 2019) with $\phi(W, H) = \frac{1}{4}(\|W\|_F^2 + \|H\|_F^2)^2 + \frac{1}{2}(\|W\|_F^2 + \|H\|_F^2)$.

We used a box constraint $P = \{(W, H) \in \mathbb{R}^{m \times r} \times \mathbb{R}^{r \times n} \,|\, 0 \leq W_{lj} \leq 3, 0 \leq H_{lj} \leq 1\}$. We compared BregFW with EucFW because ShortFW and OpenFW stopped at the 2nd iteration. We generated $W^*$ from an i.i.d. uniform distribution in $[0, 1]$ and normalized each column of $W^*$ (random seed 42). We also generated $H^*$ from an i.i.d. Dirichlet distribution. The initial point $(W_0, H_0)$ was generated from an i.i.d. uniform distribution in $[0, 1]$. Figure 9 shows the primal and FW gaps for $(m, n, r) = (100, 5000, 20)$ up to the 5000th iteration. Table 8 shows the average performance over 20 different instances for $(m, n, r) = (100, 5000, 20)$ up to the 5000th iteration. BregFW is slightly better than EucFW in terms of the primal gap, while the FW gap for EucFW is smaller than that for BregFW.

Furthermore, we consider real-world data using the MovieLens 100K Dataset. We set $P = \{(W, H) \in \mathbb{R}^{m \times r} \times \mathbb{R}^{r \times n} \,|\, 0 \leq W_{lj} \leq 5.0, \|H\|_* \leq \xi\}$. For $\phi(W, H) = \frac{3}{4}(\|W\|_F^2 + \|H\|_F^2)^2 + \frac{\|V\|_F}{2}(\|W\|_F^2 + \|H\|_F^2)$ by (Mukkamala & Ochs, 2019, Proposition 2.1), $(f, \phi)$ is also $L$-smooth adaptable. Figure 10 shows the primal and FW gaps up to the 1000th iteration with $\xi = 10\sqrt{\lambda_{\max}(VV^\top)}$, where $\lambda_{\max}(\cdot)$ is the largest eigenvalue. The primal gap of BregFW is 6.041269e+05, and that of EucFW is 6.041272e+05. In this setting, the primal and FW gaps of BregFW are slightly better than those of EucFW.

