# OpenReview forum: "Fast Frank–Wolfe Algorithms with Adaptive Bregman Step-Size for Weakly Convex Functions"
_ICLR.cc/2026/Conference — ICLR 2026 Poster_

### Official Review · Reviewer_229D · 2025-10-31

**Soundness:** 3
**Presentation:** 4
**Contribution:** 3
**Rating:** 8
**Confidence:** 3

**Summary:**

This paper proposed Frank-Wolfe (FW) algorithms with adaptive Bregman step-size strategies for a class of constrained optimization problems. The objective function satisfies the smooth adaptable property with respect to some Bregman distance function and the $q$-Holder error bound condition. For convex optimization, the proposed FW algorithms achieves global linear or sublinear convergence. For weakly-convex optimization, the proposed algorithms achieves global linear or sublinear convergence if the objective function satisfies the quadratic growth condition.

**Strengths:**

- The paper is well-motivated, organized and well-written.

- The proposed algorithm is parameter-free (with adaptive strategy for parameters) and provides global convergence guarantees under convex and nonconvex scenarios.

**Weaknesses:**

See questions below.

**Questions:**

- The global linear convergence rates of the proposed algorithms only holds for specific classes of functions: $q=2$ or $q= 1+ \nu$ or $\nu = 1$ or for initial iterates. Though the footnotes give explanation, the presented results in Table 1 may be misleading at the first glance. I suggest to add more description in the table.


- Theorem 5.2 requires $\rho < \mu$ and Theorem 5.2 requires $\rho < \mu \leq L$. These assumptions seem nontrivial.

(i) It is nice to see that Example D.4 satisfy the assumption.Is there a general class of functions satisfying the assumption?

(ii) $\rho, \mu$ and $L$ characterize the convexity of $f$ to some extent: $\rho$ is basically the $\rho$-smad parameter when $\phi$ is the quadratic function (typical Euclidean distance). $\mu$ is requiring the strong convexity over the solution set, which is related to the star-convexity. Is there relation between $\rho, \mu$ and $L$? For example, would some property of $\phi$ guarantee that $\rho \leq L$? Inituitive discussions are also welcome.

- Typo: Line 223, 'right-hand size'

---

> ### Author Response · Authors · 2025-11-20
> **Rebuttal by Authors**
>
> We thank the reviewer for the positive feedback on our work, particularly acknowledging that our paper is well-written with solid theoretical results.
> We also greatly appreciate the constructive comments.
>
> **(Q1) The global linear convergence rates of the proposed algorithms only holds for specific classes of functions: $q = 2$ or $q = 1 + \nu$ or $\nu = 1$ or for initial iterates. Though the footnotes give explanation, the presented results in Table 1 may be misleading at the first glance. I suggest to add more description in the table.**
>
> We have revised Table 1, including linear convergence and sublinear convergence.
>
> **(Q2) Theorem 5.2 requires $\rho < \mu$ and Theorem 5.2 requires $\rho < \mu \leq L$. These assumptions seem nontrivial.**
>
> **(i) It is nice to see that Example D.4 satisfy the assumption. Is there a general class of functions satisfying the assumption?**
>
> We have added more examples in Example E.5.
> While $\rho$-weak convexity is a global property, local $\mu$-quadratic growth holds only in a neighborhood. Because $\mu$ is defined locally, it is difficult to compute exactly, and hence verifying $\rho < \mu$ in practice would be a technical assumption.
> The (local) quadratic growth condition includes a wide range of function classes. For example, it covers (locally) strongly convex functions and functions satisfying the (local) Polyak-Łojasiewicz condition.
>
> Regarding $L$, it can be chosen freely by the user; if $\mu>L$ holds, we can simply redefine $L_0:=\max(\mu,L)$. Clearly, $(f,\phi)$ satisfies the $L_0$-smad property, and we can thereby ensure $\mu\leq L$.
>
> **(ii) $\rho$, $\mu$ and $L$ characterize the convexity of $f$ to some extent: $\rho$ is basically the $\rho$-smad parameter when $\phi$ is the quadratic function (typical Euclidean distance). $\mu$ is requiring the strong convexity over the solution set, which is related to the star-convexity. Is there relation between $\rho$, $\mu$ and $L$? For example, would some property of $\phi$ guarantee that $\rho \leq L$? Inituitive discussions are also welcome.**
>
> We acknowledge that your suggestion is correct and constitutes an excellent observation. In examples such as phase retrieval in Section 6, $\rho \leq L$ indeed holds. Both $\rho$-weak convexity and the $L$-smad condition can be interpreted as providing a lower and an upper bound, respectively, on the first-order approximation of $f$. For this reason, based solely on properties of $\phi$, it is in general difficult to guarantee $\rho \leq L$. As a simple remedy, one may redefine $L_0 := \max(\rho, L)$, in which case $\rho \leq L_0$ automatically holds.
>
> To simplify the discussion,  if we use $\phi(\cdot)=1/2||\cdot||^2$ and assume that $f$ is twice continuously differentiable and locally strongly convex, then $\mu \leq L$ holds.
>
> **(Q3) Typo: Line 223, 'right-hand size'**
>
> We've accepted your kind suggestions in the revised version.

---

> > ### Comment · Reviewer_229D · 2025-11-22
> >
> > Thank you to the authors for their response. I would like to keep my score.

---

### Official Review · Reviewer_4P4Y · 2025-11-01

**Soundness:** 3
**Presentation:** 3
**Contribution:** 3
**Rating:** 6
**Confidence:** 4

**Summary:**

This paper proposes new Frank–Wolfe (FW) algorithms that incorporate a Bregman-based adaptive step-size rule, removing the need for Lipschitz continuity of the gradient of the objective function. The authors extend FW to L-smooth adaptable functions, broadening applicability beyond standard L-smooth settings. They achieve sublinear to linear convergence rates under weaker conditions (e.g., Hölder error bound or local quadratic growth). This submission also provide both convex and weakly convex convergence analyses and an away-step variant for polytopes with provable linear rates. The paper include numerical validation on ℓp-loss and phase retrieval problems, showing empirical improvements over classical FW and its variants. In summary, it generalizes FW theory to handle non-Lipschitz and weakly convex objectives while retaining theoretical rigor and empirical competitiveness.

**Strengths:**

This submission has significant theoretical generalization. The relaxation from Lipschitz smoothness to relative smoothness is well motivated and aligns FW with modern Bregman-based optimization. It unifies and extends several prior frameworks, offering linear convergence under weaker assumptions.

The authors provide rigorous analysis. The theoretical sections in this submission are mathematically precise, covering: convex and weakly convex settings, both FW and away-step variants, and multiple growth conditions (HEB, quadratic growth). The mathematical proofs appear comprehensive and grounded in established geometric constants (pyramidal width, etc.). The authors also provided the adaptive Bregman step size. This adaptive scheme is elegant by extending previous results  and self-tunes both L and ν parameters. The termination proof (Remark 3.2) ensures practicality.

Moreover, the authors conduct numerical experiments and the empirical results are consistent with the theoretical analysis. The Experiments on non-Lipschitz settings convincingly show faster convergence and robustness where Euclidean FW fails.

**Weaknesses:**

There are no major weakness about this submission. The following is just some minor weaknesses:

The authors should add more numerical experiments. Only two primary experiments are shown (ℓp loss and phase retrieval). While results are positive, including comparisons on structured convex problems (e.g., LASSO, matrix completion) would better demonstrate generality.

This submission need more clarifications. Definitions (e.g., L-smad, kernel generating distances) are presented quickly with minimal intuition. Some long theorems could be summarized qualitatively before stating full formulas. Adding geometric or schematic illustrations (e.g., showing Bregman vs. Euclidean geometry) would enhance readability.

The authors should add more discussion of related work. The paper references many FW variants, but comparison to mirror descent or relative smoothness-based proximal algorithms is somewhat limited. Highlighting differences in oracle requirements and computational cost would clarify its niche.

**Questions:**

There are no other questions.

---

> ### Author Response · Authors · 2025-11-20
> **Rebuttal by Authors**
>
> We thank the reviewer for the general appreciation of our work as well as constructive comments that we address below.
>
> **(W1) The authors should add more numerical experiments. Only two primary experiments are shown ($\ell_p$ loss and phase retrieval). While results are positive, including comparisons on structured convex problems (e.g., LASSO, matrix completion) would better demonstrate generality.**
>
> We have already conducted additional numerical experiments, which are reported in Appendix F. Specifically, Appendix F presents additional experiments on nonnegative linear inverse problems, low-rank minimization, and NMF. The nonnegative linear inverse problems are convex optimization problems, whereas the low-rank minimization and NMF are nonconvex.
>
> **(W2) This submission need more clarifications. Definitions (e.g., $L$-smad, kernel generating distances) are presented quickly with minimal intuition. Some long theorems could be summarized qualitatively before stating full formulas. Adding geometric or schematic illustrations (e.g., showing Bregman vs. Euclidean geometry) would enhance readability.**
>
> Due to page limitations, we could not provide the intuitions of some definitions. For the $L$-smad property and kernel generating distances $\phi$, Lemma B.4 provides an intuitive understanding of the $L$-smad condition: it shows that the first-order approximation of $f$ is majorized by the Bregman distance $D_\phi$. We have added more discussions on the Bregman distance and the $L$-smad property on page 4.
>
> **(W3) The authors should add more discussion of related work. The paper references many FW variants, but comparison to mirror descent or relative smoothness-based proximal algorithms is somewhat limited. Highlighting differences in oracle requirements and computational cost would clarify its niche.**
>
> Due to page limitations, we could not discuss more mirror descent or relative smoothness-based proximal algorithms. We have added an overview of existing Bregman-based algorithm studies on Appendix A. The subproblems arising in the mirror descent and relative smoothness-based proximal algorithms are not necessarily solvable in closed form. It is known that only in a few cases (such as minimizing the KL divergence over the unit simplex) can they be computed more efficiently than projections or (Euclidean) proximal mappings, whereas in many situations their computational cost can in fact exceed that of projection or proximal mapping computations. In contrast, the Frank-Wolfe algorithm requires a linear minimization oracle (LMO), and LMOs are often much cheaper than projection oracles, as discussed in Section 1.

---

> > ### Comment · Reviewer_4P4Y · 2025-11-27
> >
> > Thanks for your rebuttal. I remain my original score.

---

### Official Review · Reviewer_gC98 · 2025-11-01

**Soundness:** 3
**Presentation:** 3
**Contribution:** 3
**Rating:** 6
**Confidence:** 2

**Summary:**

This paper proposes a Frank–Wolfe (FW) algorithm with an adaptive Bregman step-size strategy. The proposed algorithm covers relatively smooth and weakly convex setups, which are broader than the conventional $L$-smooth convex setup. It achieves local linear convergence under weak convexity and the local quadratic growth condition. Moreover, when the constraint set is a polytope, the paper proposes a variant of the away-step FW algorithm that overcomes the zigzagging issue of the classical FW algorithm. The latter algorithm also achieves local linear convergence under the aforementioned assumptions. While the convergence is local in the nonconvex case, both algorithms achieve global linear convergence when the objective function is convex and satisfies the Hölder error bound condition. Finally, the paper provides numerical experiments demonstrating its efficiency.

**Strengths:**

- To the best of this reviewer's understanding, the γ-update in line 5 of Algorithm 2 and line 8 of Algorithm 3, which is motivated by equation (2.2), is a novel aspect of the proposed algorithm. It seems that this idea enables the proposed algorithms to establish convergence guarantees for the considered broad setup, and the fact that the paper indeed provides corresponding convergence results is meaningful. The paper also presents convincing experimental results.
- The considered setup, L-smooth adaptable (i.e., both Lφ − f and Lφ + f are convex on the constraint set C), indeed appears to be an extension of the conventional setup, as mentioned in the paper.
- The paper seems to be overall well structured.

**Weaknesses:**

- **W1.** While this reviewer believes that the results are meaningful and worth the effort, the reviewer is not fully convinced of the overall technical novelty. In short, to the best of the reviewer’s understanding, this paper leverages line search to develop an adaptive step size, which is arguably a classical technique in optimization. The reviewer is curious whether there were any new challenges the authors needed to overcome to handle this setup. The reviewer would be happy to be corrected if something has been overlooked, as noted in the related question Q1.

**Questions:**

- **Q1.** Could the authors elaborate on any novel proof techniques or specific technical challenges they had to overcome while establishing the results, particularly those related to the extension of the setup?

- **Q2.** There exists a line of research that develops parameter-free methods without employing line search in other setups [1–5]. Do the authors anticipate particular challenges in removing the line search component from the proposed framework? If so, could they elaborate on the underlying reasons? (The reviewer thinks that this question may be out of the scope of this paper and is mainly motivated by the reviewer’s interest in hearing the authors’ intuition.)


[1] Yura Malitsky and Konstantin Mishchenko. Adaptive gradient descent without descent. International Conference on Machine Learning, 2020.

[2] Yura Malitsky and Konstantin Mishchenko. Adaptive proximal gradient method for convex optimization. Neural Information Processing Systems, 2024.

[3] Puya Latafat, Andreas Themelis, Lorenzo Stella, and Panagiotis Patrinos. Adaptive proximal algorithms for convex optimization under local Lipschitz continuity of the gradient. Mathematical Programming, 2024.

[4] Tianjiao Li and Guanghui Lan. A simple uniformly optimal method without line search for convex optimization. Mathematical Programming, 2025.

[5] Danqing Zhou, Shiqian Ma, and Junfeng Yang. AdaBB: Adaptive Barzilai-Borwein method for convex optimization. Mathematics of Operations Research, 2025.

---

> ### Author Response · Authors · 2025-11-20
> **Rebuttal by Authors**
>
> We thank the reviewer for positive comments on recognizing our solid theoretical analysis and convincing numerical results. We also greatly appreciate the constructive feedback and comments.
>
> **(W1) While this reviewer believes that the results are meaningful and worth the effort, the reviewer is not fully convinced of the overall technical novelty. In short, to the best of the reviewer’s understanding, this paper leverages line search to develop an adaptive step size, which is arguably a classical technique in optimization. The reviewer is curious whether there were any new challenges the authors needed to overcome to handle this setup. The reviewer would be happy to be corrected if something has been overlooked, as noted in the related question Q1.**
>
> **(Q1) Could the authors elaborate on any novel proof techniques or specific technical challenges they had to overcome while establishing the results, particularly those related to the extension of the setup?**
>
> Regarding what constitutes the novel proof techniques or specific technical challenges in the theoretical guarantee of the line search, as in classical line-search schemes, the key point is to ensure that the procedure is well-defined. We discuss this well-definedness issue in Remark 3.2. In particular, for each iteration of the line search, we need to prove the existence of $L$ and $\nu$ that satisfy the line search condition. Lemma B.7 provides this guarantee. As in previous works that assume $L$-smoothness, we similarly need to assume the $L$-smad condition; under this assumption and the strict convexity of $\phi$, we can show the existence of $\nu$ at the $t$-th iteration. Since the strict convexity of $\phi$ is a standard assumption when constructing a Bregman distance, this is a natural and reasonable assumption. For technically challenging points on a nonconvex optimization problem, we have added Remark 5.2.
>
> **(Q2) There exists a line of research that develops parameter-free methods without employing line search in other setups [1–5]. Do the authors anticipate particular challenges in removing the line search component from the proposed framework? If so, could they elaborate on the underlying reasons? (The reviewer thinks that this question may be out of the scope of this paper and is mainly motivated by the reviewer’s interest in hearing the authors’ intuition.)**
>
> **[1] Yura Malitsky and Konstantin Mishchenko. Adaptive gradient descent without descent. International Conference on Machine Learning, 2020.**
>
> **[2] Yura Malitsky and Konstantin Mishchenko. Adaptive proximal gradient method for convex optimization. Neural Information Processing Systems, 2024.**
>
> **[3] Puya Latafat, Andreas Themelis, Lorenzo Stella, and Panagiotis Patrinos. Adaptive proximal algorithms for convex optimization under local Lipschitz continuity of the gradient. Mathematical Programming, 2024.**
>
> **[4] Tianjiao Li and Guanghui Lan. A simple uniformly optimal method without line search for convex optimization. Mathematical Programming, 2025.**
>
> **[5] Danqing Zhou, Shiqian Ma, and Junfeng Yang. AdaBB: Adaptive Barzilai-Borwein method for convex optimization. Mathematics of Operations Research, 2025.**
>
> Designing a parameter-free variant of our method that does not rely on line search is certainly an appealing direction, but we believe it lies beyond the scope of the present paper.
>
> Please note that the proofs in [1–5] essentially rely on properties of the Euclidean distance, and their extensions to Bregman distance-based algorithms are rather limited. In particular, Bregman distances do not satisfy the triangle inequality, and they do not enjoy the direct relationship between the Euclidean norm and the Euclidean inner product that is available. Moreover, regarding possible extensions to Frank–Wolfe algorithms, we note that the convex combination update for $x_{t+1}$ is not necessarily well suited to designing line-search-free algorithms and to their convergence analysis based on Lyapunov functions.

---

### Official Review · Reviewer_2nKH · 2025-11-01

**Soundness:** 3
**Presentation:** 3
**Contribution:** 2
**Rating:** 4
**Confidence:** 4

**Summary:**

This work proposed a Frank-Wolfe algorithm for constrained optimization whose objective function is convex/weakly-convex and relatively smooth, the algorithm is further equipped with adaptive Bregman stepsize and away-step specifically for polytope cases. Convergence rates are provided both convex and weakly convex cases. Numerical experiments are complemented to verify the effectiveness of the proposed algorithms.

**Strengths:**

1. Extend the scope beyond convex and L-smooth, which is more general and fit more practical problems.
2. The stepsize is adaptive and "drop-in", which is easy to use.
3. The writing is clear, the flow of the work is easy to follow.

**Weaknesses:**

1. Even though extending into nonconvexity, the results still require some strong conditions like HEB, such conditions are still a bit strong, and lacks nontrivial examples throughout the work to verify the effectiveness.
2. The work seems to be a combination of FW with Bregman divergence, also many existing works on EB/QG conditions and relative smoothness, the novelty in terms of techniques may be limited a bit.
3. Line 304, "We will now establish faster convergence rates than O(1/t) up to linear convergence depending on the choice of parameters.", but I may argue that the acceleration comes from the problem setting (additional EB condition compared to the vanilla convex setting), rather than your parameter setting.
4. For the nonconvex part, Theorem 5.2 and 5.3 further require $\rho<\mu$, which has not been verified, it would be helpful to include discussion or examples illustrating when this inequality holds, or how one might estimate these quantities in practice. This would clarify the scope of applicability of the nonconvex guarantees.

**Questions:**

See above

---

> ### Author Response · Authors · 2025-11-20
> **Rebuttal by Authors**
>
> We thank the reviewer for acknowledging that our paper is clear and that our algorithm has practical aspects. We also appreciate constructive feedback.
>
> **(W1) Even though extending into nonconvexity, the results still require some strong conditions like HEB, such conditions are still a bit strong, and lacks nontrivial examples throughout the work to verify the effectiveness.**
>
> We have added a discussion on the assumptions. Specifically, Remark 5.2 explains why these assumptions are needed, and Example E.5 provides examples that satisfy them.
> In existing research, the Frank-Wolfe algorithm is typically analyzed under the assumption of strong convexity. Due to relaxing the assumption from strong convexity to weak convexity, we encounter difficulties such as those described in Remark 5.2. Since (local) strongly convex functions satisfy the (local) quadratic growth condition, but not vice versa, the latter (that is, our) assumption is, in fact, quite mild for proving linear convergence. Moreover, the quadratic growth condition is a special case of the HEB condition. This fact implies that the Frank-Wolfe algorithm can be applied to a wide range of problems. Regarding weak convexity, which yields a relatively tractable subclass of nonconvex functions, it should be noted that this class is still much broader than the class of convex functions.
>
> **(W2) The work seems to be a combination of FW with Bregman divergence, also many existing works on EB/QG conditions and relative smoothness, the novelty in terms of techniques may be limited a bit.**
>
> The linear convergence result for nonconvex optimization problems is, to the best of our knowledge, the first such result for the Frank-Wolfe algorithm. We have added Remark 5.2, which explains the technically challenging point. We believe it is significant that we have demonstrated that the theoretical guarantee of linear convergence holds even for certain types of nonconvex problems, as illustrated in Example E.5, which includes functions used in image restoration.
>
>
> **(W3) Line 304, "We will now establish faster convergence rates than $O(1/t)$ up to linear convergence depending on the choice of parameters.", but I may argue that the acceleration comes from the problem setting (additional EB condition compared to the vanilla convex setting), rather than your parameter setting.**
>
> More precisely, we have revised it to "We will now establish faster convergence rates than $O(1/t)$ up to linear convergence depending on $\nu$ in Equation B.2 and $q$ in Definition 2.4."
>
> **(W4) For the nonconvex part, Theorem 5.2 and 5.3 further require $\rho < \mu$, which has not been verified, it would be helpful to include discussion or examples illustrating when this inequality holds, or how one might estimate these quantities in practice. This would clarify the scope of applicability of the nonconvex guarantees.**
>
> While $\rho$-weak convexity is a global property, local $\mu$-quadratic growth holds only in a neighborhood. Because $\mu$ is defined locally, it is difficult to compute exactly, and hence verifying $\rho < \mu$ in practice would be a technical assumption. If $\rho < \mu$ fails, that is, $\rho$ is large and $\mu$ is small, then functions would not satisfy properties like local convexity. We have added more examples satisfying $\rho < \mu$ in Example E.5.

---

### Official Review · Reviewer_BkRp · 2025-11-02

**Soundness:** 3
**Presentation:** 2
**Contribution:** 3
**Rating:** 6
**Confidence:** 2

**Summary:**

The paper studies Frank-Wolfe algorithm under relatively smooth and (weakly-) convex assumptions. The authors proposes new stepsizes that utilize Bregman distance, which generalizes standard Euclidian setting. The paper derive convergence guarantees: sublinear and local linear rates under weaker assumptions than classical Lipschitz‐gradient smoothness and strong convexity, and demonstrate experimentally that their methods outperform existing FW algorithms.

**Strengths:**

1. The paper extends the analysis of FW‐type methods to the class of (L-smad) functions and weakly convex objectives. These results are new and correct.
2. The proposed Adaptive Bregman step-size strategy automatically adapts to L-smad constant, which does not require extensive hyper-parameter search or estimation of $L$.
3. The paper shows not only global sublinear convergence but also local linear convergence in the convex case under a Hölder error-bound condition (HEB) and in the nonconvex case under a local quadratic growth condition. This gives stronger theoretical guarantees than many prior FW analyses.

**Weaknesses:**

1. Weak-convexity, quadratic growth and HEB assumptions while being more general then previous assumption, still are strong. Under these assumptions, linear convergence rate is not surprising.

**Questions:**

1. How sensitive is the performance of the adaptive Bregman step‐size strategy (Algorithm 2) to the parameters $\beta, \tau$,  (which control the inner loop for estimating $M$ and $\kappa$)? Do the authors provide guidelines on tuning those for new problems?
2. What is a complexity of Procedure step_size in Algorithm2?
3. Do short step-sizes perform better than adaptive step-sizes on numerical experiments?
4. In the nonconvex (weakly convex) setting, the local linear convergence assumes a local $\mu$-quadratic growth condition. Practically, how can one check or ensure this condition holds in a given application? Also, given that there is a linear local convergence of the method, how one can identify when the algorithm reach this local neighborhood?

---

> ### Author Response · Authors · 2025-11-20
> **Rebuttal by Authors**
>
> We thank the reviewer for constructive comments on recognizing our proposed algorithm and our solid theoretical analysis. We also greatly appreciate the constructive feedback and comments.
>
> **(W1) Weak-convexity, quadratic growth and HEB assumptions while being more general then previous assumption, still are strong. Under these assumptions, linear convergence rate is not surprising.**
>
> We have added a discussion on the assumptions. Specifically, Remark 5.2 explains why these assumptions are needed, and Example E.5 provides examples that satisfy them.
> In existing research, the Frank-Wolfe algorithm is typically analyzed under the assumption of strong convexity. Due to relaxing the assumption from strong convexity to weak convexity, we encounter difficulties such as those described in Remark 5.2. Since (local) strongly convex functions satisfy the (local) quadratic growth condition, but not vice versa, the latter (that is, our) assumption is, in fact, quite mild for proving linear convergence. Moreover, the quadratic growth condition is a special case of the HEB condition. This fact implies that the Frank-Wolfe algorithm can be applied to a wide range of problems. Regarding weak convexity, which yields a relatively tractable subclass of nonconvex functions, it should be noted that this class is still much broader than the class of convex functions.
>
> **(Q1) How sensitive is the performance of the adaptive Bregman step‐size strategy (Algorithm 2) to the parameters $\beta$, $\tau$, (which control the inner loop for estimating $M$ and $\kappa$)? Do the authors provide guidelines on tuning those for new problems?**
>
> We provide a bound on the total number of evaluations of Algorithm 2 in Theorem 3.3. Intuitively, the line search procedure would finish by $\lceil\max(1 - \log\eta/\log\tau, \log\nu/\log\beta + 1)\rceil$ iterations at each iteration except the first iteration. Using the same argument as in Pedregosa et al. (2020), we further obtain that, asymptotically, no more than 16\% of the iterations require more than one evaluation of the line search procedures when $\eta = 0.9$ and $\tau = 2$. In practice, $\nu$ rarely becomes excessively small, and the above discussion indicates that the line search is not particularly sensitive to the choice of its parameters. As long as $\beta$, $\tau$, and $\eta$ are chosen within a reasonable (not extreme) range, the line search typically terminates within 1-2 iterations in most cases.
>
> **(Q2) What is a complexity of Procedure step\_size in Algorithm2?**
>
> We provide a bound on the total number of evaluations of Algorithm 2 in Theorem 3.3. Pedregosa et al. (2020) analyzed the complexity of an adaptive line search strategy. By following essentially the same argument as Theorem 1 in Pedregosa et al. (2020), we can derive the complexity of Algorithm 2 as $n_t \leq \max((1 - \log\eta/\log\tau)(t+1) + \max(\log \tau L/L_{-1}, 0), (\log\nu/\log\beta + 1)(t+1))$, where $n_t$ is the total number of evaluations of the sufficient decrease condition and $L_{-1}$ is the initial $L$-smad estimate.
>
> **(Q3) Do short step-sizes perform better than adaptive step-sizes on numerical experiments?**
>
> Even if the exact value of $L$ is known, short step-sizes are not necessarily more efficient than adaptive step-sizes. In fact, short step-sizes are tied to a global upper bound $L$, whereas adaptive step-sizes can exploit more favorable local smoothness and thus adjust to the local behavior of the objective function. This observation applies both in the Bregman distance case and in the Euclidean distance case. Moreover, the exact value of $L$ is generally not computable. Therefore, from a practical standpoint, adaptive step-sizes are preferable.
>
> **(Q4) In the nonconvex (weakly convex) setting, the local linear convergence assumes a local $\mu$-quadratic growth condition. Practically, how can one check or ensure this condition holds in a given application? Also, given that there is a linear local convergence of the method, how one can identify when the algorithm reach this local neighborhood?**
>
> We verify the local quadratic growth condition in Example E.5. In practice, however, this condition is difficult to check because it requires explicit information about the solution set $X^*$. As a more practical alternative, it is sufficient to establish a (local) Polyak-Łojasiewicz inequality.
>
> Regarding linear convergence, one can monitor the primal gap or the Frank-Wolfe gap: a sudden acceleration in their decay typically indicates that the iterates have entered the local neighborhood. In particular, the FW gap does not rely on any knowledge of the optimal solution or optimal value, and thus can be used as a practical criterion.

---

### Author Response · Authors · 2025-11-27

Dear ACs and Reviewers,

We hope these revisions address your concerns and improve the clarity and quality of the manuscript. Thank you again for your insightful comments, which have been instrumental in refining our work. Please do not hesitate to let us know if there are further areas requiring clarification or improvement. If further comments are received, revisions will require time, so we would appreciate it if you could provide your comments as soon as possible.

Regards,

Authors

---

### Author Response · Authors · 2025-12-03
**Additional Comments by Authors**

Dear ACs,

The reviewers acknowledge the novelty of our work, in particular the linear convergence of Frank–Wolfe for nonconvex optimization and the proposed adaptive step-size strategy. All reviewers also recognize our theoretical results and the proposed step-size strategy, and highlight the generalization of Frank–Wolfe algorithms to the L-smad and weakly convex settings as a key contribution. Reviewers 2nKH, 229D, and 4P4Y describe the paper as well-written, well-structured, and well-motivated, while reviewers gC98, and 4P4Y emphasize that the experimental results are consistent with the theoretical analysis. Moreover, the reviewers did not raise any comments that are impossible to address; instead, they provided constructive feedback.
- ***More Discussions on assumptions for nonconvex optimization:*** Regarding the questions on the assumptions used in our theoretical analysis, we have added a discussion on the assumptions. Specifically, Remark 5.2 explains why these assumptions are needed. In existing research, the Frank–Wolfe algorithm is typically analyzed under the assumption of strong convexity. By relaxing this assumption from strong convexity to weak convexity, we encounter difficulties such as those described in Remark 5.2. Since (local) strongly convex functions satisfy the (local) quadratic growth condition, but not vice versa, the latter (that is, our) assumption is, in fact, quite mild for proving linear convergence. Moreover, the quadratic growth condition is a special case of the HEB condition, which implies that the Frank–Wolfe algorithm can be applied to a wide range of problems. Regarding weak convexity, which yields a relatively tractable subclass of nonconvex functions, it should be noted that this class is still much broader than the class of convex functions.
- ***More examples:*** We also added concrete examples (Example E.5) that satisfy assumptions for nonconvex optimization.
- ***Additional theoretical result (complexity):*** We provide a bound on the total number of evaluations of our adaptive step-size strategy (Algorithm 2) in Theorem 3.3. This indicates that Algorithm 2 is not particularly sensitive to the choice of its parameters.
- ***Novelty:*** Reviewer 2nKH has listed limited novelty in terms of techniques as a weakness. However, as other reviewers have noted, our method and its analysis exhibit meaningful novelty. The Bregman adaptive step-size strategy, which estimates $L$ and $\nu$, is novel. Regarding the away-step FW, there is no similar approach in the literature; our algorithm is entirely new.  The linear convergence result for nonconvex optimization problems is, to the best of our knowledge, the first such result for the Frank-Wolfe algorithm. We have added Remark 5.2, which explains the technically challenging point. We believe it is significant that we have demonstrated that the theoretical guarantee of linear convergence holds even for certain types of nonconvex problems, as illustrated in Example E.5, which includes functions used in image restoration.

We believe that we have adequately addressed all of the other reviewers’ comments as well. Unfortunately, we did not receive any response to our rebuttal within the discussion period from reviewers BkRp, 2nKH, and gC98.

Regards,

Authors

---

### Meta-Review · Area_Chair_Jdga · 2025-12-16

**Summary:**

The reviewers appreciate the authors' work to extend the known FW theory to the class of L-smad functions and weakly convex functions. They acknowledge rigorous analysis, nice presentation, and the new adaptive Bregman step-size strategy. On the other hand, they also recognize that the work might be a straightforward extension of the classical FW theory with Bregman divergence to a slightly relaxed class of functions with (generalized) quadratic growth, utilizing the standard line-search approach.

**Reviewer Concerns:**

I think almost all concerns were addressed by the authors and already incorporated into the paper. However, I'm not fully convinced by the resolve of the weakness that $\rho < \mu$, raised by the reviewers. The authors address this weakness with Example E.5. I hope that the authors would clarify it in more detail with a property showing that this assumption is not restrictive in general.

**Reviewer Scores:**

Almost all scores are positive and already recommend acceptance. It is not obvious whether Reviewer 2nKH would increase the score.

---

### Decision · Program_Chairs · 2026-01-26

Accept (Poster)